# A concerted neuron–astrocyte program declines in ageing and schizophrenia

Emi Ling[1,2 ✉], James Nemesh[1,2], Melissa Goldman[1,2], Nolan Kamitaki[1,2,3], Nora Reed[1,2], Robert E. Handsaker[1,2], Giulio Genovese[1,2], Jonathan S. Vogelgsang[4,5], Sherif Gerges[1,2], Seva Kashin[1,2], Sulagna Ghosh[1,2], John M. Esposito[4], Kiely Morris[4], Daniel Meyer[1,2], Alyssa Lutservitz[1,2], Christopher D. Mullally[1,2], Alec Wysoker[1,2], Liv Spina[1,2], Anna Neumann[1,2], Marina Hogan[1,2], Kiku Ichihara[1,2], Sabina Berretta[1,4,5,6,7 ✉] & Steven A. McCarroll[1,2,7 ✉]

Human brains vary across people and over time; such variation is not yet understood in cellular terms. Here we describe a relationship between people's cortical neurons and cortical astrocytes. We used single-nucleus RNA sequencing to analyse the prefrontal cortex of 191 human donors aged 22–97 years, including healthy individuals and people with schizophrenia. Latent-factor analysis of these data revealed that, in people whose cortical neurons more strongly expressed genes encoding synaptic components, cortical astrocytes more strongly expressed distinct genes with synaptic functions and genes for synthesizing cholesterol, an astrocyte-supplied component of synaptic membranes. We call this relationship the synaptic neuron and astrocyte program (SNAP). In schizophrenia and ageing—two conditions that involve declines in cognitive flexibility and plasticity[1,2]—cells divested from SNAP: astrocytes, glutamatergic (excitatory) neurons and GABAergic (inhibitory) neurons all showed reduced SNAP expression to corresponding degrees. The distinct astrocytic and neuronal components of SNAP both involved genes in which genetic risk factors for schizophrenia were strongly concentrated. SNAP, which varies quantitatively even among healthy people of similar age, may underlie many aspects of normal human interindividual differences and may be an important point of convergence for multiple kinds of pathophysiology.

In natural, non-laboratory settings—in which individuals have diverse genetic inheritances, environments and life histories, as humans do—almost all aspects of biology exhibit quantitative variation across individuals[3]. Natural variation makes it possible to observe a biological system across many contexts and potentially learn underlying principles that govern its function[4,5].

Here we sought to recognize changes that multiple cell types in the human brain characteristically implement together. The need to be able to recognize tissue-level gene-expression programs comes from a simple but important idea in the physiology of the brain and other tissues: cells of different types collaborate to perform essential functions, working together to construct and regulate structures such as synaptic networks.

We analysed the prefrontal cortex of 191 human brain donors using single-nucleus RNA sequencing (snRNA-seq) and developed a computational approach, based on latent-factor analysis, to recognize commonly recurring multicellular gene-expression patterns in such data. Tissue-level programs of which the expression varies across individuals could provide new ways to understand healthy brain function and also brain disorders, as disease processes probably act through endogenous pathways and programs in cells and tissues.

A longstanding challenge in genetically complex brain disorders is to identify the aspects of brain biology on which disparate genetic effects converge; here we applied this idea to try to better understand schizophrenia.

### snRNA-seq analysis of the dlPFC

We analysed the dorsolateral prefrontal cortex (dlPFC; Brodmann area 46), which serves working memory, attention, executive functions and cognitive flexibility[6], abilities that decline in schizophrenia and with advancing age[1,2]. Analyses included frozen post-mortem dlPFC samples from 191 donors (aged 22–97 years, median 64 years), including 97 without known psychiatric conditions and 94 affected by schizophrenia (Extended Data Fig. 1 and Supplementary Table 1). To generate data that were well controlled across donors and therefore amenable to integrative analysis, we processed a series of 20-donor sets of dlPFC tissue, each as a single pooled sample (or village[7]; Fig. 1a) and then, during computational analysis, we used combinations of many transcribed single-nucleotide polymorphisms (SNPs) to identify the source donor of each nucleus (Fig. 1a,b and Extended Data Fig. 2).

[1]Stanley Center for Psychiatric Research, Broad Institute of MIT and Harvard, Cambridge, MA, USA. [2]Department of Genetics, Harvard Medical School, Boston, MA, USA. [3]Department of Biomedical Informatics, Harvard Medical School, Boston, MA, USA. [4]McLean Hospital, Belmont, MA, USA. [5]Department of Psychiatry, Harvard Medical School, Boston, MA, USA. [6]Program in Neuroscience, Harvard Medical School, Boston, MA, USA. [7]These authors jointly supervised this work: Sabina Berretta, Steven A. McCarroll. ✉e-mail: eling@broadinstitute.org; sberretta@mclean.harvard.edu; smccarro@broadinstitute.org

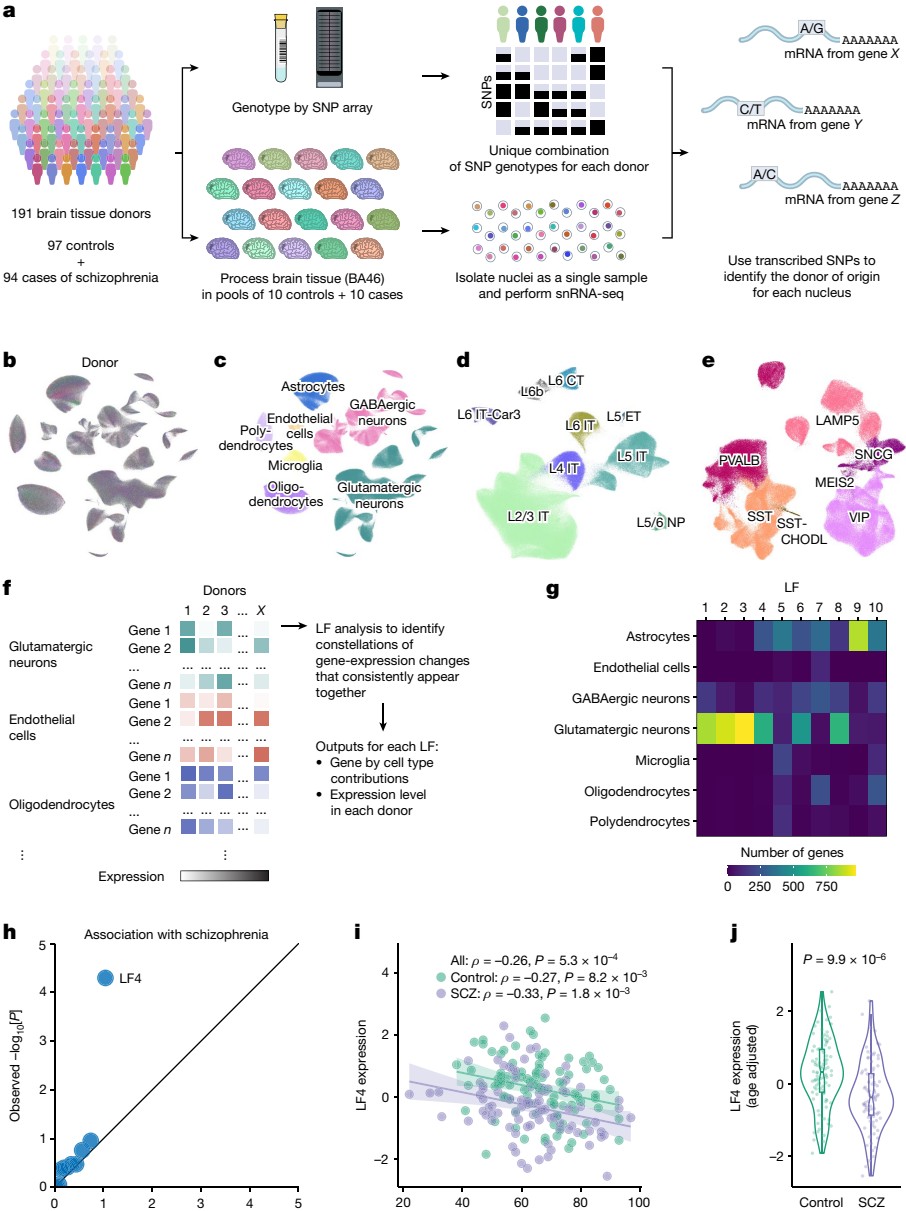

**Fig. 1 | Identification of concerted multicellular gene-expression changes common to schizophrenia and ageing. a**, Generation of snRNA-seq data, in a series of 20-donor 'villages'. The diagram was created using images by thekua (person icon), B. Lachner (laboratory tools) and pnx (brain exterior side view) under a Creative Commons licence CC0 1.0. **b**, Uniform manifold approximation and projection (UMAP; coloured by donor) analysis of the RNA-expression profiles of 1,217,965 nuclei analysed from 191 donors. **c**, Assignments of nuclei to cell types (same projection as in **b**). **d**,**e**, Assignments of nuclei to glutamatergic ($n$ = 524,186) (**d**) and GABAergic ($n$ = 238,311) (**e**) neuron subtypes. CT, corticothalamic; ET, extratelencephalic; IT, intratelencephalic; NP, near-projecting. **f**, Latent factor analysis. Cell-type-resolution expression data from all donors and cell types were combined into a single analysis. Latent factor analysis identified constellations of gene-expression changes that consistently appeared together. **g**, The cell type specificity of the latent factors inferred from 180 donors, shown as the cell type distributions of the 1,000 most strongly loading gene–cell type combinations per factor. Factors 4–7 and 10 are strongly driven by gene-expression co-variation spanning multiple cell types. **h**, The association of schizophrenia (SCZ) with interindividual variation in the expression levels of the ten latent factors in **g**, shown as a quantile–quantile plot comparing the observed schizophrenia associations with the ten factors ($-\log_{10}[P]$) to the distribution of association statistics expected by chance; only LF4 significantly associated with schizophrenia. See also Supplementary Fig. 6. **i**, The relationship between quantile-normalized LF4 donor expression levels and age (Spearman's $\rho$; $n$ = 180 donors). The shaded regions represent the 95% confidence intervals. **j**, Quantile-normalized LF4 donor scores ($n$ = 93 controls, 87 cases), adjusted for age. The $P$ value was calculated using a two-sided Wilcoxon rank-sum test. For the violin plot, the box limits show the interquartile range, the whiskers show 1.5× the interquartile interval, the centre lines show the median values and the notches show the confidence intervals around the median values.

Each of the 1,217,965 nuclei was classified into one of seven cell types—glutamatergic neurons (43% of all nuclei), GABAergic neurons (20%), astrocytes (15%), oligodendrocytes (12%), polydendrocytes (oligodendrocyte progenitor cells, 5.5%), microglia (3.6%) and endothelial cells (1.3%) (Fig. 1c and Supplementary Fig. 1)—as well as neuronal subtypes defined in earlier taxonomies (Fig. 1d,e and Supplementary Figs. 2 and 3). Each donor contributed nuclei of all types and subtypes (Supplementary Figs. 1, 4 and 5), although subsequent analyses excluded 11 atypical samples (Supplementary Fig. 1d).

## Inference of multicellular gene programs

The data revealed substantial interindividual variation in cell-type-specific gene expression levels, with highly expressed genes in each cell type exhibiting a median coefficient of variation (across donors) of about 15%.

Interindividual variation in gene expression almost certainly arises from cell-type-specific gene-expression programs, and could in principle also be shaped by concerted changes in multiple cell types. To identify such relationships, we applied latent factor analysis, a form of machine learning that infers underlying factors from the tendency of many measurements to fluctuate together[8]. Critically, we analysed cell-type-resolution data from all cell types at once, using interindividual variation to enable the recognition of relationships between expression patterns in different cell types (Fig. 1f). Each inferred factor was defined by a set of gene-by-cell-type loadings (revealing the distinct genes it involves in each cell type) and a set of expression levels (of the factor) in each donor (Fig. 1f).

Ten latent factors together explained 30% of interindividual variation in gene expression levels; these factors appeared to be independent of one another in their gene use patterns (loadings) and their expression levels across the individual donors (Extended Data Fig. 3a–d). Interindividual variation in the factors' inferred expression levels arose from interindividual variation within each 20-donor experimental set (Extended Data Fig. 3e). Each factor was primarily driven by gene expression in one or a few cell types (Fig. 1g).

Schizophrenia was associated with just one of these latent factors (LF4) (Fig. 1h, Extended Data Fig. 4a–e and Supplementary Table 2)—a factor that was also associated with donor age (Fig. 1i). Donors with and without schizophrenia both exhibited the decline in LF4 with age (Fig. 1i and Extended Data Fig. 1c,d). Joint regression analysis confirmed independent decreases in LF4 expression by age and in schizophrenia, and detected no effect of sex (Supplementary Table 3).

Factors similar to LF4 emerged in all analyses testing LF4's robustness to analysis parameters (Supplementary Fig. 6). The LF4 expression scores of individuals also did not correlate with medication use, time of day at death, post-mortem interval or sequencing depth (Extended Data Fig. 4f–k). We also found evidence that the LF4 constellation of gene-expression changes manifests at the protein level (Supplementary Fig. 7).

## Neuronal and astrocyte genes driving LF4

Of the 1,000 gene/cell-type expression traits with the strongest LF4 loadings, 99% involved gene expression in glutamatergic neurons (610), GABAergic neurons (125) or astrocytes (253) (Fig. 1g). LF4 involved similar genes and expression effect directions in glutamatergic and GABAergic neurons, but a distinct set of genes and effect directions in astrocytes (Fig. 2a and Extended Data Fig. 4l). To identify biological processes in LF4, we applied gene set enrichment analysis (GSEA)[9] to the LF4 gene loadings, separately for each cell type.

In both glutamatergic and GABAergic neurons, LF4 involved increased expression of genes with synaptic functions (Fig. 2b, Extended Data Fig. 4m and Supplementary Table 4). The most strongly enriched synaptic annotations for both glutamatergic and GABAergic neurons involved the synaptic vesicle cycle and the presynaptic compartment; the core genes driving these enrichments encoded components of the SNARE complex and their interaction partners (*STX1A*, *SNAP25* and *SYP*), effectors and regulators of synaptic vesicle exocytosis (*SYT11*, *RAB3A* and *RPH3A*) and other synaptic vesicle components (*SV2A* and *SYN1*). In glutamatergic neurons, LF4 also appeared to involve genes encoding postsynaptic components, including signalling proteins (*PAK1*, *GSK3B* and *CAMK4*) and ion channels and receptors (*CACNG8*, *KCNN2*, *CHRNB2*, *GRM2* and *GRIA3*).

People with schizophrenia and people of advanced age exhibited reduced levels of synapse-related gene expression by cortical neurons of all types (Fig. 2c and Extended Data Fig. 5).

In astrocytes, LF4 involved gene-expression effects distinct from those in neurons (Fig. 2a and Extended Data Fig. 4l). Gene sets with roles in fatty acid and cholesterol biosynthesis and export, including genes encoding the SREBP1 and SREBP2 transcription factors and their regulators and targets, were positively correlated with LF4 and underexpressed in the cortical astrocytes of donors with schizophrenia (Fig. 2d and Supplementary Table 4) or advanced age (Extended Data Fig. 6a). These effects appeared to be specific to astrocytes relative to other cell types (Extended Data Fig. 7).

## Concerted neuron–astrocyte expression

To understand these results in terms of specific biological activities, we focused on gene sets corresponding to neuronal synaptic components and three kinds of astrocyte activities: adhesion to synapses, uptake of neurotransmitters and cholesterol biosynthesis (see the 'Selected gene sets' section of the Methods).

The proportion of astrocyte gene expression devoted to each of these three astrocyte activities was strongly correlated with the proportion of neuronal gene expression devoted to synaptic components (Fig. 2e and Supplementary Fig. 8), even after adjusting for age and case–control status (Extended Data Fig. 8). Donors with schizophrenia, as well as donors with advanced age, tended to have reduced expression of these genes (Fig. 2e and Extended Data Fig. 6).

As this gene expression program involves concerted effects on the expression of (distinct) genes for synaptic components in neurons and astrocytes, we call it SNAP, although it also involves genes with unknown functions and involves more modest expression effects in additional cell types. We used the LF4 expression scores of donors to measure SNAP expression.

## Astrocyte gene programs and SNAP

To better appreciate the astrocytic contribution to SNAP, we further analysed the RNA-expression data from 179,764 individual astrocytes. The analysis readily recognized a known, categorical distinction among three subtypes of adult cortical astrocytes: protoplasmic astrocytes, which populate the grey matter and were the most abundant subtype; fibrous astrocytes; and interlaminar astrocytes (Fig. 3a and Extended Data Fig. 9a–d). Neither schizophrenia nor age were associated with variation in the relative abundances of these subtypes (Extended Data Fig. 9e,f).

We next identified latent factors that collectively explained 25% of quantitative gene-expression variation among individual astrocytes (using consensus non-negative matrix factorization (cNMF)[10], which better scaled to the single-cell-level data) (Extended Data Fig. 10a,b). The factors appeared to capture diverse biological activities, including translation (cNMF1); zinc and cadmium ion homeostasis (cNMF7); and inflammatory responses (cNMF8) (Supplementary Table 5). One factor (cNMF2) corresponded to the astrocyte component of SNAP (Extended Data Fig. 10c–e and Supplementary Table 6); the strong co-expression relationships in SNAP were therefore robust to the computational approach used (Extended Data Fig. 10c–e and Supplementary Fig. 9).

As cNMF2 is informed by variation in the single-astrocyte expression profiles, we consider it a more precise description of the astrocyte-specific gene-expression effects in SNAP, and refer to it here as SNAP-a. Across donors, the average astrocyte expression of SNAP-a was associated even more strongly with schizophrenia case–control status and with age (Fig. 3b–e and Extended Data Fig. 10f–i).

The strongest positive gene-set associations to SNAP-a involved adhesion to synaptic membranes and intrinsic components of synaptic membranes (Supplementary Table 5). The 20 genes most strongly

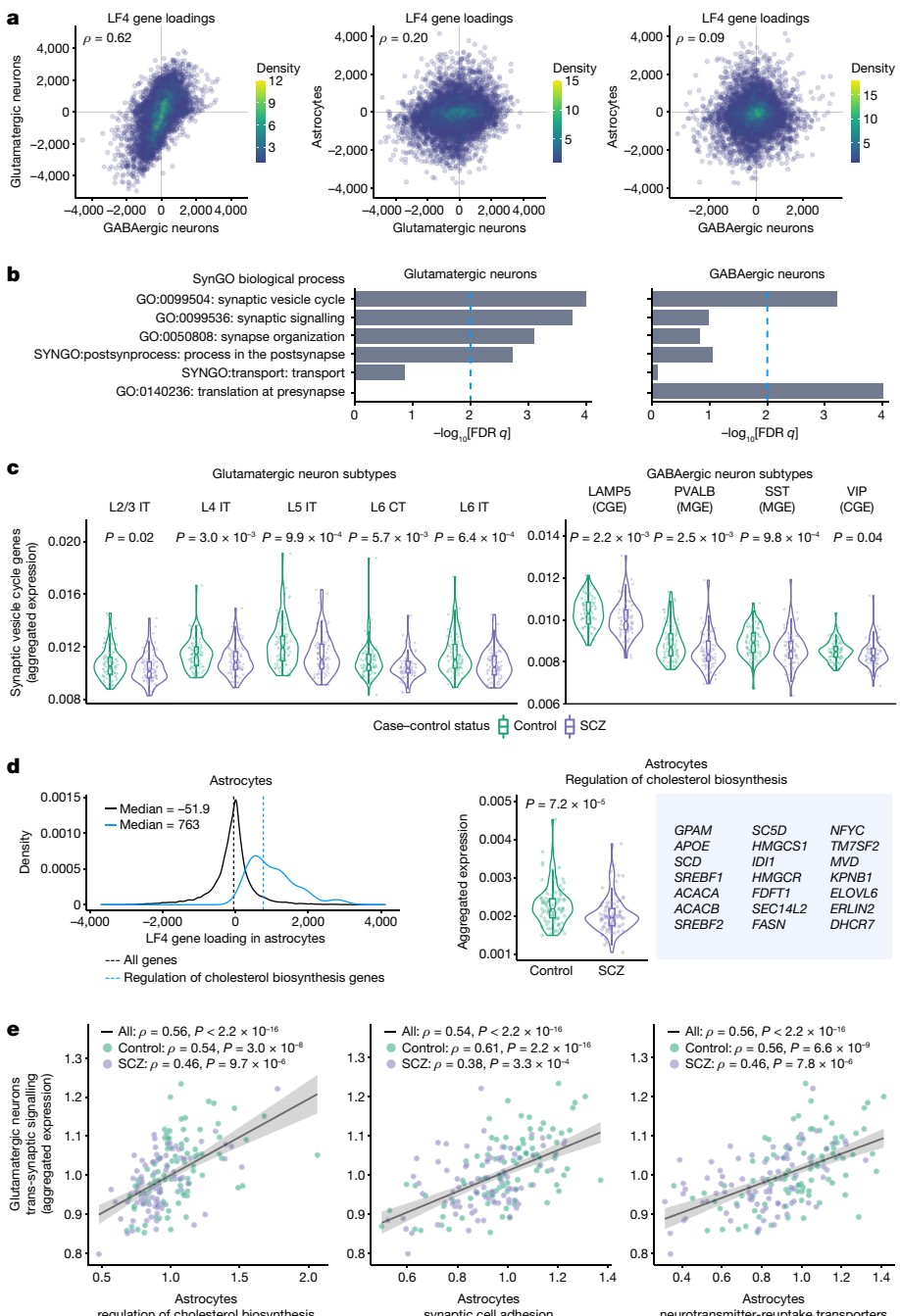

**Fig. 2 | Genes recruited by SNAP in neurons and astrocytes. a**, Comparisons of SNAP gene recruitment between cell types. For each pairwise cell type comparison, the LF4 gene loadings of all genes expressed (≥1 unique molecular identifier (UMI) per $10^5$) in both cell types in the comparison (Spearman's $\rho$; $n = 10,346, 11,232$ and $11,217$ genes, respectively) are shown. **b**, Concentrations of synaptic gene sets (as annotated by SynGO) in LF4's neuronal components. FDR, false-discovery rate. **c**, The fraction of gene expression (UMIs) devoted to synaptic-vesicle-cycle genes in subtypes of glutamatergic and GABAergic neurons, across 180 donors. $P$ values for case–control comparisons were calculated using two-sided Wilcoxon rank-sum tests. CGE, caudal ganglionic eminence; MGE, medial ganglionic eminence. **d**, The distributions of astrocyte LF4 gene loadings for all expressed genes (black; $n = 18,347$) and genes annotated for functions in cholesterol biosynthesis (blue; $n = 21$; hereafter, cholesterol biosynthesis genes according to their GO annotation, although subsets contribute to cholesterol export and/or to synthesis of additional fatty acids)

(left). Right, the proportion of astrocytic gene expression devoted to the annotated cholesterol biosynthesis genes shown, across 180 donors. The $P$ value was calculated using a two-sided Wilcoxon rank-sum test. **e**, Concerted gene-expression variation in neurons and astrocytes. The relationships (across 180 donors) between astrocytic gene expression related to three biological activities (synapse adhesion, neurotransmitter uptake and cholesterol biosynthesis) and neuronal gene expression related to synapses (Spearman's $\rho$). Quantities plotted are the fraction of all detected nuclear mRNA transcripts (UMIs) derived from these genes in each donor's astrocytes ($x$ axis) or neurons ($y$ axis) relative to the median expression among control donors. The shaded regions represent the 95% confidence intervals for the estimated slopes. For the box plots nested within the violin plots in **c** and **d**, the box limits show the interquartile range, the whiskers show 1.5× the interquartile interval, the centre line shows the median value and the notches show the confidence intervals around the median values.

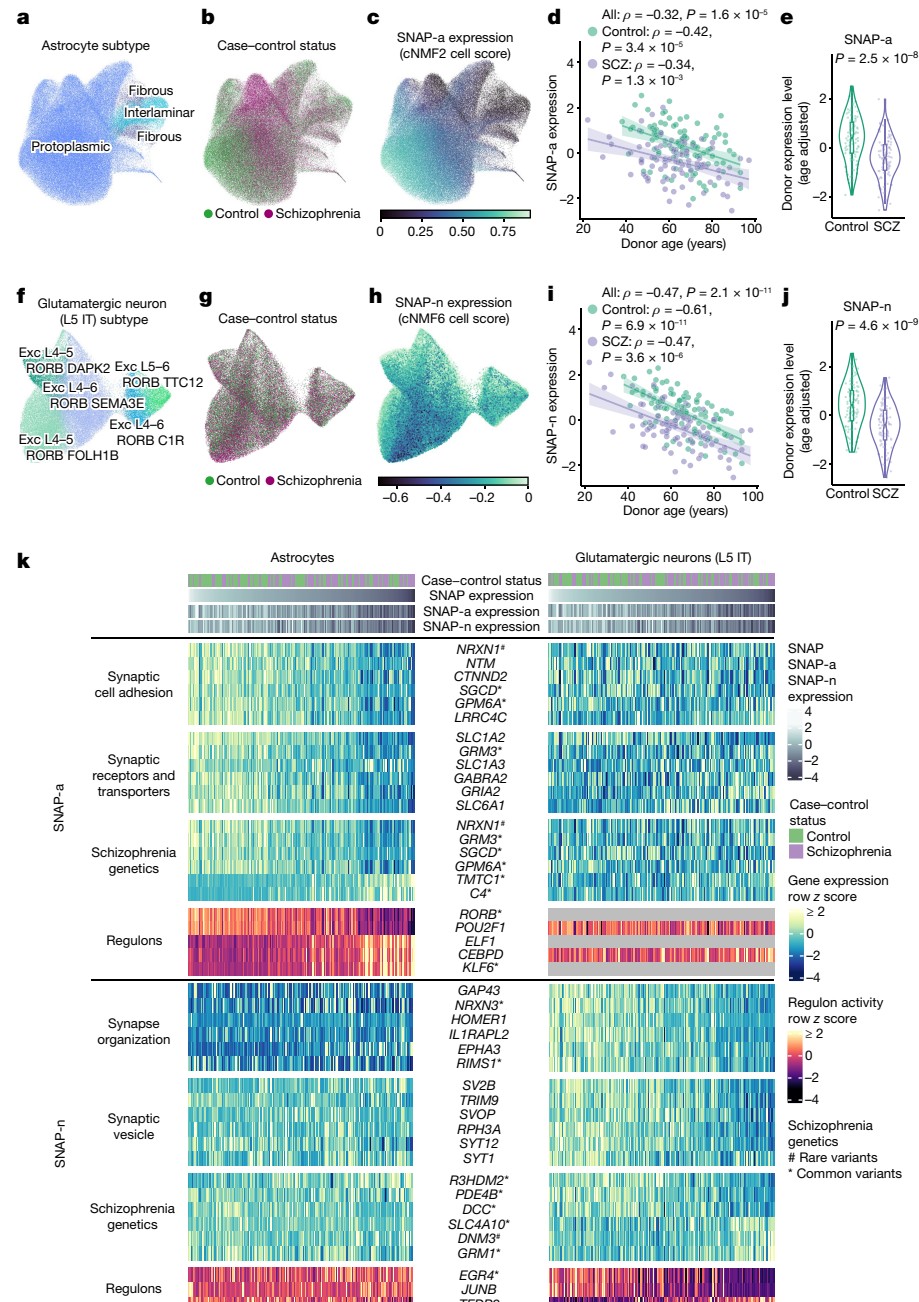

**Fig. 3 | Biological states and transcriptional programs of astrocytes and L5 IT glutamatergic neurons in schizophrenia. a–c**, UMAP analysis of RNA expression patterns from 179,764 astrocyte nuclei from 180 donors. Nuclei are coloured by astrocyte subtype (**a**), schizophrenia affected/unaffected status (**b**) and expression of the astrocyte component of SNAP (SNAP-a) (**c**). **d**, The relationship between donor quantile-normalized SNAP-a expression scores and age (Spearman's $\rho$). $n$ = 180 donors. The shaded regions represent the 95% confidence intervals. **e**, The distributions of SNAP-a donor scores (age adjusted and quantile normalized) for people with and without schizophrenia. $n$ = 93 controls, 87 cases. The $P$ value was calculated using a two-sided Wilcoxon rank-sum test. For the box plots, the box limits show the interquartile range, the whiskers show 1.5× the interquartile interval, the centre line shows the median value and the notches show the confidence intervals around the median values. **f–j**, Similar plots to those in **a–e**, respectively, but for the L5 IT glutamatergic neuron contribution to SNAP (SNAP-n). $n$ = 75,929 nuclei. Exc, excitatory neuron subtype. **k**, Variation in the expression levels across 180 individual persons (columns, ordered from left to right by SNAP expression levels) of a select set of strongly SNAP-recruited genes (rows) in astrocytes (left panel) and L5 IT glutamatergic neurons (right panel) of the 180 brain donors. One set of genes (SNAP-a; top) exhibits co-regulation in astrocytes; and a distinct set of genes (SNAP-n; bottom) exhibits co-regulation in neurons. Genes indicated by asterisks and hashes are at genomic loci associated with common and rare genetic variation in schizophrenia, respectively[22]. The grey bars indicate that regulon activity was not detected.

associated with SNAP-a (Supplementary Fig. 10) included eight genes with roles in adhesion of cells to synapses (*NRXN1*, *NTM*, *CTNND2*, *LSAMP*, *GPM6A*, *LRRC4C*, *LRRTM4* and *EPHB1*) (reviewed previously[11,12]). SNAP-a also appeared to strongly recruit genes encoding synaptic neurotransmitter reuptake transporters: *SLC1A2* and *SLC1A3* (encoding glutamate transporters EAAT1 and EAAT2) and *SLC6A1* and *SLC6A11* (encoding GABA transporters GAT1 and GAT3) were all among the 1% of genes most strongly associated with SNAP-a.

We sought to relate SNAP-a to an emerging appreciation of astrocyte heterogeneity and its basis in gene expression[13]. An earlier analysis of

astrocyte molecular and morphological diversity in mice identified gene-expression modules based on their co-expression relationships[14]. SNAP-a exhibited the strongest overlap ($P = 3.5 \times 10^{-4}$, $q = 0.015$, gene set enrichment analysis (GSEA)) (Supplementary Table 5) with the module that had correlated most closely with the size of the territory covered by astrocyte processes (the turquoise module in ref. 14, with overlap driven by genes including *EZR* and *NTM*). A potential interpretation is that SNAP-a supports these perisynaptic astrocytic processes[15].

Earlier studies identified reactive astrocyte states that are induced by strong experimental perturbations and injuries, and were described as polarized cell states[16]. We found that more than half of the human orthologues of markers for these states were expressed at levels that correlated negatively and in a continuous, graded manner with SNAP-a expression (Extended Data Fig. 11). At the single-astrocyte level, SNAP-a expression exhibited continuous, quantitative variation rather than discrete state shifts (Extended Data Fig. 10f,g), consistent with observations of abundant astrocyte biological variation less extreme than experimentally polarized states[17].

We performed an analogous cNMF analysis on the RNA-expression profiles of 75,929 glutamatergic neurons, focusing on a single, abundant subtype so that the variation among individual cells would be driven primarily by dynamic cellular programs rather than by subtype identity (Fig. 3f). One factor corresponded to the neuronal gene-expression effects of SNAP; we refer to this factor as SNAP-n (Fig. 3g–j and Supplementary Table 7). Like SNAP-a, the average expression of SNAP-n was associated with age and with schizophrenia (Fig. 3i,j). SNAP-n and SNAP-a were associated with each other still more strongly, even in a control-only age-adjusted analysis, highlighting the close coupling of neuronal and astrocyte gene expression (Extended Data Fig. 12). Although SNAP-n was associated with synaptic gene sets, the specific genes driving these enrichments were distinct from those driving SNAP-a (Fig. 3k, Supplementary Fig. 11 and Supplementary Table 8).

Expression of SNAP-a and SNAP-n was associated with the expression of many transcription factors and their predicted targets, and engaged distinct pathways in astrocytes and neurons (Fig. 3k and Extended Data Figs. 12c and 13b): for example, SREBP1 and its well-known transcriptional targets[18] in astrocytes, and JUNB (AP-1) and its well-known targets[19,20] in neurons (Extended Data Fig. 14) (the latter may reflect average neuronal activity levels in the PFC, which neuroimaging has found to decline (hypofrontality) in schizophrenia[21]). SNAP-a expression in astrocytes was also associated with a RORB regulon (underexpressed in SNAP[low] donors) and a KLF6 regulon (overexpressed in SNAP[low] donors) (Fig. 3k and Extended Data Fig. 13b); common genetic variation at *RORB* and *KLF6* is associated with schizophrenia[22].

## Schizophrenia genetics and SNAP

A key question when studying disease through human post-mortem tissue is whether observations involve disease-causing/disease-exacerbating processes, or reactions to disease circumstances such as medications. We found no relationship between SNAP expression and donor use of antipsychotic medications (Extended Data Fig. 4j,k), or between cholesterol-biosynthesis gene expression in astrocytes and donor statin intake (Extended Data Fig. 7b), but this does not exclude the possibility that astrocytes are primarily reacting to disease-associated synaptic hypofunction in neurons, as opposed to contributing to such hypofunction.

Human genetic data provide more powerful evidence, as inherited alleles affect risk or exacerbate disease processes rather than being caused by disease. We therefore sought to evaluate the extent to which SNAP-a and SNAP-n involved genes and alleles implicated by genetic studies of schizophrenia.

Previous research[22–24] found that genes expressed most strongly by neurons (relative to other cell types), but not genes expressed most

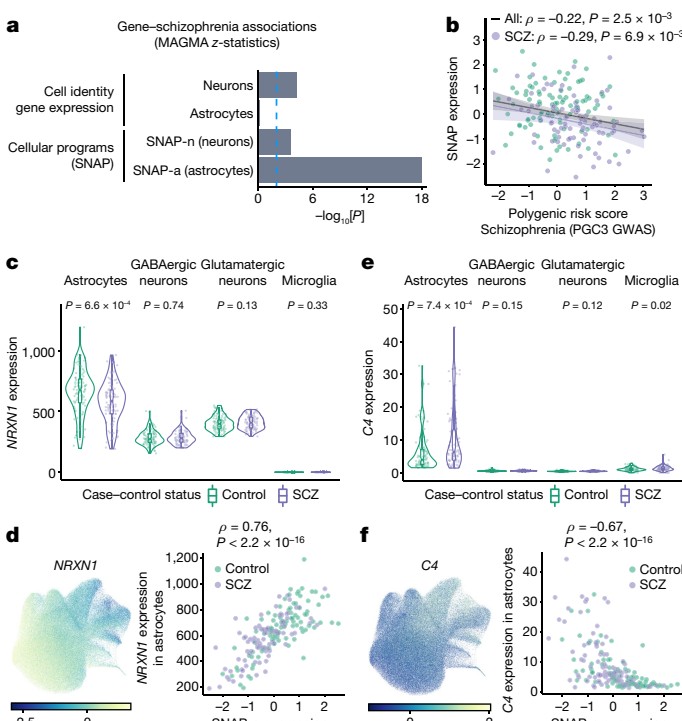

**Fig. 4 | The relationship between SNAP and schizophrenia genetics. a**, Enrichment of schizophrenia genetic association (from common variants, using MAGMA to generate a schizophrenia association *z* score for each gene) in the 2,000 genes most preferentially expressed in glutamatergic neurons and astrocytes (cell identity gene expression, upper bars), or the 2,000 genes of which the expression is most strongly recruited by SNAP-n and SNAP-a (cellular programs, lower bars). Values plotted are $-\log_{10}[P]$ from a joint regression analysis in which each gene set is an independent and competing predictive factor. See also the Supplementary Note. **b**, The relationship between donor SNAP expression (quantile normalized) and donor schizophrenia polygenic risk scores (Spearman's $\rho$; $n = 180$ donors; PGC3 GWAS from ref. 22). The shaded regions represent the 95% confidence intervals. **c**, *NRXN1* expression (per $10^5$ detected nuclear transcripts) in each cell type in individual donors. $n = 93$ controls, 87 cases. $P$ values were calculated using two-sided Wilcoxon rank-sum tests. For the box plots, the box limits show the interquartile range, the whiskers show 1.5× the interquartile interval, the centre line shows the median value and the notches show the confidence intervals around the median values **d**, *NRXN1* expression in individual astrocytes (using the same projection as in Fig. 3a–c) (left). The values represent Pearson residuals from variance stabilizing transformation. Right, the relationship between the 180 donors' *NRXN1* expression in astrocytes and SNAP-a expression (Spearman's $\rho$). **e,f**, Similar plots to those in **c** and **d**, but for *C4*.

strongly by glia, are enriched for the genes implicated by genetic analyses in schizophrenia[22–24]; we replicated these findings in our data (Fig. 4a and Supplementary Note). However, such analyses treat cell types as fixed levels of gene expression (cell identities), rather than as collections of dynamic transcriptional activities; SNAP-a involves a great many genes that are also strongly expressed in other cell types.

We found that the genes that are dynamically recruited by SNAP-a in astrocytes were enriched in genetic signals for schizophrenia: they were 14 times more likely than other protein-coding genes to reside at genomic loci implicated by common genetic variation in schizophrenia ($P = 5 \times 10^{-25}$, 95% confidence interval = 8.7–24, logistic regression) and 7 times more likely to have strong evidence from rare variants in schizophrenia (95% confidence interval = 2.3–21, $P = 5 \times 10^{-4}$, logistic regression) (Supplementary Note).

To evaluate whether common variation in the genes recruited by SNAP-a contributes more broadly to schizophrenia risk, beyond these

strongest associations, we used gene-level association statistics from the largest schizophrenia genome-wide association study to date[22,25]. As expected, the strongest neuron-identity genes (as defined in the earlier work) exhibited elevated schizophrenia association, whereas the strongest astrocyte-identity genes did not (Fig. 4a and Supplementary Note). However, in the same analysis, the genes most strongly associated with SNAP-a and SNAP-n were highly significant as additional predictive factors, particularly the genes associated with SNAP-a (Fig. 4a). Analysis by linkage disequilibrium (LD) score regression[26] also confirmed enrichment of schizophrenia risk factors among SNAP-a genes (Supplementary Fig. 12).

Polygenic risk involves thousands of common alleles across the genome, of which the effects converge on unknown biological processes. A polygenic risk score for schizophrenia was associated with reduced expression of SNAP but not with the other latent factors (Fig. 4b and Supplementary Fig. 13). Higher polygenic risk was also associated with a greater decrease in SNAP among people with schizophrenia (Fig. 4b).

To better understand such relationships, we examined the relationship between SNAP-a and genetic risk through two specific genes: neurexin-1 (*NRXN1*) and complement component 4 (*C4*).

Exonic deletions within *NRXN1* greatly increase the risk for schizophrenia[27,28]. Our data indicate that astrocytic, but not neuronal, *NRXN1* expression was reduced in people with schizophrenia and among people aged over 70 years (Fig. 4c and Extended Data Fig. 15a,b). Interindividual variation in astrocytic *NRXN1* expression was strongly associated with SNAP-a (Fig. 4d).

An increased copy number of the complement component 4 (*C4A*) gene more modestly increases the risk for schizophrenia[29]; however, far more interindividual variation in *C4* gene expression (>80%) arises from unknown, dynamic effects on *C4* expression[29,30]. We found that astrocytes, rather than neurons or microglia, are the main site of *C4* (including *C4A* and *C4B*) RNA expression in the human prefrontal cortex (Fig. 4e and Extended Data Fig. 15c). Donors with lower-than-average expression of SNAP-a tended to have greatly increased *C4* expression: such donors included 43 out of the 44 donors with the highest *C4* expression levels, and their astrocytes expressed 3.2-fold more *C4* compared with astrocytes of donors with above-average expression of SNAP-a (Fig. 4f). *C4* expression was also greatly increased among donors aged over 70 years (Extended Data Fig. 15d,e).

## Discussion

Here we identified SNAP—concerted gene-expression programs implemented by cortical neurons and astrocytes to corresponding degrees in the same individuals. SNAP expression varied even among unaffected control brain donors and may be a core axis of human neurobiological variation, with potential implications for cognition and plasticity that will be important to understand.

SNAP appears to involve many genes that contribute to synapses and to astrocyte–synapse interactions[31,32] (Figs. 2 and 3k, Supplementary Table 9 and Supplementary Figs. 10 and 11). The genes associated with SNAP-a suggested a potential role in supporting perisynaptic astrocyte processes, motile, morphologically plastic astrocyte projections whose interactions with synapses can promote synaptic stability[15]. Diverse lines of study increasingly reveal a key role for astrocytes in regulating the ability of synaptic networks to acquire and learn new information, for example, by lowering thresholds for activity and synaptic plasticity[33,34].

A notable aspect of SNAP involved the astrocytic regulation of genes with roles in fatty acid and cholesterol biosynthesis and cholesterol export, which strongly correlated (across donors) with expression of synaptic-component genes by neurons (Fig. 2d,e). Earlier research has defined a potential rationale for this neuron–astrocyte coordination: synapses and dendritic spines—synapse-containing morphological structures—require large amounts of cholesterol, which astrocytes supply[35]. Decreases in cholesterol biosynthesis have previously been noted in mouse models of brain disorders[36,37] that (like schizophrenia and ageing) involve cognitive losses, cortical thinning and reduction in neuropil.

Schizophrenia and ageing both brought substantial reductions in SNAP expression (Fig. 1i,j). Neuropsychological, neuroimaging and neuronal microstructural studies have long noted similar changes in schizophrenia and ageing[1,2,38–47]. Inherited genetic risk for schizophrenia is associated with decreased measures of cognition in older individuals[48,49], and schizophrenia greatly increases the risk of dementia later in life[50]. Our results suggest that these relationships between schizophrenia and ageing arise from shared cellular and molecular changes.

Underexpression of SNAP could, in principle, underlie longstanding microstructural observations[41–47] of reduced numbers of dendritic spines on cortical neurons in older humans and primates and in people with schizophrenia. These microstructural observations appear to arise from highly plastic thin spines and may therefore reflect reduced rates of continuous synapse formation and stabilization (rather than pruning of mature synapses)[42–47]. The gene-expression changes that we observed in the human dlPFC (Fig. 2c) suggest that cortical neurons of all types, including glutamatergic and GABAergic neurons, may be affected by such changes.

It is intriguing to consider whether pharmacotherapies or other interventions could be developed to promote SNAP as a way to address cognitive symptom domains in schizophrenia and ageing such as cognitive flexibility, working memory and executive function deficits, continuous and disabling features that are typically not improved by available treatments[1].

An important future direction will be to determine the extent to which SNAP is present in other brain areas, and the relationship of SNAP with molecular and physiological changes in dendrites, synapses and perisynaptic astrocyte processes. Additional questions involve the molecular mechanisms that accomplish neuron–astrocyte coordination and the extent to which SNAP supports learning and/or cognitive flexibility.

SNAP was made visible by human interindividual biological variation. Although controlled laboratory experiments usually try to eliminate genetic and environmental variation, natural variation may be able to reveal cell–cell coordination and regulatory programs in many tissues and biological contexts, offering new ways to identify pathophysiological processes within and beyond the human brain.

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

## Methods

### Ethical compliance

Brain donors were recruited by the Harvard Brain Tissue Resource Center/NIH NeuroBioBank (HBTRC/NBB), in a community-based manner, across the United States. Human brain tissue was obtained from the HBTRC/NBB. The HBTRC procedures for informed consent by the donor's legal next-of-kin and distribution of de-identified post-mortem tissue samples and demographic and clinical data for research purposes are approved by the Mass General Brigham Institutional Review Board. Post-mortem tissue collection followed the provisions of the United States Uniform Anatomical Gift Act of 2006 described in the California Health and Safety Code section 7150 and other applicable state and federal laws and regulations. Federal regulation 45 CFR 46 and the associated guidance indicate that the generation of data from de-identified post-mortem specimens does not constitute human participant research that requires institutional review board review.

### Donors for snRNA-seq

Donor information with anonymized donor IDs is available in Supplementary Table 1. Consensus diagnosis of schizophrenia was performed by retrospective review of medical records and extensive questionnaires concerning social and medical history provided by family members. Several regions from each brain were examined by a neuropathologist. We excluded participants with evidence for gross and/or macroscopic brain changes, or with clinical history consistent with cerebrovascular accident or other neurological disorders. Participants with Braak stage III or higher (modified Bielchowsky stain) were excluded. None of the participants had substantial reported history of substance dependence within 10 or more years from death, as further corroborated by negative toxicology reports. The absence of recent substance abuse is typical for samples from the HBTRC, which receives exclusively community-based tissue donations.

Exposure to psychotropic and neurotropic medications was assessed on the basis of medical records. Estimated daily milligram doses of antipsychotic drugs were converted to the approximate equivalent of chlorpromazine as a standard comparator[51]. These values are reported as lifetime, as well as last six months of life, grams per patient. Exposure to other classes of psychotropic drugs was reported as present or absent.

### Single-nucleus library preparation and sequencing

We analysed the dlPFC (Brodmann area 46 (BA46)), which exhibits functional and microstructural abnormalities in schizophrenia[52,53] and in ageing[46]. Frozen tissue blocks containing BA46 were obtained from the HBTRC. We used snRNA-seq rather than single-cell RNA-seq to avoid effects of cell morphology on ascertainment, and because nuclear (but not plasma) membranes remain intact in frozen post-mortem tissue. Nuclear suspensions from frozen tissue were generated according to a protocol that we have made available at Protocols.io (https://doi.org/10.17504/protocols.io.4r3l22e3xl1y/v1). To ensure that batch compositions were balanced, researchers were not blinded to the batch allocation or processing order of each specimen. To maximize the technical uniformity of the snRNA-seq data, we processed sets of 20 brain specimens (each consisting of affected and control donors) at once as a single pooled sample. Specimens were allocated into batches of 20 specimens per batch, ensuring that the same number of cases and age-matched controls (10 per group), and men and women (10 per group) were included in each batch. Some donors were resampled across multiple batches to enable quality-control analyses (Extended Data Fig. 2). Specimens from cases and age-matched controls were also processed in alternating order within each batch. Researchers had access to unique numerical codes assigned to the donor-of-origin of each specimen as well as basic donor metadata (for example, case–control status, age, sex).

From each donor, 50 mg of tissue was dissected from the dlPFC—sampling across the cortical layers and avoiding visible concentrations of white matter—and used to extract nuclei for analysis. Generation of gel beads-in-emulsion and library preparation was performed according to the 10x Chromium Single Nuclei 3′ v3.1 protocol (version CG000204_ChromiumNextGEMSingleCell3'v3.1_Rev D). We encapsulated nuclei into droplets using approximately 16,500 nuclei per reaction, understanding that about 95% of all doublets (cases in which two nuclei were encapsulated in the same droplet) would consist of nuclei from distinct donors and therefore be recognized by the Dropulation analysis[7] as containing combinations of SNP alleles from distinct donors. cDNA amplification was performed using 13 PCR cycles.

Raw sequencing reads were aligned to the hg38 reference genome using the standard Drop-seq (v.2.4.1)[54] workflow, modified so that reads from *C4* transcripts would not be discarded as multi-mapping (see the 'MetaGene discovery' section below). Reads were assigned to annotated genes if they mapped to exons or introns of those genes. Ambient/background RNA was removed from digital gene expression (DGE) matrices using CellBender (v.0.1.0)[55] remove-background.

### Genotyping and donor assignment from snRNA-seq data

We used combinations of hundreds of transcribed SNPs to assign each nucleus to its donor of origin using Dropulation (v.2.4.1)[7]. Previous Dropulation analyses of stem cell experiments used whole-genome sequencing (WGS) data on the individual donors for such analyses[7]. For this study, we developed a cost-efficient approach based on SNP array data with imputation. Genomic DNA from the individual brain donors was genotyped by SNP array (Illumina GSA).

Raw Illumina IDAT files from the GSAMD-24v1-0_20011747 array (2,085 samples) and GSAMD-24v3-0-EA_20034606 array (456 samples) were genotyped using GenCall (v.3.0.0)[56] and genotypes were phased using SHAPEIT4 (v.4.2.2)[57] by processing the data through the MoChA workflow (v.2022-12-21)[58,59] (https://github.com/freeseek/mochawdl) using the default settings and aligning markers against the GRCh38 genome. *APOE* genotypes for marker rs429358 were removed due to unreliable genotypes. To improve phasing, genotypes from the McLean cohort were combined with genotypes from the Genomic Psychiatry Cohort with IDAT files available also from the GSAMD-24v1-0_20011747 array (5,689 samples)[60]. After removing 128 samples recognized as duplicates, phased genotypes were then imputed using IMPUTE5 (v.1.1.5)[61] by processing the output data from the MoChA workflow using the MoChA imputation workflow and using the high-coverage 1000 Genomes reference panel for GRCh38[62], including 73,452,470 non-singleton variants across all the autosomes and chromosome X. Only SNPs with imputation quality INFO > 0.95 were used for donor assignments. Using this approach, we found that 99.6% of nuclei could be assigned confidently to a donor (Extended Data Fig. 2a).

To evaluate the accuracy of this method of donor assignment, we genotyped a pilot cohort of 11 donors using both WGS and SNP array. Importantly, the two methods had 100% concordance on the assignment of individual nuclei to donors, validating both our computational donor-assignment method and the sufficiency of the SNPs-plus-imputation approach (Extended Data Fig. 2c). SNP data for the individual donors are available at NeMO (https://assets.nemo-archive.org/dat-bmx7s1t).

After donor assignment, DGE matrices from all libraries in each batch (7 to 8 libraries per batch) were merged for downstream analyses.

### Cell-type assignments

All classification models for cell assignments were trained using scPred (v.1.9.2)[63]. DGE matrices were processed using the following R and python packages: Seurat (v.3.2.2)[64], SeuratDisk (v.0.0.0.9010)[65], anndata (v.0.8.0)[66], numpy (v.1.17.5)[67], pandas (v.1.0.5)[68,69] and Scanpy (v.1.9.1)[70].

**Cell types. Model training.** The classification model used for cell-type assignments was trained on the DGE matrix from batch 6 (BA46_2019-10-16), which was annotated as follows. Nuclei with fewer than 400 detected genes and 100 detected transcripts were removed from the DGE matrix from this batch. After normalization and variable gene selection, the DGE matrix was processed through an initial clustering analysis using independent component analysis (ICA, using fastICA (v.1.2-1))[71] as previously described[72]. This analysis produced clustering solutions with 43 clusters of seven major cell types (astrocytes, endothelial cells, GABAergic neurons, glutamatergic neurons, microglia, oligodendrocytes and polydendrocytes) that could be identified based on expression of canonical marker genes (markers in Supplementary Fig. 1) (note that around 9% of cells within clusters annotated as endothelial cells do not express canonical endothelial cell markers but, rather, those of pericytes; these ~1,400 cells have been grouped together with endothelial cells for downstream analyses). scPred was trained on this annotated DGE matrix, and the resulting model was subsequently used to make cell-type assignments for the remaining batches' DGE matrices.

**Filtering.** After an initial cell-type classification using the above model, the DGE matrices were filtered further to remove any remaining heterotypic doublets missed by scPred. First, raw DGE matrices from each of the 11 batches were subsetted to form separate DGE matrices for each of the 7 major cell types (77 subsetted DGE matrices total). Each subsetted DGE matrix was normalized using sctransform (v.0.3.1)[64] with 7,000 variable features, scaling and centring. For each cell type, normalized DGE matrices from the 11 batches were merged and clustered together in Scanpy (v.1.9.1)[70] using 50 principal components, batch correction by donor using BBKNN (v.1.5.1)[73] and Leiden clustering using a range of resolutions. The most stable clustering resolution for each cell type was selected using clustree (v.0.4.4)[74]. Clusters expressing markers of more than one cell type were determined to be heterotypic doublets; cell barcodes in these clusters were discarded from the above DGE matrices, and these filtered DGE matrices were then carried forward for integrated analyses across batches.

**Neuronal subtypes.** Classification models for neuronal subtypes were trained using DGE matrices from a previous study[75] that were subsetted to glutamatergic or GABAergic neuron nuclei in middle temporal gyrus (MTG). Although a similar dataset exists for human brain nuclei from the primary motor cortex (M1)[76], we trained the model only on the MTG dataset as the M1 lacks a traditional layer 4 (L4), whereas BA46 does have a L4.

The neuronal subtypes in this dataset include glutamatergic neuron subtypes of distinct cortical layers and with predicted intratelencephalic (IT), extratelencephalic (ET), corticothalamic (CT) and near-projecting (NP) projection patterns, as well as the four cardinal GABAergic neuron subtypes arising from the caudal (CGE: *LAMP5*+, *VIP*+) and medial (MGE: *PVALB*+, *SST*+) ganglionic eminences.

We made the following adjustments to the MTG annotations before model training. First, as subtype-level annotations (for example, L5 IT, as used previously[76] for M1) were not available for the MTG dataset, we inferred these based on M1/MTG cluster correspondences (from extended data figure 10 in ref. 76). Second, we reassigned the following glutamatergic neuron types in the MTG from the L4 IT subtype (as inferred by integration with M1 in ref. 76) to the L2/3 IT subtype: Exc L3–5 RORB FILIP1L, Exc L3–5 RORB TWIST2 and Exc L3–5 RORB COL22A1. This was done on the basis of their properties described in other studies—for example, the Exc L3–5 RORB COL22A1 type has been described as a deep L3 type by Patch-seq[77]—and by the expression of their marker genes on a two-dimensional projection of the RNA-expression profiles of glutamatergic neuron nuclei (Supplementary Fig. 2).

Feature plots for neuronal subtypes (Supplementary Figs. 2 and 3) were generated using markers from the repository in https://bioportal.

bioontology.org/ontologies/PCL (v1.0, 2020-04-26)[75,76,78], specifically those for neuronal subtypes from MTG.

**Astrocyte subtypes.** Normalized, filtered DGE matrices from the 11 batches were merged and clustered together in scanpy using 8 principal components, batch correction by donor using bbknn[73] and Leiden clustering using a range of resolutions. The most stable resolution that created distinct clusters for putative astrocyte subtypes (resolution 1.3) was selected using clustree[74]. Feature plots for astrocyte subtypes previously described in both the MTG and M1[75,76] (Extended Data Fig. 9) were generated using markers from the repository at https://bioportal. bioontology.org/ontologies/PCL (v.1.0, 2020-04-26)[75,76,78]. Leiden clusters were assigned to one of three astrocyte subtypes on the basis of expression of these subtype markers.

### Donor exclusion
Donors were excluded on the basis of unusual gene-expression profiles and/or cell-type proportions (potentially related to agonal events) as outlined below.

**Expression.** Donors with fewer than 1,000 total UMIs in any cell type were first excluded. Next, for each cell type, gene-by-donor expression matrices comprising the remaining donors were scaled to 100,000 UMIs per donor and filtered to the top expressing genes (defined as having at least 10 UMIs per 100,000 for at least one donor; these were among the top 12–19% of expressed genes). These filtered expression matrices by cell type were merged into a single expression matrix that was used to calculate each donor's pairwise similarity to the other donors (Pearson correlations of $\log_{10}$-scaled expression values across genes). The median of these pairwise correlation values was determined to be the conformity score for each donor. To identify outliers, these donor conformity scores were converted to modified $z$ scores ($M_i$) for each donor as described previuously[79]:

$$M_i = 0.6745 \times (x_i - \tilde{x})/\text{MAD}$$

where $x_i$ is the donor's conformity score, $\tilde{x}$ is the median of donor conformity scores and MAD is the median absolute deviation of donor conformity scores.

Donors whose modified $z$ scores had absolute values of >5 were excluded. This approach flagged a total of five donors (one who had low UMI counts and four who were outliers on the basis of expression).

**Cell-type proportions.** Each donor's pairwise similarity to the other donors was determined on the basis of cell-type proportions (that is, the values plotted in Supplementary Fig. 1c,d). Donor conformity scores and modified $z$ scores based on these values were calculated for each donor using the same approach described above for expression values. Donors whose modified $z$ scores had absolute values of >15 were excluded. This approach flagged a total of nine donors, two of whom were also flagged as expression outliers.

Between the two approaches, in total, 11 unique donors were flagged as outliers (4 control, 7 schizophrenia) and excluded from downstream analyses.

### Latent factor analysis
**snRNA-seq data.** Our approach was to (1) create a gene-by-donor matrix of expression measurements for each of seven cell types; (2) concatenate these matrices into a larger matrix in which each gene is represented multiple times (once per cell type); and (3) perform latent factor analysis[8,80] on this larger matrix. We selected probabilistic estimation of expression residuals (PEER)[81] over other approaches (such as principal component analysis (PCA)) for inferring latent variables as it is more sensitive and less dependent on the number of factors modelled. A major pitfall to avoid when performing latent

factor analysis is obtaining highly correlated factors due to overfitting. The latent factors that we have inferred are independent from each other when we compare their gene loadings (Extended Data Fig. 3c), enabling us to proceed with downstream analyses based on these factors.

Raw, filtered DGE matrices from each of the 11 batches were subsetted to form separate DGE matrices for each of the 7 major cell types (77 subsetted DGE matrices total). For each subsetted DGE matrix, cell barcodes from outlier donors were excluded, the DGE matrix was normalized using sctransform (v.0.3.1)[64] with 3,000 variable features, and the output of Pearson residual expression values (with all input genes returned) was exported to a new DGE matrix. For each cell type, these new expression values in the 11 normalized DGE matrices were summarized across donors (taking the sum of residual expression values) to create a gene-by-donor expression matrix. Each of these expression matrices was filtered to the top 50% of expressed genes (based on feature counts scaled to 100,000 transcripts per donor), yielding expression matrices with approximately 16,000 to 18,000 genes per cell type. Within each expression matrix, each gene name was modified with a suffix to indicate the cell type of origin (for example, ACAP3 to ACAP3_astrocyte), and the seven expression matrices were combined to produce a single expression matrix with expression values from all seven cell types for each donor (a schematic is shown in Fig. 1f). This expression matrix was used as the input to latent factor analysis with PEER (v.1.0)[81] using the default parameters and a range of requested factors $k$.

Although we looked for correlations between these factors and technical variables, these analyses were negative, with one exception: latent factor 2 (LF2) appeared to capture quantitative variation in the relative representation of deep and superficial cortical layers in each dissection (Extended Data Fig. 3f).

Latent factor donor expression values were adjusted for age by taking the residuals from a regression of the donor expression values against age.

To improve the visualization of latent factor donor expression values while leaving the results of statistical analyses unchanged, quantile-normalized values were calculated in R using the function qnorm(rank($x$)/(length($x$) + 1)). The figure legends indicate when these quantile-normalized values are used.

**Proteomics data.** Protein intensities from the *LRRK2* Cohort Consortium (LCC) cohort of a previous study[82] were downloaded from the ProteomeXchange Consortium (PXD026491) and subset to those peptides that passed the $q$-value threshold in at least 25% of all analysed samples. These were further subset to intensities from control donors without the LRRK2(G2019S) mutation and without erythrocyte contamination ($n$ = 22 donors). After normalization of the protein intensities using sctransform (v.0.3.1)[64], the output of Pearson residual expression values (with all input proteins returned) was exported to a new matrix. This matrix of normalized protein intensities was used as the input to latent factor analysis with PEER (v.1.0)[81] using the default parameters.

For comparisons of CSF protein loadings to SNAP gene loadings in Supplementary Fig. 7, each gene in SNAP was represented by a single composite loading representing gene loadings from all cell types. This composite loading was determined for each gene by first calculating the median expression of each gene (in each cell type), then calculating a new loading onto SNAP weighted across cell types by these median expression values.

## Rhythmicity analysis

For Extended Data Fig. 4f, rhythmicity analyses were performed as described previously[83] using scripts available at GitHub (https://github.com/KellyCahill/Circadian-Analysis-) and donor time of death in zeitgeber time. Analyses also used the following packages: lme4 (v.1.1-31)[84], minpack.lm (v.1.2-4)[85].

## GSEA

For GSEA[9,86] of latent factors inferred by PEER, the C5 Gene Ontology collection (v.7.2)[87,88] from the Molecular Signatures Database[89,90] was merged with the SynGO (release 20210225)[91] biological process (BP) and cell component (CC) gene lists. Gene sets from this merged database that were enriched in each latent factor were identified with GSEAPreranked in GSEA (v.4.0.3)[9,86] using 10,000 permutations and gene loadings as the ranking metric.

For astrocyte latent factors inferred by cNMF[10], GSEA was performed as described above with the addition of the following custom gene sets to the database:
- PGC3_SCZ_GWAS_GENES_1TO2_AND_SCHEMA1_GENES: a gene set comprising genes implicated in human-genetic studies of schizophrenia, including genes at 1–2 gene loci from GWAS (PGC3)[22] and genes with rare coding variants (FDR < 0.05)[23].
- Gene sets for each of the seven astrocyte subclusters identified in ref. 14.
- Gene sets for each of the 62 colour module eigengenes identified by WGCNA in ref. 14.
- Gene sets for each of the six astrocyte subcompartments analysed in ref. 92, comprising genes encoding the proteins that were unique to or enriched in these subcompartments.

For L5 IT glutamatergic neuron latent factors inferred by cNMF, GSEA was performed as described above with the addition of the following custom gene sets to the database:
- PGC3_SCZ_GWAS_GENES_1TO2_AND_SCHEMA1_GENES: a gene set comprising genes implicated in human genetic studies of schizophrenia, including genes at 1–2 gene loci from GWAS (PGC3 (ref. 22)) and genes with rare coding variants (FDR < 0.05)[23].

## Selected gene sets

On the basis of the results of the GSEA described above, we selected several of the top-enriched gene sets for further analyses. These are referred to in the figures with labels modified for brevity, but are described in further detail below. Lists of genes in each gene set are provided in Supplementary Table 9.
- Integral component of postsynaptic density membrane (Extended Data Figs. 6 and 8 and Supplementary Fig. 8): core genes contributing to the enrichment of GO:0099061 (v.7.2, integral component of postsynaptic density membrane) in the glutamatergic neuron component of LF4 (SNAP).
- Neurotransmitter reuptake transporters (Fig. 2e, Extended Data Figs. 6 and 8 and Supplementary Fig. 8): genes from among the 100 genes most strongly recruited by cNMF2 (SNAP-a) with known functions as neurotransmitter-reuptake transporters. These include core genes contributing to the enrichment of GO:0140161 (v.7.2, monocarboxylate: sodium symporter activity) in SNAP-a.
- Presynapse (Extended Data Figs. 6 and 8 and Supplementary Fig. 8): core genes contributing to the enrichment of GO:0098793 (v.7.2, presynapse) in the GABAergic neuron component of LF4 (SNAP).
- Regulation of cholesterol biosynthesis (Fig. 2d,e, Extended Data Figs. 6–8 and 13d and Supplementary Fig. 8): core genes contributing to the enrichment of GO:0045540 (v.7.2, regulation of cholesterol biosynthetic process) in the astrocyte component of LF4 (SNAP). This enrichment is of interest as cholesterol is an astrocyte-supplied component of synaptic membranes[35,93,94]. Products of this biosynthetic pathway also include other lipids and cholesterol metabolites with roles at synapses, including 24S-hydroxycholesterol, a positive allosteric modulator of NMDA receptors[95]. Although we refer to this gene set by this label based on its annotation by GO, we note that subsets of these genes contribute to cholesterol export and/or to synthesis of additional fatty acids.

- Schizophrenia genetics (Fig. 3k and Extended Data Fig. 13a): prioritized genes from ref. 23 (FDR < 0.05) or ref. 22.
- Synapse organization (Fig. 3k): core genes contributing to the enrichment of GO:0050808 (v.7.2, synapse organization) in cNMF6 (SNAP-n).
- Synaptic cell adhesion (Figs. 2e and 3k, Extended Data Figs. 6, 8 and 13a and Supplementary Fig. 8): genes from among the 20 genes most strongly recruited by cNMF2 (SNAP-a) with known functions in synaptic cell adhesion. This biological process was selected due to the enrichment of GO:0099560 (v.7.2, synaptic membrane adhesion) in SNAP-a.
- Synaptic receptors and transporters (Fig. 3k and Extended Data Fig. 13a,c): genes from among the 100 genes most strongly recruited by cNMF2 (SNAP-a) with known functions as synaptic receptors and transporters.
- Synaptic vesicle (Fig. 3k): core genes contributing to the enrichment of GO:0008024 (v.7.2, synaptic vesicle) in cNMF6 (SNAP-n).
- Synaptic vesicle cycle (Fig. 2c and Extended Data Fig. 5): core genes contributing to the enrichment of GO:0099504 (v.7.2, synaptic vesicle cycle) in the glutamatergic and GABAergic neuron components of LF4 (SNAP).
- Trans-synaptic signalling (Fig. 2e and Extended Data Figs. 6 and 8): core genes contributing to the enrichment of GO:0099537 (v.7.2, trans-synaptic signalling) in the glutamatergic neuron component of LF4 (SNAP).

Gene sets displayed in Fig. 2b are the SynGO terms most strongly enriched in each top-level category (among biological processes: process in the presynapse, synaptic signalling, synapse organization, process in the postsynapse, transport and metabolism, respectively).

### Analysis of astrocyte and glutamatergic L5 IT neuron gene-expression programs

**Consensus non-negative matrix factorization.** cNMF (v.1.2)[10] was performed on both astrocyte and glutamatergic L5 IT neurons. We used cNMF owing to its scalability to the astrocyte and glutamatergic L5 IT neuron datasets. The cNMF protocol detailed in the tutorial for PBMCs at GitHub (https://github.com/dylkot/cNMF/blob/master/Tutorials/analyze_pbmc_example_data.ipynb) was followed for the initial data filtering and analysis. For both datasets, data were filtered to remove cells with fewer than 200 genes or 200 UMIs. Genes expressed in fewer than 10 cells were removed. Factorization was run on raw counts data after filtering, with iterations of factorization run for each $k$ (factors requested), with a $k$ ranging from 3 to 30.

The astrocyte raw counts data contained 179,764 cells and 42,651 genes, of which 0 cells and 9,040 genes were excluded. On the basis of PCA of the gene expression matrix and the cNMF stability report, factorization with $k = 11$ was selected for further analysis. The 11 cNMF factors together explained 25% of variation in gene expression levels among single astrocytes.

The L5 IT raw counts data contained 75,929 cells and 42,651 genes, of which 0 cells and 8,178 genes were excluded. On the basis of the PCA of the gene expression matrix and the cNMF stability report, factorization with $k = 13$ was selected for further analysis. The 13 cNMF factors together explained 44% of variation in gene expression levels among single L5 IT glutamatergic neurons. To align the direction of interpretation across all three analyses (SNAP, SNAP-a, and SNAP-n), we took the negative of cNMF factor 6 (SNAP-n) cell scores, gene loadings and donor scores.

The latent factor usage matrix (cell by factor) was normalized before analysis to scale each cell's total usage across all factors to 1.

**Co-varying neighbourhood analysis.** To further assess the robustness of the astrocyte gene-expression changes represented by SNAP and SNAP-a, we used a third computational approach—co-varying neighbourhood analysis (CNA, v.0.1.4)[96]. The protocol provided in the CNA tutorial at GitHub (https://nbviewer.org/github/yakirr/cna/blob/master/demo/demo.ipynb) was followed for data preprocessing and analysis.

Pilot association tests to find transcriptional neighbourhoods associated with schizophrenia case–control status were first performed using the default value for $N_{null}$. These pilot analyses evaluated the effects of batch correction (by batch or donor) and covariate correction (by age, sex, post-mortem interval, number of UMIs or number of expressed genes). Nearly all analyses yielded highly similar neighbourhoods associated with case–control status with the same global $P$ value ($P = 1 \times 10^{-4}$), with the exception of batch correction by donor which yielded $P = 1$. The final association test described in Supplementary Fig. 9 was performed with an increased value for $N_{null}$ ($N_{null} = 1,000,000$) and without additional batch or covariate correction.

### Regulatory network inference

The goal of pySCENIC[97,98] is to infer transcription factors and regulatory networks from single-cell gene-expression data. The pySCENIC (v0.11.2) protocol detailed in the tutorial for PBMCs at GitHub (https://github.com/aertslab/SCENICprotocol/blob/master/notebooks/PBMC10k_SCENIC-protocol-CLI.ipynb) was followed for the initial data filtering and analysis. For both astrocytes and L5 IT glutamatergic neurons, data were filtered to remove cells with fewer than 200 genes, and genes with fewer than 3 cells. Cells with high MT expression (>15% of their total transcripts) were removed.

The gene regulatory network discovery adjacency matrix was inferred by running Arboreto on the gene counts matrix and a list of all transcription factors provided by the authors (https://resources.aertslab.org/cistarget/tf_lists/allTFs_hg38.txt) to generate an initial set of regulons. This set was further refined using ctx, which removes targets that are not enriched for a motif in the transcription factor using a provided set of human specific motifs (https://resources.aertslab.org/cistarget/motif2tf/motifs-v9-nr.hgnc-m0.001-o0.0.tbl) and cis targets (https://resources.aertslab.org/cistarget/databases/homo_sapiens/hg38/refseq_r80/mc9nr/gene_based). Finally, aucell was run to generate the per-cell enrichment scores for each discovered transcription factor.

### Super-enhancer analysis

Preparation of input BAM files was performed as follows. FASTQ files of bulk H3K27ac HiChIP data from the middle frontal gyrus[99] were downloaded from the Gene Expression Omnibus (GEO: GSM4441830 and GSM4441833). Demultiplexed FASTQ files were trimmed with Trimmomatic (v.0.33)[100] using the parameter SLIDINGWINDOW:5:30. Trimmed reads were aligned to the hg38 reference genome with Bowtie2 (v2.2.4)[101] using the default parameters. Uniquely mapped reads were extracted with samtools (v.1.3.1)[102] view using the parameters -h -b -F 3844 -q 10.

Preparation of input constituent enhancers was performed as follows. FitHiChIP interaction files for H3K27ac from the middle frontal gyrus[99] were downloaded from the GEO (GSM4441830 and GSM4441833). These were filtered to interacting bins (at interactions with $q < 0.01$) that overlap bulk H3K27ac peaks in the one-dimensional HiChIP data in both replicates. Next, these bins were intersected with IDR-filtered single-cell assay for transposase-accessible chromatin using sequencing (scATAC–seq) peaks in isocortical and unclassified astrocytes (peaks from clusters 13, 15 and 17, downloaded from the GEO (GSE147672))[99]. Unique coordinates of these filtered regions were converted to GFF files.

Super-enhancers were called with ROSE (v.1.3.1)[103,104] using the input files prepared above and the parameters -s 12500 -t 2500. Coordinates of promoter elements for *Homo sapiens* (December 2013 GRCh38/hg38) were downloaded from the Eukaryotic Promoter

Database (EPD)[105] using the EPDnew selection tool (https://epd.expasy.org/epd/EPDnew_select.php)[106]. Using these sets of coordinates, FitHiChIP loops that overlap bulk H3K27ac peaks and scATAC peaks in astrocytes were subset to those that contained a promoter in one anchor and a super-enhancer in the other anchor. Binomial smooth plots were generated as described previously[107].

## Heritability analyses

**MAGMA.** Summary statistics from ref. 22 were uploaded to the FUMA (v.1.5.6)[108] web server (https://fuma.ctglab.nl). Gene-level z scores were calculated using SNP2GENE with the 'Perform MAGMA' function (MAGMA v.1.08) and the default parameter settings. The reference panel population was set to '1000G Phase3 EUR'. The MHC region was excluded due to its unusual genetic architecture and LD. MAGMA z scores were then used for downstream analyses as described in the Supplementary Note.

**Stratified LD score regression.** To partition SNP heritability, we used stratified LD score regression (S-LDSC; v.1.0.1)[26], which assesses the contribution of gene expression programs to disease heritability. First, for analysis of astrocyte-identity genes, we computed (within the BA46 region only), a Wilcoxon rank-sum test on a per-gene basis using presto (v.1.0.0)[109] between astrocytes and all other cell types; for analysis of astrocyte-activity genes (SNAP-a), we sorted all genes expressed in astrocytes by their SNAP-a loadings and took the top 2,000 genes. We then converted each gene set into annotations for S-LDSC by extending the window size to 100 kb (from the transcription start site and transcription end site), and ordered SNPs in the same order as the .bim file (from phase 3 of the 1000 Genomes Project[110]) used to calculate the LD scores. We then computed LD scores for annotations using a 1 cM window and restricted the analysis to Hapmap3 SNPs. We excluded the MHC region due to both its high LD and high gene density. We used LD weights calculated for HapMap3 SNPs for the regression weights. We then jointly modeled the annotations corresponding to our gene expression program, as well as all protein-coding genes, and the baseline model (baseline model v.1.2). We tested for enrichment of SNP heritability on the traits listed below. The LDSC script 'munge_sumstats.py' was used to prepare the summary statistics files. We used the resultant P values, which reflect a one-sided test that the coefficient ($\tau$) is greater than zero, as a determinant as to whether our cell type gene expression programs are enriched for SNP-heritability of a given trait[111].

We used summary statistics from the following studies in Supplementary Fig. 12: ADHD[112], ALS[113], Alzheimer's disease[114], age of smoking initiation[115], autism[116], bipolar disorder (all, type I, and type II)[117], cigarettes per day[115], educational attainment[118], epilepsy (all, focal, generalized)[119], height[120], IQ[121], insomnia[122], neuroticism[123], OCD[124], schizophrenia[22], PTSD[125], risk[126], subjective well-being[127], smoking cessation[115], smoking initiation[115], Tourette's[128] and ulcerative colitis[129].

## Polygenic risk scores

Clumped summary statistics for schizophrenia (from ref. 22) across 99,194 autosomal markers were downloaded from the Psychiatric Genomics Consortium portal (file PGC3_SCZ_wave3_public.clumped.v2.tsv). After liftOver of markers to GRCh38 using custom tools, 99,135 markers were available for scoring. We processed the output data from the MoChA imputation workflow[58,59] using BCFtools (v.1.16) and the MoChA score (v.2022-12-21)[58,59] workflow (https://github.com/freeseek/score) to compute schizophrenia polygenic scores across all 2,413 imputed samples from the McLean cohort.

## C4

**MetaGene discovery.** Genes that have high sequence homology are typically difficult to capture using standard UMI counting methods. Reads from these regions map to multiple locations in the genome with low mapping quality, and are ignored by many gene expression algorithms. MetaGene discovery leverages that high sequence similarity by looking for UMIs that consistently map to multiple genes at low mapping quality consistently across many cells.

Each UMI is associated with a single gene if at least one read from the UMI uniquely maps to a single gene model. If all reads are mapped at low quality to multiple genes, then assignment of that UMI to a specific gene model is ambiguous, and that UMI is associated with all gene models. By surveying a large number of cells, a set of gene families are discovered where UMIs are consistently associated with sets of genes. This discovery process finds expected sets of gene families with high sequence homology directly from the mapping, such as *C4A/C4B*, *CSAG2/CSAG3* and *SERF1A/SERF1B*.

These UMIs are then extracted in the counts matrix as a joint expression of all genes in each set. We prefer to calculate expression as the joint expression of all genes in the set because the priors in the data prevent confidently distributing these ambiguous UMIs. For example, *C4A* and *C4B* have very few UMIs that map uniquely to either gene in the set (8 UMIs, <0.5% of all UMIs captured for this set of genes), which is a weak prior to proportionally assign ambiguous UMIs to the correct model.

This approach was validated for *C4* expression by generating a reference genome that contained only one copy of *C4*. This allowed each UMI to map uniquely to the single remaining copy of the gene using standard tools. The custom reference approach and joint expression of *C4A/C4B* on the basis of the metagene approach was concordant in 15,664 of 15,669 cells tested (Extended Data Fig. 15c).

**Imputation of *C4* structural variation.** Phased copy-number calls for structural features of the *C4* gene family were obtained by imputation using Osprey, a method for imputing structural variation. The total copy number of *C4* genes, the number of copies of *C4A* and *C4B*, and the copy number of the polymorphic HERV element that distinguishes long from short forms of *C4*[29] were imputed into the McLean cohort using a reference panel based on 1000 Genomes[62].

An imputation reference panel was constructed for GRCh38 using 2,604 unrelated individuals (out of 3,202 total) from 1000 Genomes. SNPs were included in the reference panel if (1) they were within the locus chromosome 6: 24000000–34000000 but excluding the copy-number variable region chromosome 6: 31980001–32046200; and (2) they were not multi-allelic and (3) they had an allele count (AC) of at least 3 when subset to the 2,604 reference individuals.

The imputation reference panel was merged with genotypes for the McLean cohort obtained from the GSA genotyping arrays. Markers not appearing in both datasets were dropped and the merged panel was phased with SHAPEIT4 (v.4.2.0)[57] using the default parameters plus --sequencing and the default GRCh38 genetic map supplied with SHAPEIT.

Reference copy numbers for the *C4* structural features on GRCh38 were obtained for the 3,202 1000 Genomes samples using a custom pipeline based on Genome STRiP (v.2.0)[130]. The source code for this pipeline is available at Terra (http://app.terra.bio)[131]. In brief, the pipeline uses Genome STRiP to estimate the total *C4* copy number and HERV copy number from normalized read depth of coverage, then estimates the number of copies of *C4A* and *C4B* using maximum likelihood based on reads that overlap the *C4* active site (coordinates, chromosome 6: 31996082–31996099 and chromosome 6: 32028820–32028837). These copy-number genotypes were then subset to the 2,604 unrelated individuals.

The structural features were imputed into the merged imputation panel using Osprey (v.0.1-9)[132,133] by running ospreyIBS followed by osprey using the default parameters plus '-iter 100', the SHAPEIT4 genetic map for GRCh38 chromosome 6 and a target genome interval of chromosome 6: 31980500–32046500.

The output from Osprey was post-processed using a custom R script (refine_C4_haplotypes.R) that enforces constraints between the copy-number features and recalibrates the likelihoods considering only

possible haplotypes. The enforced constraints are that the *C4A* + *C4B* copies must equal the total *C4* copy number and that the HERV copy number must be less than or equal to *C4* copy number.

## Source data and visualization

In addition to the software cited above, we used Colour Oracle (v.1.3)[134,135] as well as the following packages to prepare the source data and figures in this manuscript.

Python (v.3.8.3): matplotlib (v.3.5.2)[136] and seaborn (v.0.10.1)[137]. R (v.4.1.3): cluster (v.2.1.2)[138], ComplexHeatmap (v.2.10.0)[139,140], data.table (v.1.14.8)[141], DescTools (v.0.99.48)[142], dplyr (v.1.1.2)[143], gdata (v.2.19.0)[144], ggforce (v.0.4.1)[145], ggplot2 (v.3.4.2)[146], ggpmisc (v.0.5.3)[147], ggpointdensity (v.0.1.0)[148], ggpubr (v.0.5.0)[149], ggrastr (v.1.0.2)[150], ggrepel (v.0.9.3)[151], grid (v.4.1.3)[152], gridExtra (v.2.3)[153], gtable (v.0.3.3)[154], matrixStats (v.0.63.0)[155], pheatmap (v.1.0.12)[156], plyr (v.1.8.8)[157], purrr (v.1.0.1)[158], RColorBrewer (v.1.1-3)[159], readxl (v.1.4.2)[160], reshape2 (v.1.4.4)[161], scales (v.1.2.1)[162], splitstackshape (v.1.4.8)[163], stats (v.4.1.3)[152], stringi (v.1.7.12)[164], stringr (v.1.5.0)[165], tidyr (v.1.3.0)[166] and viridis (v.0.6.2)[167].

## Reporting summary

Further information on research design is available in the Nature Portfolio Reporting Summary linked to this article.

## Data availability

Sequencing data generated in this study and processed sequencing files are available at the Neuroscience Multi-omic Data Archive (NeMO) (https://assets.nemoarchive.org/dat-bmx7s1t). The data are available under controlled use conditions set by human privacy regulations. To access the data, the requester must first create an account in DUOS (https://duos.broadinstitute.org) using their institutional email address. The signing official from the requester's institution must also register in DUOS to issue the requester a library card agreement. The requester will then need to fill out a data access request through DUOS, which will be reviewed by the Broad Institute's Data Access Committee. Once a request is approved, NeMO will be notified to authorize access to the data. Processed expression data can also be queried using the interactive public web interface that we created (https://dlpfc.mccarrolllab.org/app/dlpfc). The following publicly available datasets were also analysed: ProteomeXchange Dataset PXD026491 (ref. 82) and Gene Expression Omnibus Series GSE147672 (ref. 99). Source data are provided with this paper.

## Code availability

Software and core computational analysis to align and process sequencing reads and perform donor assignment are freely available at GitHub (https://github.com/broadinstitute/Drop-seq). Published or publicly available software, tools, algorithms and packages are cited with their version numbers in the text and Reporting Summary. Other custom code is available on request from the corresponding authors.

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

**Acknowledgements** This work was supported by the Broad Institute's Stanley Center for Psychiatric Research, a grant from the Simons Foundation (811233, S.A.M.), the National Institute of Mental Health (grants U01MH115727 to S.A.M. and P50MH115874 Project 5 to S.B.) and the National Human Genome Research Institute (grant T32 HG002295 to N.K.). Human tissue was obtained from the NIH NeuroBioBank. We thank H. de Rivera, R. Kohli and G. Lind for technical assistance; R. Hodge for advice on myelin removal; T. Bakken and N. Jorstad for advice on glutamatergic neuron subtype classification; F. Koopmans for SynGO analysis scripts; S. Nadendla, H. Huot Creasy, J. Receveur, T. Hodges, V. Felix and other members of NeMO for assistance with data deposition; the members of the McCarroll laboratory and the Stanley Center for advice and discussions; M. Babadi, K. Dickson, M. Florio, S. Hyman, Y. H. Kim, A. Nadig, R. Nehme, C. Patil, E. Robinson, M. Sheng and M. Tegtmeyer for comments on manuscript drafts; and the brain tissue donors and their families, without whom this study would not be possible.

**Author contributions** E.L., S.A.M. and S.B. designed the study. E.L., M.G., N.R. and S.A.M. developed and evaluated experimental strategies for snRNA-seq from pooled human brain tissue. E.L., M.G., N.R., A.L. and C.D.M. prepared and dissected tissue, performed snRNA-seq and prepared sequencing libraries. E.L., J.N., M.G. and S.A.M. performed sequencing, alignment and quality-control analyses. E.L., J.N., A.W. and S.A.M. developed analysis pipelines. E.L. and S.A.M. analysed the data with input from S.B., J.N. and N.K. R.E.H. performed analyses of *C4*. G.G. performed imputation and calculated polygenic risk scores. J.S.V. and S.B. provided tissue donor metadata. S. Gerges calculated MAGMA *z* scores and performed heritability enrichment analyses with S-LDSC. S.K. developed the scPred analysis pipeline and the RNA-expression web resource. S. Ghosh developed the pySCENIC analysis pipeline. J.M.E., K.M. and S.B. evaluated and provided tissue for snRNA-seq experiments. D.M. contributed to analysis pipelines. L.S. contributed to tissue sample management and standardization of the single-nucleus library preparation and sequencing protocol. A.N., M.H. and K.I. contributed to project management and sequencing. E.L., S.A.M. and S.B. wrote the paper with input from the other authors.

**Competing interests** The authors declare no competing interests.

**Additional information**

**Correspondence and requests for materials** should be addressed to Emi Ling, Sabina Berretta or Steven A. McCarroll.

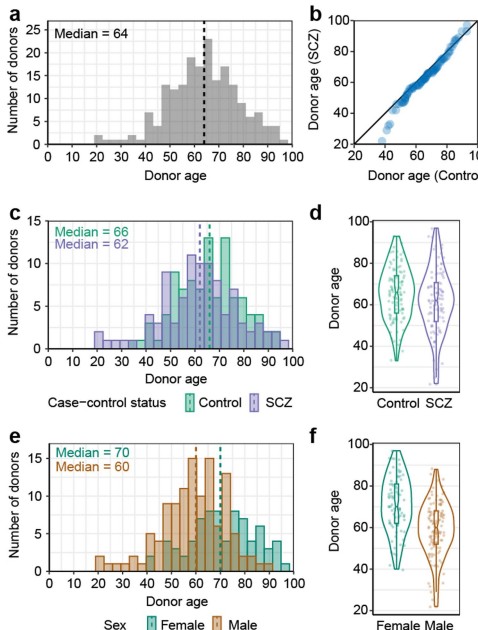

**Extended Data Fig. 1 | Ages of brain tissue donors. a**, Distribution of the ages
of brain donors (*n* = 191 donors). **b**, Distributions of donors' ages by schizophrenia
status, displayed as a quantile-quantile plot that compares ages of unaffected
control donors (*n* = 97 donors) to ages of donors with schizophrenia (*n* = 94
donors). **c**–**d**, Distributions of donors' ages separated by schizophrenia status
(*n* = 97 unaffected and 94 affected), displayed as **(c)** histograms and **(d)** violin
plots. **e**–**f**, Distributions of donors' ages, separated by sex (*n* = 75 women and
116 men), displayed as **(e)** histograms and **(f)** violin plots. Note that while female
brain donors are on average older than male donors, expression of SNAP (LF4)
did not associate with sex in either a naive or age-adjusted analysis (Extended
Data Fig. 4d,e), nor in a simultaneous regression on age, sex, and schizophrenia
affected/unaffected status (Supplementary Table 3).

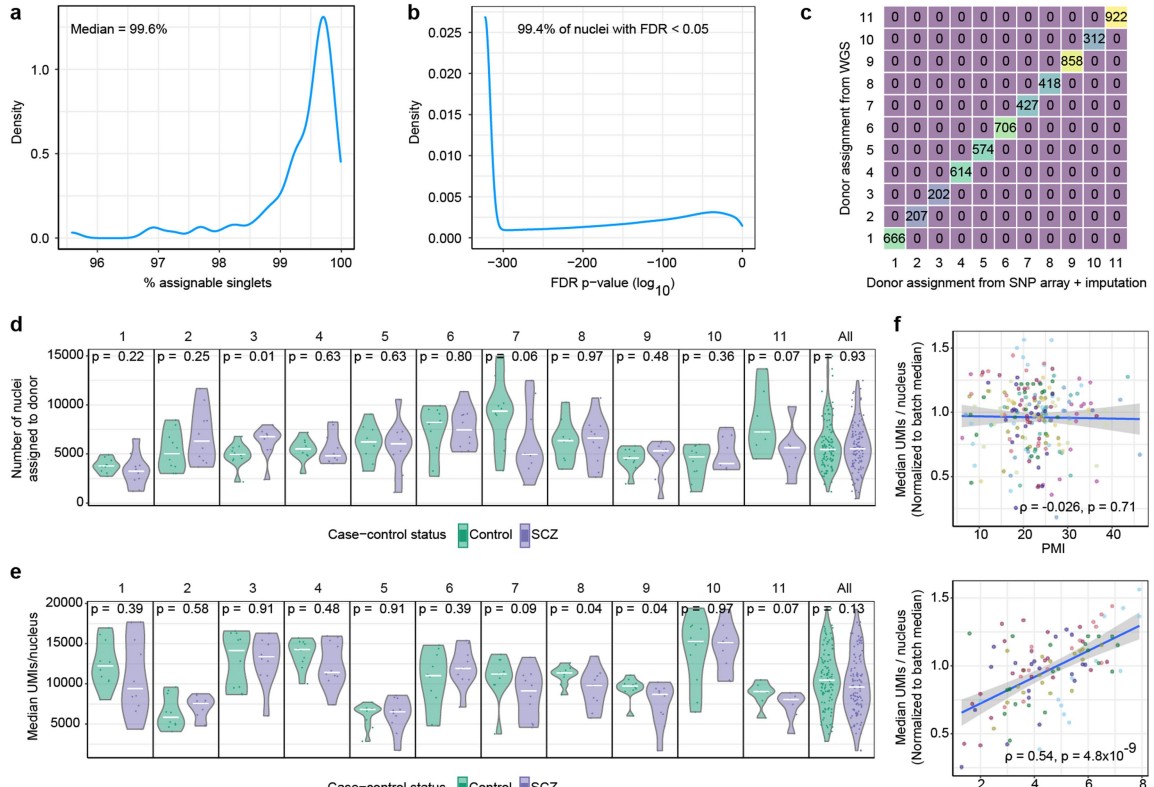

**Extended Data Fig. 2 | Single-donor assignment and sequencing metrics.**
**a**, Density plot showing the fraction of all nuclei that were determined to be
"singlets" (containing alleles from just one donor); $n = 1,262,765$ assignable
singlets out of 1,271,830. **b**, Density plot showing donor-assignment
likelihoods (as false discovery rates, on a log scale) for the 1,271,830 singlet
nuclei. **c**, Validation of the computational assignment of nuclei to individual
brain donors whose genomes have been analysed (individually) by SNP
array-genotyping plus imputation. The matrix displays the concordance of
single-donor assignment between whole-genome sequencing (WGS) (y-axis)
and SNP array + imputation (x-axis) for a pilot set of 11 donors whose genomes
were analysed by both methods. (Accuracy of donor assignment when WGS
data are available has been previously shown by)[7]. Each row/column
corresponds to one of the 11 donors, and each entry in the table displays the
number of nuclei that were assigned to a given donor (at a false discovery rate

of 0.05). **d**, Number of nuclei assigned to each donor in each of 11 batches or
(rightmost panel) across all batches, separated by schizophrenia case-control
status ($n = 10$ controls and 10 schizophrenia cases per batch). P-values from a
two-sided Wilcoxon rank-sum test comparing the affected to the unaffected
donors are reported at the top of each panel. Central lines represent medians.
**e**, Median number of UMIs ascertained per donor in each batch or (rightmost
panel) across all batches, separated by schizophrenia case-control status
($n = 10$ controls and 10 schizophrenia cases per batch). P-values from a
two-sided Wilcoxon rank-sum test comparing the affected to the unaffected
donors are reported at the top of each panel. Central lines represent medians.
**f**, Relationship of median UMIs/nucleus (normalized to the median value of the
donors in each donor's batch) to (top) post-mortem interval (PMI) and (bottom)
RIN score (Spearman's ρ). Colours represent different batches. Shaded regions
represent 95% confidence intervals.

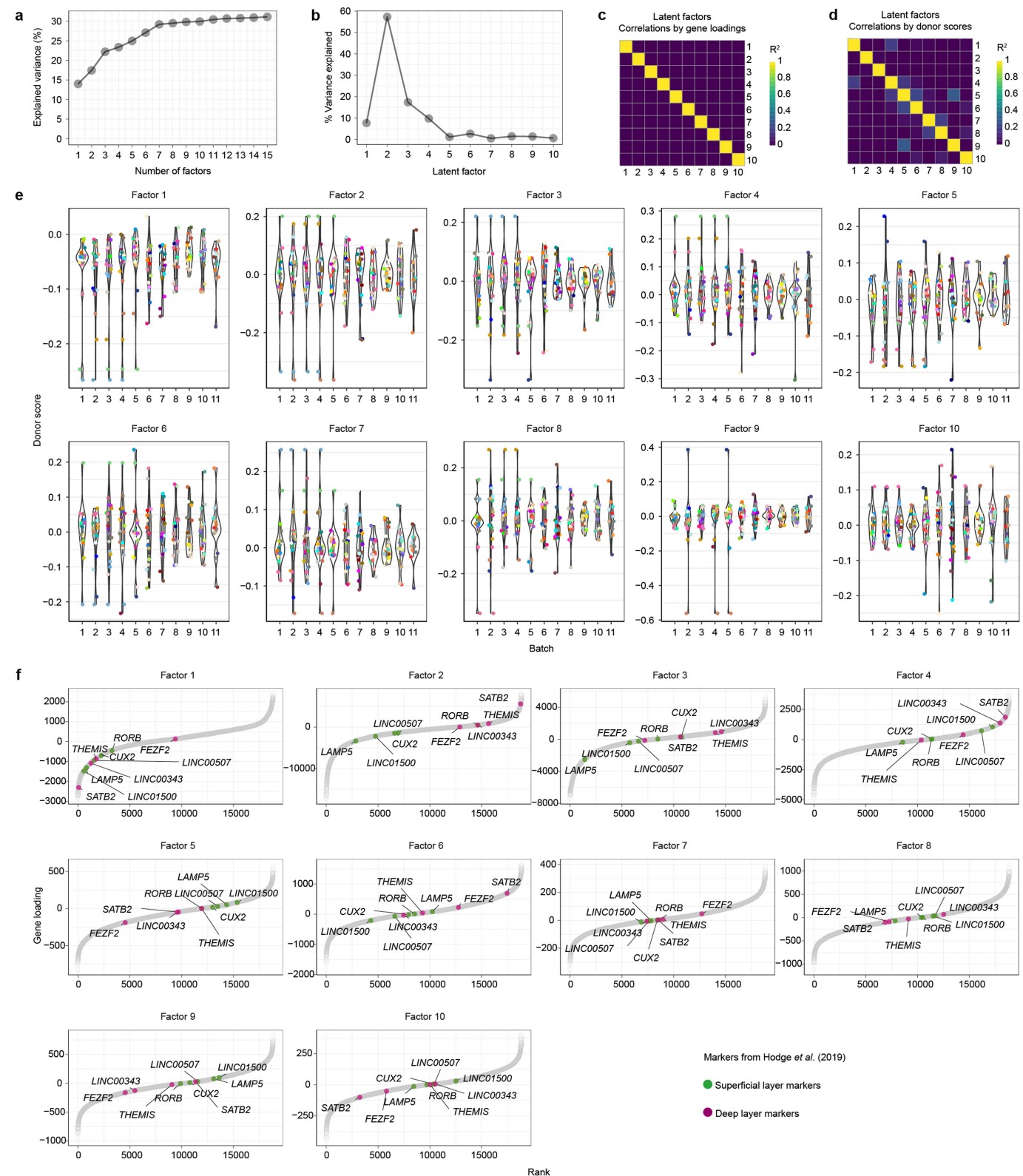

**Extended Data Fig. 3 | Properties of the latent factors inferred from snRNA-seq data. a**, Total % variance in expression explained by latent factors with different numbers of requested factors $k$. **b**, Fraction of variance explained by each latent factor in an analysis with 10 requested factors. **c–d**, Independence of latent factors, visualized as Pearson correlation heatmaps of factors' **(c)** gene loadings ($n = 125,437$ gene/cell-type combinations) and **(d)** donor scores ($n = 180$ donors). **e**, Expression level of each latent factor (panels) in each donor (points), split by batch ($n = 20$ donors per batch). **f**, Relationship of latent factors to markers of superficial and deep cortical layers from[75]. Markers label dominant classes of glutamatergic neurons (superficial: *LAMP5*, *LINC00507*, *RORB*; deep: *THEMIS*, *FEZF2*) or spatially restricted subtypes (superficial: Exc L2 LAMP5 LTK, marked by *CUX2* and *LINC01500*; deep: Exc L5-6 THEMIS C1QL3, marked by *SATB2* and *LINC00343*). Factor 2 exhibits the most distinct segregation of these superficial and deep layer markers when genes are ranked by their loadings onto each factor. $n = 18,830$ genes expressed in glutamatergic neurons; coloured dots are plotted over the dots of genes not among the markers listed above (grey).

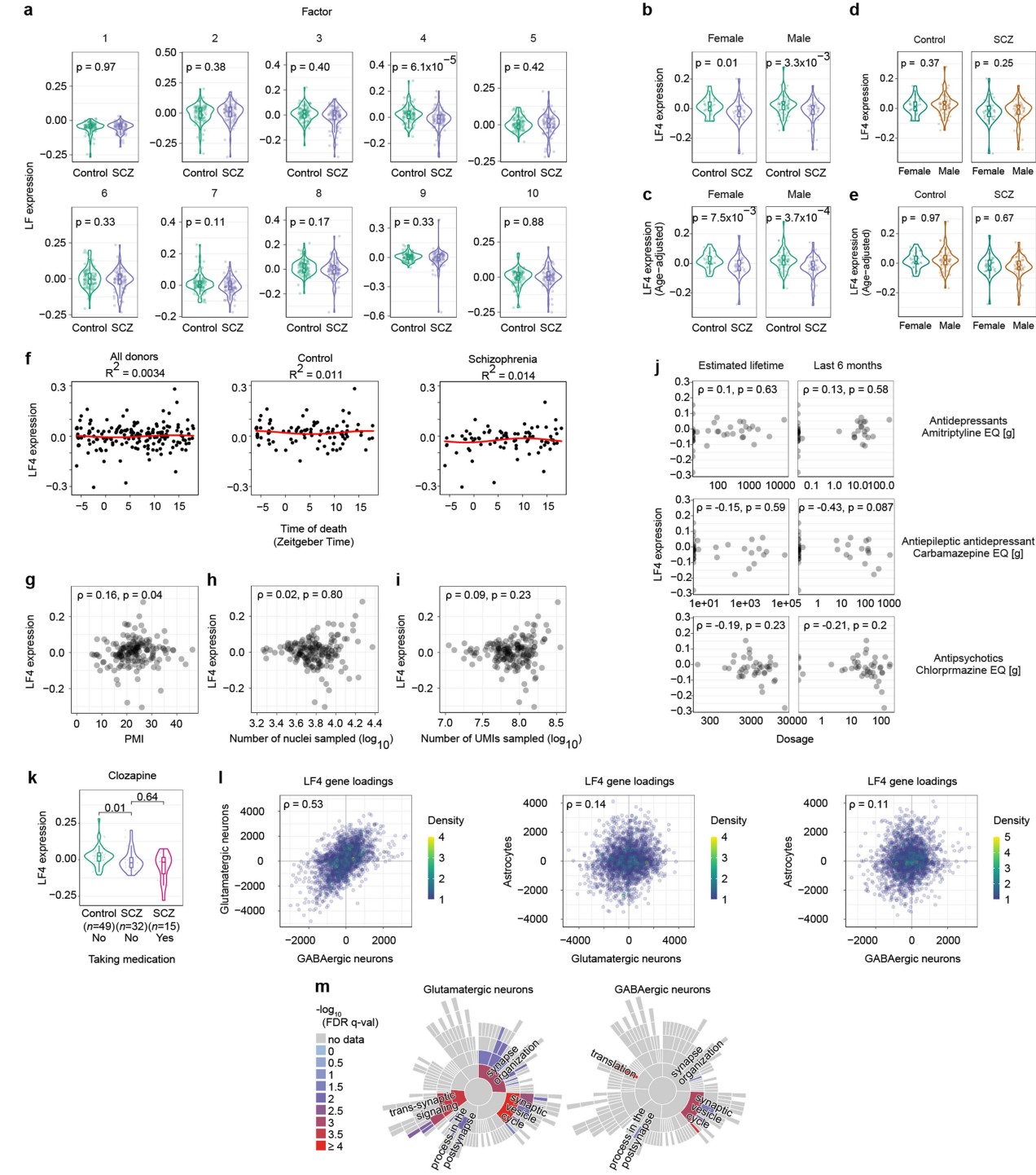

**Extended Data Fig. 4** | See next page for caption.

**Extended Data Fig. 4 | Properties of Latent Factor 4 (LF4). a**, Expression of each latent factor by case-control status ($n$ = 93 controls and 87 cases). P-values are from a two-sided Wilcoxon rank-sum test. Box plots show interquartile ranges; whiskers, 1.5x the interquartile interval; central lines, medians; notches, confidence intervals around medians. **b**, Expression of LF4 by case-control status, split by sex (female: $n$ = 31 controls and 39 cases; male: $n$ = 62 controls and 48 cases). P-values are from a two-sided Wilcoxon rank-sum test. Box plots show interquartile ranges; whiskers, 1.5x the interquartile interval; central lines, medians; notches, confidence intervals around medians. Note that the more-modest p-value for the females-only analysis relative to the males-only analysis appears to represent the smaller sample (70 females vs. 110 males) rather than a weaker relationship to schizophrenia status; please see also Extended Data Fig. 10h. **c**, Similar plots as in **b**, here displaying LF4 expression values adjusted for donor age. **d**, Expression of LF4 by sex, split by case-control status (controls: $n$ = 31 females and 62 males; cases: $n$ = 39 females and 48 males). P-values are from a two-sided Wilcoxon rank-sum test. Box plots show interquartile ranges; whiskers, 1.5x the interquartile interval; central lines, medians; notches, confidence intervals around medians. **e**, Similar plots as in **d**, here displaying LF4 expression values adjusted for donor age. **f–k**, Relationship of LF4 expression measurements to other available donor and tissue characteristics: **(f)** time of death in zeitgeber time (ZT), with rhythmicity analyses performed as in[83]; **(g)** post-mortem interval; **(h)** number of nuclei sampled; **(i)** number of UMIs sampled; **(j)** use of psychiatric medications (left column) across each donor's lifespan or (right column) in the last 6 months prior to death; and **(k)** use of clozapine. Correlation coefficients in **g–j** are Spearman's ρ. P-values in **k** are from a two-sided Wilcoxon rank-sum test. Box plots show interquartile ranges; whiskers, 1.5x the interquartile interval; central lines, medians; notches, confidence intervals around medians. **l**, See also Fig. 2a. LF4 involves broadly similar gene-expression effects in glutamatergic and GABAergic neurons, and a distinct set of gene-expression effects in astrocytes. Genes plotted are the protein-coding genes that are expressed (at levels of at least 10 UMIs per $10^5$) in both cell types (Spearman's ρ; $n$ = 1,538, 1,067, and 1,131 genes respectively). **m**, Concentrations of the strongest enriched neuronal gene-expression changes in LF4 among synaptic functions as annotated by SynGO[91]. Plots show categories of SynGO biological processes.

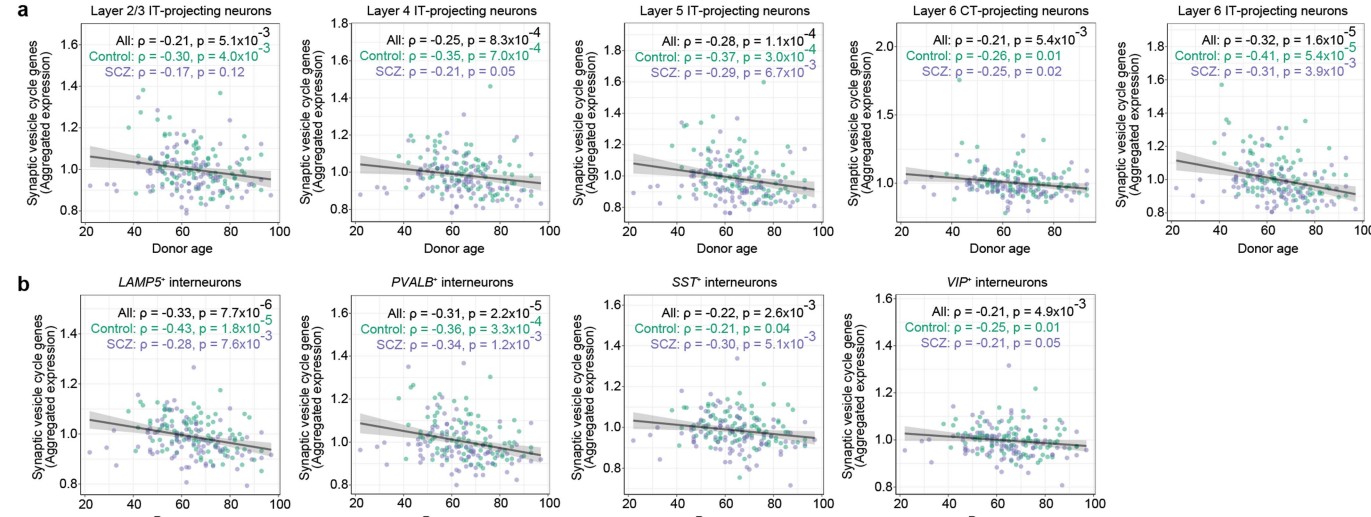

**Extended Data Fig. 5 | Relationship of synaptic vesicle cycle gene expression in neuronal subtypes to advancing age. a–b**, See also Fig. 2c. Neuronal expression of synaptic vesicle cycle genes in the most abundant subtypes of **(a)** glutamatergic and **(b)** GABAergic neurons (across 180 donors), plotted against donor age (Spearman's ρ). Expression values are the fraction of all UMIs in each donor (from the indicated subtype) that are derived from these genes, normalized to the median expression among control donors. Shaded regions represent 95% confidence intervals. The observed decline in schizophrenia and aging was consistent with earlier observations that expression of genes for synaptic components is reduced in schizophrenia[168] and with advancing age[169].

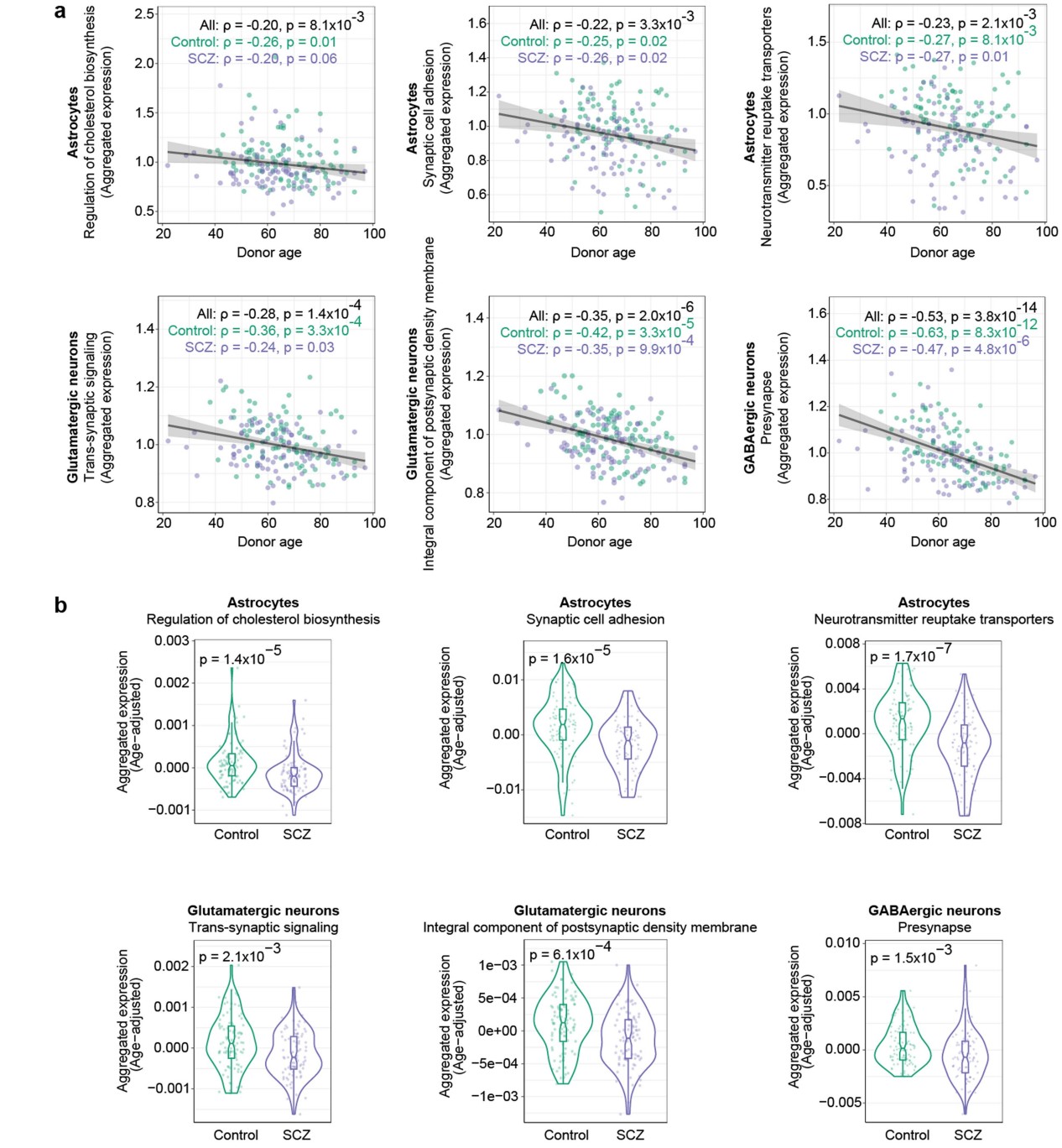

**Extended Data Fig. 6 | Relationship of gene-set expression in astrocytes and neurons to advancing age and schizophrenia. a**, Expression of gene sets enriched in the astrocyte and neuronal components of LF4 (across 180 donors), plotted against donor age (Spearman's ρ). Expression values are the fraction of all UMIs in each donor (from the indicated cell type) that are derived from these genes, normalized to the median expression among control donors. Shaded regions represent 95% confidence intervals. **b**, Expression (by donor, separated by schizophrenia case-control status; *n* = 180 donors) of gene sets enriched in the astrocyte and neuronal components of LF4. Expression values are the fraction of all UMIs in each donor (from the indicated cell type) that are derived from these genes, adjusted for donor age. P-values from a two-sided Wilcoxon rank-sum test comparing the affected to the unaffected donors are reported at the top of each panel. Box plots show interquartile ranges; whiskers, 1.5x the interquartile interval; central lines, medians; notches, confidence intervals around medians.

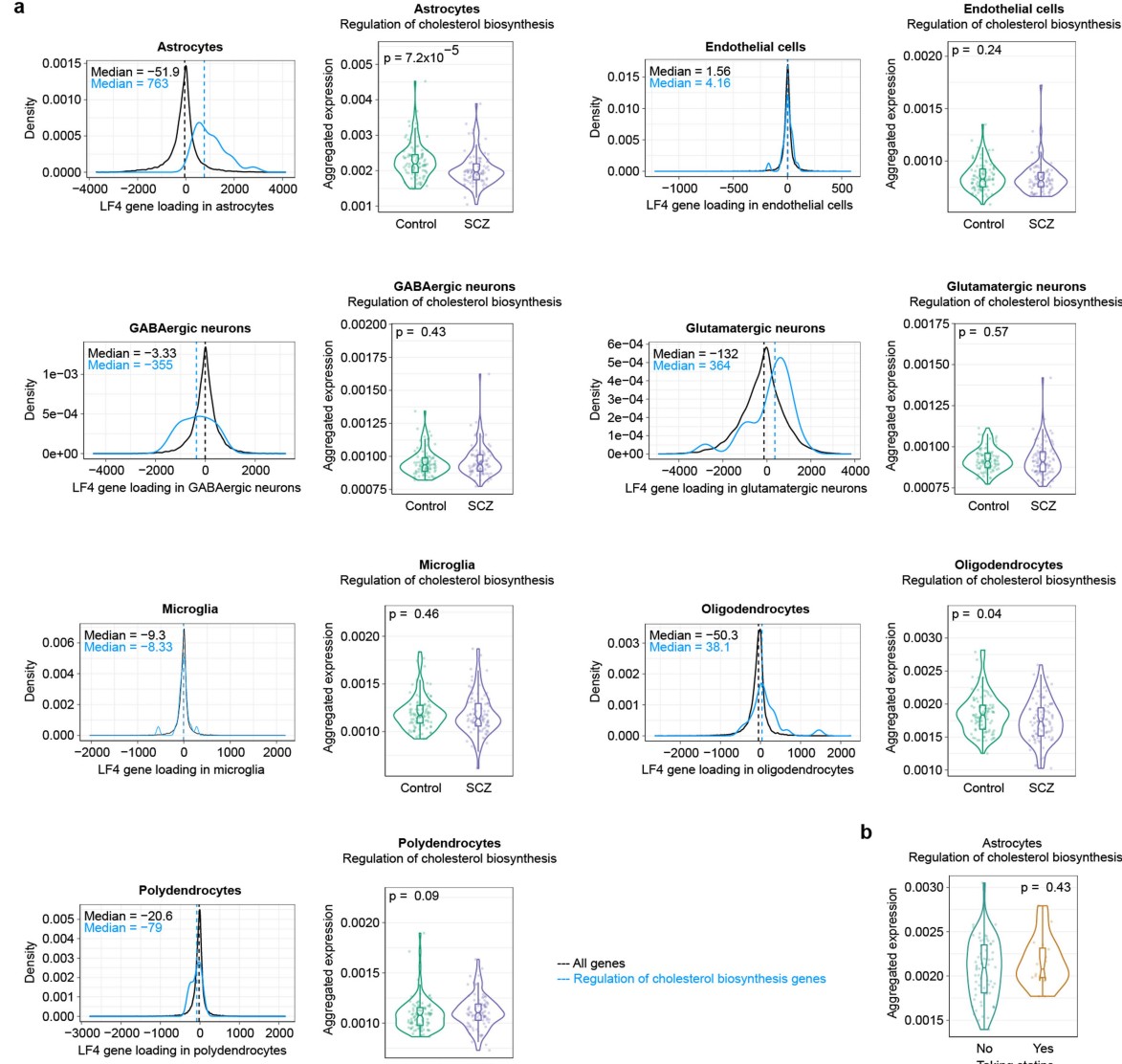

**Extended Data Fig. 7 | Expression of cholesterol-biosynthesis genes in cortical cell types. a**, See also Fig. 2d. For each cortical cell type: (Left) Distributions of LF4 gene loadings for (black) all expressed genes and (blue) specifically for genes annotated by GO as having roles in cholesterol biosynthesis (core genes contributing to the enrichment of GO:0045540 ("cholesterol biosynthesis genes") in that cell type's component of LF4. (Right) Each cell type's expression of cholesterol biosynthesis genes (by donor, split by schizophrenia case-control status; $n$ = 180 donors). Expression values are the fraction of all UMIs in each donor (from the indicated cell type) that are derived from these genes. P-values are from a two-sided Wilcoxon rank-sum test

comparing the affected to the unaffected donors. Box plots show interquartile ranges; whiskers, 1.5x the interquartile interval; central lines, medians; notches, confidence intervals around medians. **b**, Expression in astrocytes of cholesterol biosynthesis genes by donor, separated by statin intake among donors with available medication data ($n$ = 63 donors not taking statins and 16 donors taking statins). Expression values are the fraction of all UMIs in each donor's astrocytes that are derived from these genes. P-value is from a two-sided Wilcoxon rank-sum test. Box plots show interquartile ranges; whiskers, 1.5x the interquartile interval; central lines, medians; notches, confidence intervals around medians.

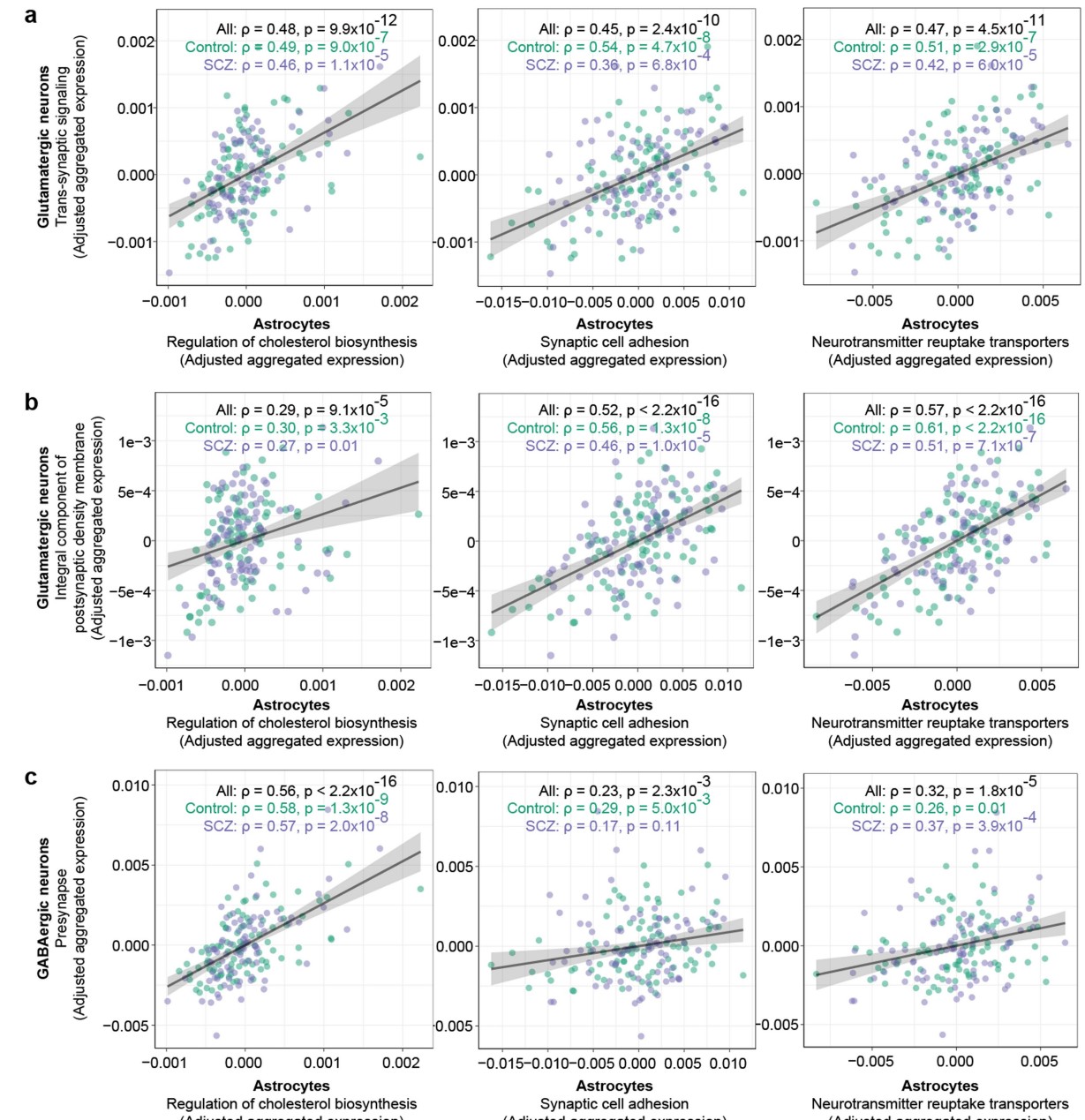

**Extended Data Fig. 8 | Concerted synaptic investments by neurons and astrocytes, adjusted for age and schizophrenia case-control status.** **a**–**c**, See also Fig. 2e. Relationship of donors' neuronal gene expression to astrocyte gene expression (Spearman's ρ), adjusted for age and case-control status. Astrocyte gene sets plotted on the x-axes represent (left) cholesterol biosynthesis, (middle) synaptic adhesion, and (right) neurotransmitter reuptake transporters. Neuronal gene sets plotted on the y-axes represent **(a)** trans-synaptic signalling, **(b)** integral component of postsynaptic density, and **(c)** presynapse genes. Expression values are the fraction of all UMIs in each donor (from the indicated cell type) that are derived from these genes, adjusted for donor age and schizophrenia case-control status. Shaded regions represent 95% confidence intervals.

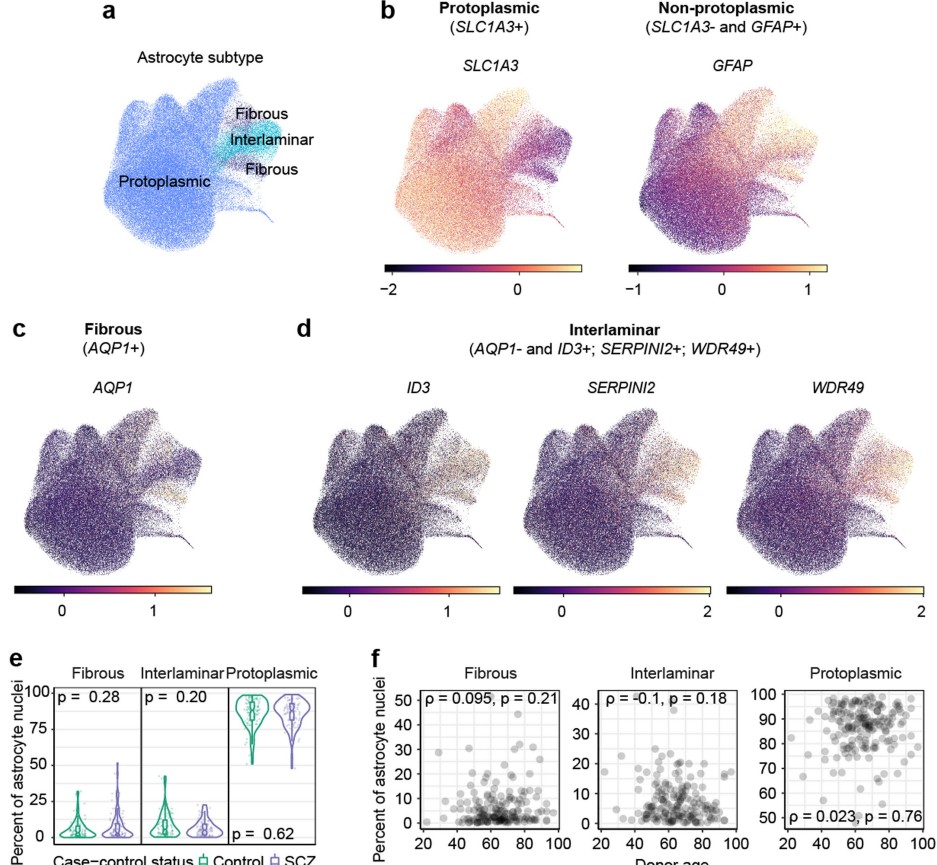

**Extended Data Fig. 9 | Astrocyte subtype classification and proportions across donors. a**, Two-dimensional projection of the RNA-expression profiles of 179,764 astrocyte nuclei from 180 donors, reproduced from Fig. 3a. Nuclei are coloured by their assignments to subtypes of astrocytes using classifications from[75] and[76]. The same projection is used in panels b to d. **b–d**, Expression levels of marker genes for subtypes of **(b)** protoplasmic astrocytes (*SLC1A3+*) and non-protoplasmic astrocytes (*SLC1A3−* and *GFAP+*) comprising the **(c)** fibrous (*AQP1+*) and **(d)** interlaminar (*AQP1−* and *ID3+*, *SERPINI2+*, and *WDR49+*) subtypes. Markers are from[75] or from transcriptomically similar subtypes in[76].

Values represent Pearson residuals from variance stabilizing transformation (VST). **e**, Proportions of astrocyte subtypes in BA46 by schizophrenia status (*n* = 93 unaffected and 87 affected). P-values from a two-sided Wilcoxon rank-sum test comparing the affected to the unaffected donors are reported at the top of each panel. Box plots show interquartile ranges; whiskers, 1.5x the interquartile interval; central lines, medians; notches, confidence intervals around medians. **f**, Relationship of sampled astrocyte subtype proportions to donor age (Spearman's ρ).

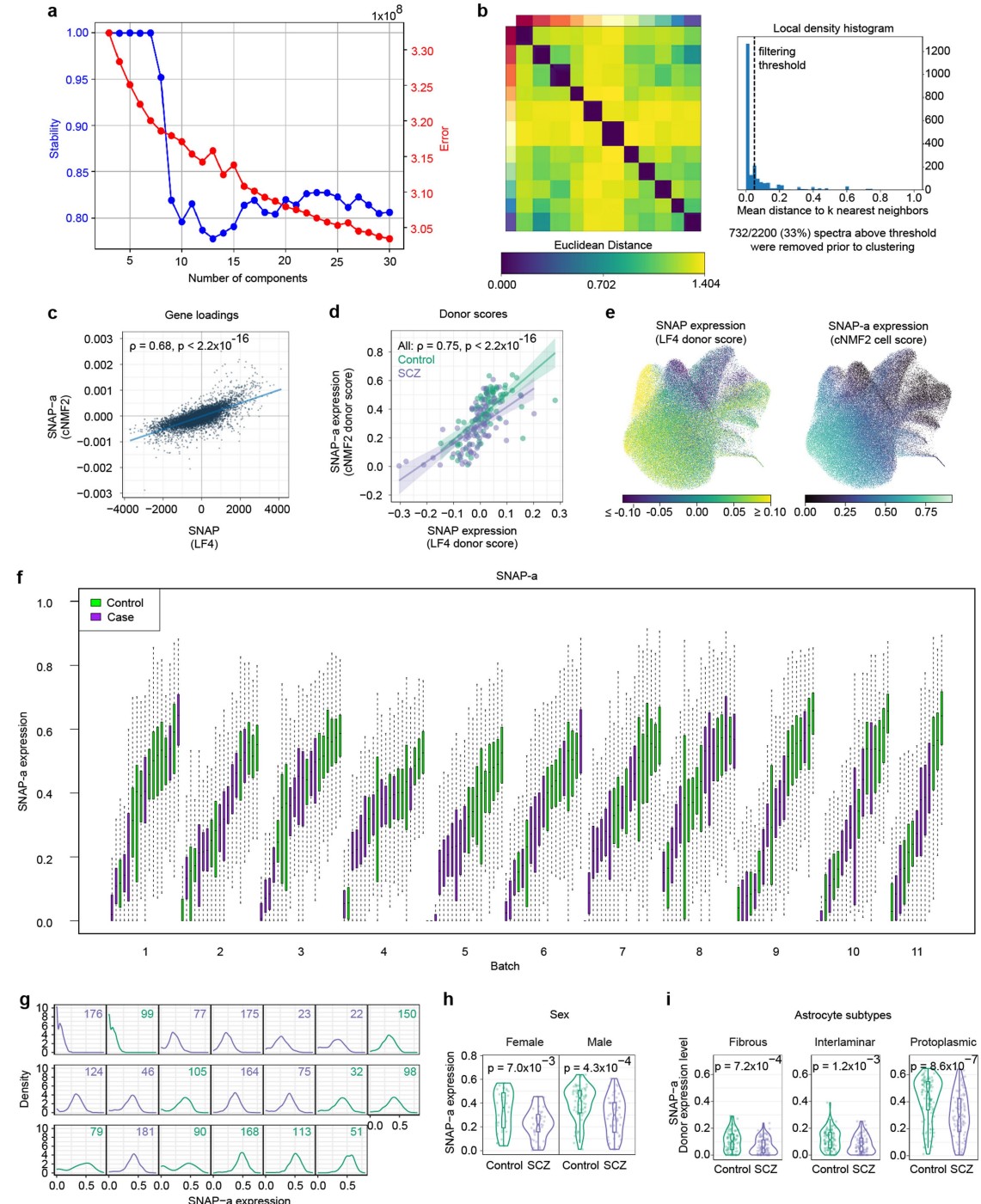

**Extended Data Fig. 10** | See next page for caption.

**Extended Data Fig. 10 | Astrocyte gene-expression programs inferred by cNMF (SNAP-a) and their relationship to SNAP. a**, Visualization of the trade-off between error and stability of cNMF factors as a function of the number of factors $k$. 11 factors were requested based on these results. **b**, Clustergram of consensus matrix factorization estimates. Each colour on the x- and y-axes represents one of 11 cNMF factors. **c-d**, Relationship of SNAP-a to SNAP by **(c)** gene loadings ($n$ = 33,611 genes) and **(d)** donors' expression levels of each factor ($n$ = 180 donors) (Spearman's ρ). Shaded regions represent 95% confidence intervals. **e**, UMAP of RNA-expression patterns from 179,764 astrocyte nuclei from 180 donors, using the same projection from Fig. 3a–c. Nuclei are coloured by (left) each donor's expression of SNAP or (right) each cell's expression of the astrocyte component of SNAP (cNMF2, also referred to as SNAP-a). SNAP-a is reproduced from Fig. 3c for comparison with SNAP. **f**, Distributions of SNAP-a expression levels among astrocytes in each donor, split by experimental batch. Box plots show interquartile ranges; whiskers, 1.5x the interquartile interval; central lines, medians. **g**, Density plots showing distributions of SNAP-a expression levels among astrocytes in each donor for one representative batch (batch 4) out of 11 batches. Labels in top-right corners indicate anonymized research IDs at the Harvard Brain Tissue Resource Center. Colours represent case-control status (green: controls; purple: schizophrenia cases). At the single-astrocyte level, SNAP-a expression exhibited continuous, quantitative variation rather than discrete state shifts by a subpopulation of astrocytes, supporting the idea that astrocyte biological variation extends beyond polarized states[17,170,171], particularly in genes strongly loading onto SNAP-a[172–181]. **h**, Distributions of SNAP-a expression levels by case-control status, split by sex. P-values from a two-sided Wilcoxon rank-sum test comparing the affected to the unaffected donors are reported at the top of each panel. Box plots show interquartile ranges; whiskers, 1.5x the interquartile interval; central lines, medians; notches, confidence intervals around medians. **i**, Distributions of SNAP-a expression levels by case-control status, split by astrocyte subtype. P-values from a two-sided Wilcoxon rank-sum test comparing the affected to the unaffected donors are reported at the top of each panel. Box plots show interquartile ranges; whiskers, 1.5x the interquartile interval; central lines, medians; notches, confidence intervals around medians.

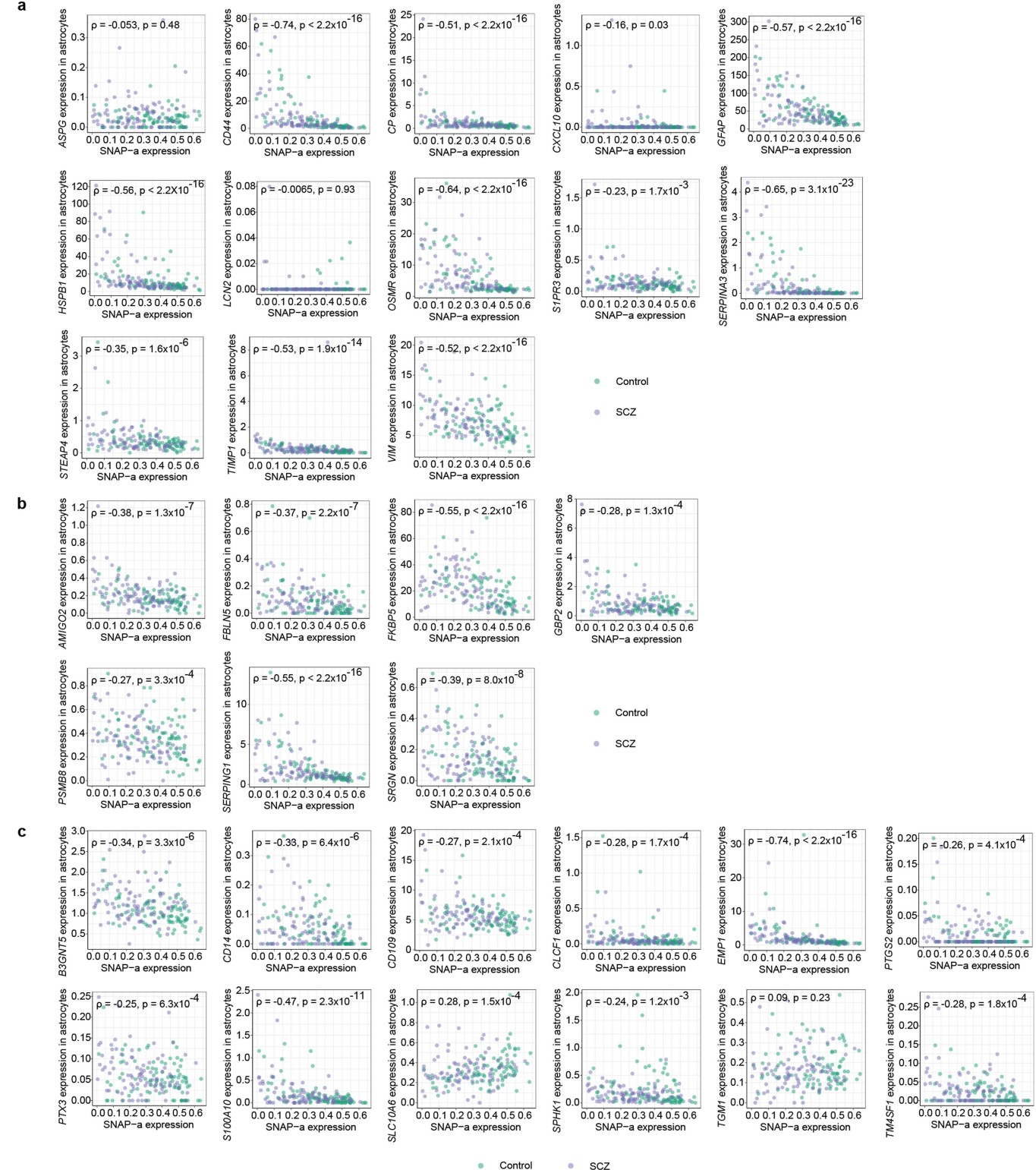

**Extended Data Fig. 11 | Relationship of reactive astrocyte marker expression to SNAP-a expression.** Relationship of donors' expression levels of reactive astrocyte marker genes to SNAP-a expression (Spearman's ρ). Markers are from[16] and represent **(a)** pan-reactive (PAN), **(b)** A1, and **(c)** A2 reactive astrocytes.

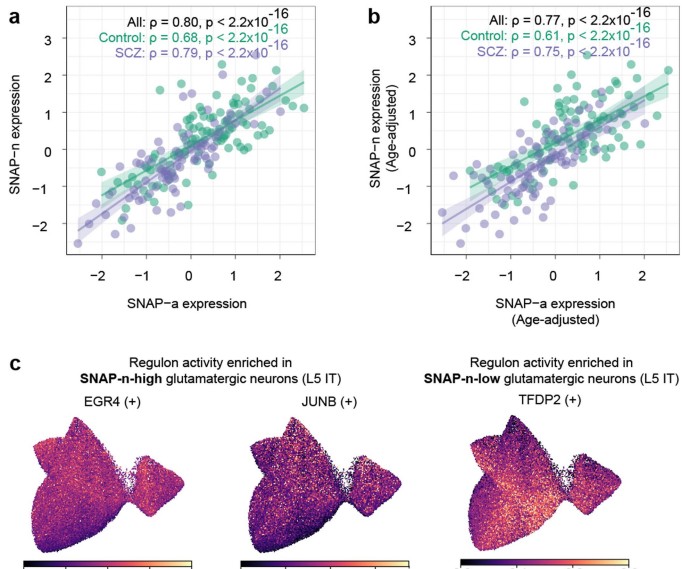

**Extended Data Fig. 12 | Biological states and transcriptional programs of L5 IT glutamatergic neurons in schizophrenia. a–b**, Relationship of SNAP-a to SNAP-n (Spearman's ρ). Values plotted are **(a)** quantile-normalized and **(b)** donor age-adjusted, quantile-normalized donor scores for each factor. Shaded regions represent 95% confidence intervals. **c**, UMAP of regulon activity scores (as inferred by pySCENIC[98]) from L5 IT glutamatergic neuron nuclei from 180 donors, using the same projection from Fig. 3f–h. Regulons plotted are the most strongly enriched in L5 IT glutamatergic neurons with high versus low SNAP-n expression. (+) indicates that the targets of the indicated regulon were found to be upregulated in expression.

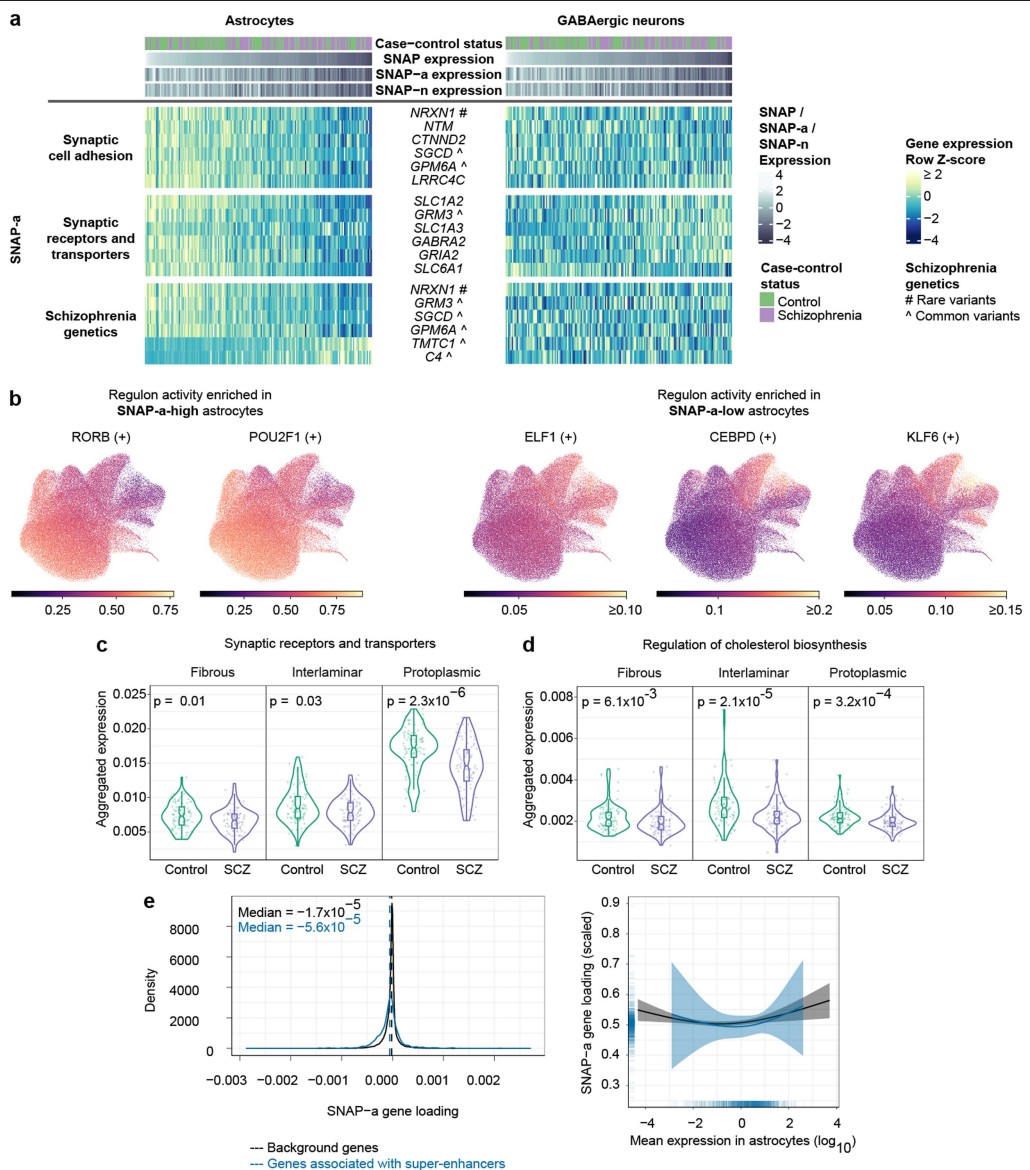

**Extended Data Fig. 13 | Astrocyte gene-expression programs underlying SNAP-a. a**, See also Fig. 3k. Concerted expression in (left) astrocytes and (right) GABAergic neurons of genes strongly recruited by SNAP-a. These were enriched in genes encoding synaptic-adhesion proteins, intrinsic components of synaptic membranes such as transporters and receptors, as well as genes strongly implicated in human genetic studies of schizophrenia. Genes in the "Schizophrenia genetics" heatmap are from among the prioritized genes from[23] (FDR < 0.05) or[22]. Genes annotated by ^ are from among all genes at loci implicated by common variants in[22], regardless of prioritization status. **b**, UMAP of regulon activity scores (as inferred by pySCENIC[98]) from 179,764 astrocyte nuclei from 180 donors, using the same projection from Fig. 3a–c. Regulons plotted are the most strongly enriched in astrocytes with high versus low SNAP-a expression. (+) indicates that the targets of the indicated regulon are predicted to be upregulated in expression. **c–d**, Transcriptional investments (by donor, separated by schizophrenia case-control status) in **(c)** genes encoding synaptic receptors and transporters and **(d)** cholesterol biosynthesis genes,

in subtypes of astrocytes. Quantities plotted are the fraction of all UMIs in each subtype that are derived from these genes. P-values from a two-sided Wilcoxon rank-sum test comparing the affected to the unaffected donors are reported at the top of each panel. Box plots show interquartile ranges; whiskers, 1.5x the interquartile interval; central lines, medians; notches, confidence intervals around medians. **e**, Relationship of SNAP-a expression to association with super-enhancers. Genes expressed in astrocytes were grouped based on whether their promoters were predicted to contact super-enhancers in astrocytes (using bulk H3K27ac HiChIP and scATAC-seq data from[99]), and SNAP-a loadings were compared between the two groups. (Left) Distributions of SNAP-a gene loadings for (blue) 1,286 genes whose promoters are predicted to contact super-enhancers in astrocytes and (black) the set of 32,325 remaining expressed background genes. (Right) Binomial smooth results of scaled SNAP-a gene loadings versus $\log_{10}$-scaled mean expression values in astrocytes, shown separately for the two groups. Shaded regions represent 95% confidence intervals.

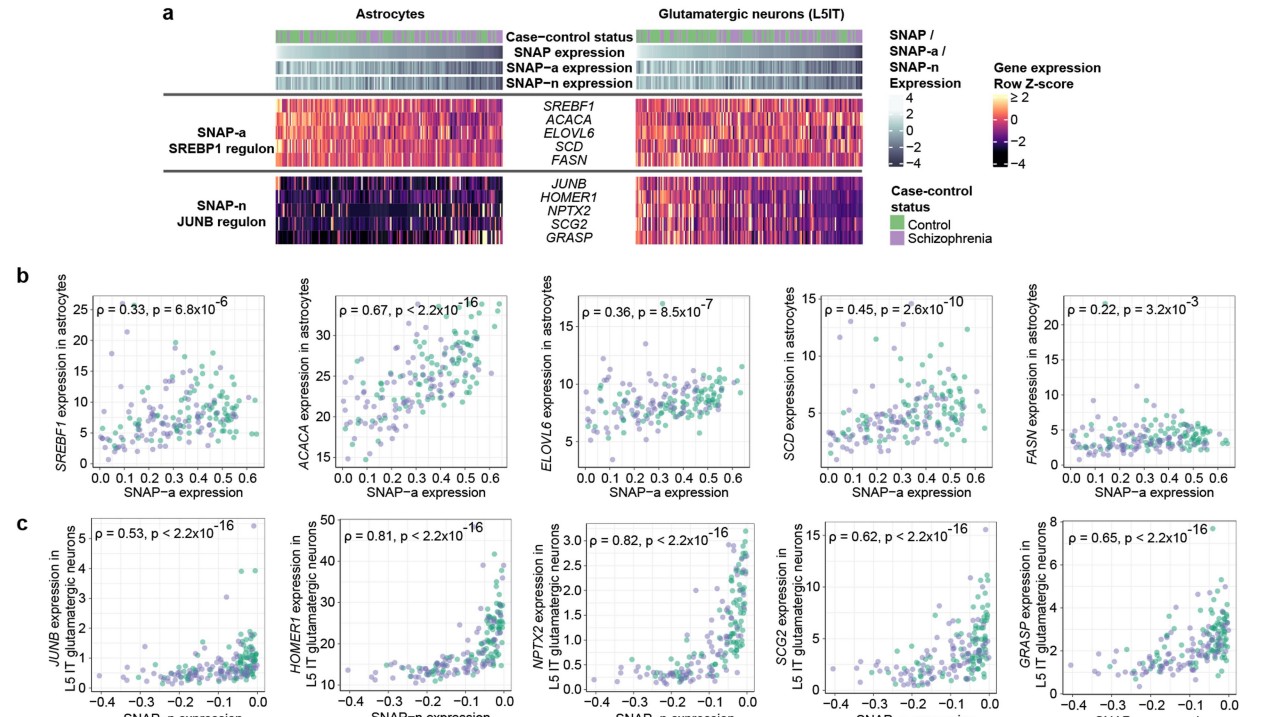

**Extended Data Fig. 14 | Expression of well-characterized transcriptional programs in SNAP-a and SNAP-n. a**, Concerted expression in (left) astrocytes and (right) L5 IT glutamatergic neurons of target genes of known transcriptional programs specifically active in SNAP-a or SNAP-n. Genes are listed in decreasing order by their importance for each regulon as scored by pySCENIC. **b**, Relationship of donors' expression levels of known SREBP1 target genes (involved in fatty acid biosynthesis)[18,182,183] to SNAP-a expression (Spearman's ρ). Target-gene expression levels in astrocytes are shown. **c**, Relationship of donors' expression levels of known JUNB target genes (that are late-response genes)[19,20,184] to SNAP-n expression (Spearman's ρ). Target-gene expression levels in L5 IT glutamatergic neurons are shown.

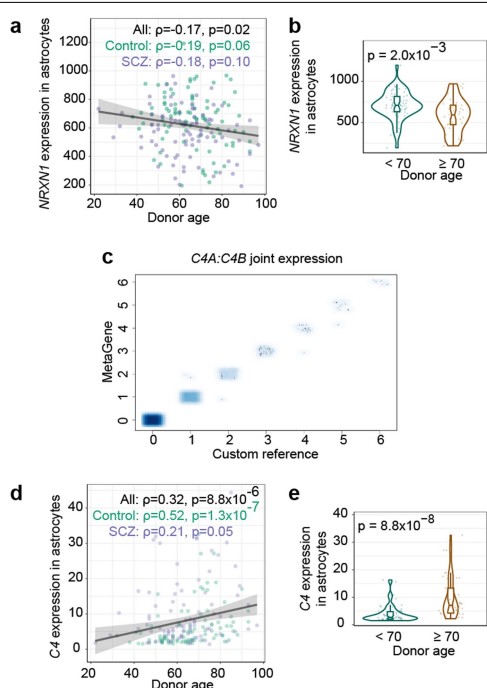

**Extended Data Fig. 15 | Relationship of astrocytic *NRXN1* and *C4* expression to advancing age. a**, Relationship of *NRXN1* expression to age in astrocytes (Spearman's ρ). Shaded region represents 95% confidence interval. **b**, Expression of *NRXN1* in astrocytes in control donors, split by donor age (*n* = 56 donors younger than 70 years old and 37 donors 70 years old or older). P-value is from a two-sided Wilcoxon rank-sum test. Box plots show interquartile ranges; whiskers, 1.5x the interquartile interval; central lines, medians; notches, confidence intervals around medians. **c**, Validation of a metagene computational approach for identifying RNA transcripts (UMIs) from the *C4* genes. Standard analysis approaches tend to discard sequence reads from *C4A* or *C4B* because these genes are almost identical in sequence, differing only at a few key positions (far from the 3' end), such that most reads are discarded due to low mapping quality. To measure expression of these genes, UMIs were either aligned to a custom reference genome that contained only one *C4* gene (x-axis) or were processed through a custom pipeline that identified UMIs associated with sets of gene families with high sequence homology, including *C4A/C4B* (y-axis). The two approaches (custom reference approach and joint expression of *C4A/C4B* via the metagene approach) arrived at concordant *C4* UMI counts in 15,664 of 15,669 cells tested. Note that these measurements do not distinguish between *C4A* and *C4B*. **d**, Relationship of *C4* expression to age in astrocytes (Spearman's ρ). Shaded region represents 95% confidence interval. **e**, Expression of *C4* in astrocytes in control donors, split by donor age (*n* = 56 donors younger than 70 years old and 37 donors 70 years old or older). P-value is from a two-sided Wilcoxon rank-sum test. Box plots show interquartile ranges; whiskers, 1.5x the interquartile interval; central lines, medians; notches, confidence intervals around medians.

|  | Steven A. McCarroll |
|  | Emi Ling |

# Reporting Summary

## Statistics

For all statistical analyses, confirm that the following items are present in the figure legend, table legend, main text, or Methods section.

| n/a | Confirmed | |
|---|---|---|
| ☐ | ☒ | The exact sample size (*n*) for each experimental group/condition, given as a discrete number and unit of measurement |
| ☐ | ☒ | A statement on whether measurements were taken from distinct samples or whether the same sample was measured repeatedly |
| ☐ | ☒ | The statistical test(s) used AND whether they are one- or two-sided *Only common tests should be described solely by name; describe more complex techniques in the Methods section.* |
| ☐ | ☒ | A description of all covariates tested |
| ☐ | ☒ | A description of any assumptions or corrections, such as tests of normality and adjustment for multiple comparisons |
| ☐ | ☒ | A full description of the statistical parameters including central tendency (e.g. means) or other basic estimates (e.g. regression coefficient) AND variation (e.g. standard deviation) or associated estimates of uncertainty (e.g. confidence intervals) |
| ☐ | ☒ | For null hypothesis testing, the test statistic (e.g. *F*, *t*, *r*) with confidence intervals, effect sizes, degrees of freedom and *P* value noted *Give P values as exact values whenever suitable.* |
| ☒ | ☐ | For Bayesian analysis, information on the choice of priors and Markov chain Monte Carlo settings |
| ☒ | ☐ | For hierarchical and complex designs, identification of the appropriate level for tests and full reporting of outcomes |
| ☐ | ☒ | Estimates of effect sizes (e.g. Cohen's *d*, Pearson's *r*), indicating how they were calculated |

*Our web collection on statistics for biologists contains articles on many of the points above.*

## Software and code

Policy information about availability of computer code

| Data collection | Software and core computational analysis for the following data collection steps are freely available at https://github.com/broadinstitute/Drop-seq: Drop-seq (v2.4.1) (Macosko et al. 2015) - align and process sequencing reads Dropulation (v2.4.1) (Wells et al. 2023) - perform donor assignment |
|---|---|
| Data analysis | Software/tools/algorithms/packages used for data analyses are listed below and also cited in the text. anndata (v0.8.0) (Virshup et al. 2021) - https://anndata.readthedocs.io BBKNN (v1.5.1) (Polański et al. 2020) - https://github.com/Teichlab/bbknn BCFtools (v1.16) (Danecek et al. 2021) - https://www.htslib.org Bowtie2 (v2.2.4) (Langmead et al. 2012) - https://bowtie-bio.sourceforge.net/bowtie2 CellBender (v0.1.0) (Fleming et al. 2023) - https://github.com/broadinstitute/CellBender cluster (v2.1.2) (Maechler et al. 2022) - https://CRAN.R-project.org/package=cluster clustree (v0.4.4) (Zappia et al. 2018) - https://github.com/lazappi/clustree CNA (v0.1.4) (Reshef et al. 2022) - https://github.com/immunogenomics/cna cNMF (v1.2) (Kotliar et al. 2019) - https://github.com/dylkot/cNMF Color Oracle (v1.3) (Jenny et al. 2006; Jenny et al. 2007) - https://github.com/nvkelso/color-oracle-java ComplexHeatmap (v2.10.0) (Gu et al. 2016; Gu 2022) - https://github.com/jokergoo/ComplexHeatmap data.table (v1.14.8) (Dowle et al. 2023) - https://CRAN.R-project.org/package=data.table DescTools (v0.99.48) (Signorell 2023) - https://CRAN.R-project.org/package=DescTools dplyr (v1.1.2) (Wickham et al. 2023) - https://CRAN.R-project.org/package=dplyr fastICA (v1.2-1) (Marchini et al. 2017) - https://CRAN.R-project.org/package=fastICA |

FUMA (v1.5.6) (Watanabe et al. 2017) - https://fuma.ctglab.nl
gdata (v2.19.0) (Warnes et al. 2023) - https://CRAN.R-project.org/package=gdata
GenCall (v3.0.0) (Kermani 2006)
Genome STRiP (v2.0) (Handsaker et al. 2015) - https://software.broadinstitute.org/software/genomestrip
ggforce (v0.4.1) (Pedersen 2022) - https://CRAN.R-project.org/package=ggforce
ggplot2 (v3.4.2) (Wickham 2016) - https://ggplot2.tidyverse.org
ggpmisc (v0.5.3) (Aphalo 2023) - https://CRAN.R-project.org/package=ggpmisc
ggpointdensity (v0.1.0) (Kremer 2019) - https://CRAN.R-project.org/package=ggpointdensity
ggpubr (v0.5.0) (Kassambara 2022) - https://CRAN.R-project.org/package=ggpubr
ggrastr (v1.0.2) (Petukhov et al. 2023) - https://CRAN.R-project.org/package=ggrastr
ggrepel (v0.9.3) (Slowikowski 2023) - https://CRAN.R-project.org/package=ggrepel
grid (v4.1.3) (R Core Team 2022) - https://www.R-project.org
gridExtra (v2.3) (Auguie 2017) - https://CRAN.R-project.org/package=gridExtra
GSEA (v4.0.3) (Subramanian et al. 2005; Mootha et al. 2003) - https://www.gsea-msigdb.org/gsea/index.jsp
gtable (v0.3.3) (Wickham et al. 2023) - https://CRAN.R-project.org/package=gtable
IMPUTE5 (v1.1.5) (Rubinacci et al. 2020) - https://jmarchini.org/software/#impute-5
lme4 (v1.1-31) (Bates et al. 2015) - https://github.com/lme4/lme4
MAGMA (v1.08) (de Leeuw et al. 2015) - http://ctglab.nl/software/magma
Matplotlib (v3.5.2) (Hunter et al. 2007) - https://matplotlib.org
matrixStats (v0.63.0) (Bengtsson 2022) - https://CRAN.R-project.org/package=matrixStats
minpack.lm (v1.2-4) (Elzhov et al. 2022) - https://CRAN.R-project.org/package=minpack.lm
MoChA WDL (v2022-12-21) (Loh et al. 2018; Loh et al. 2020) - https://github.com/freeseek/mochawdl
NumPy (v1.17.5) (Harris et al. 2020) - https://numpy.org
Osprey (v0.1-9) (Handsaker et al. 2022) - https://github.com/broadinstitute/Osprey
pandas (v1.0.5) (The pandas development team 2020; McKinney 2010) - https://pandas.pydata.org
PEER (v1.0) (Stegle et al. 2012) - https://github.com/PMBio/peer
pheatmap (v1.0.12) (Kolde 2019) - https://CRAN.R-project.org/package=pheatmap
plyr (v1.8.8) (Wickham 2011) - https://plyr.had.co.nz
presto (v1.0.0) (Korsunsky et al. 2022) - https://immunogenomics.github.io/presto
purrr (v1.0.1) (Wickham et al. 2023) - https://CRAN.R-project.org/package=purrr
pySCENIC (v0.11.2) (Aibar et al. 2017; Van de Sande et al. 2020) - https://github.com/aertslab/SCENICprotocol
RColorBrewer (v1.1-3) (Neuwirth 2022) - https://CRAN.R-project.org/package=RColorBrewer
readxl (v1.4.2) (Wickham et al. 2023) - https://CRAN.R-project.org/package=readxl
reshape2 (v1.4.4) (Wickham 2007) - https://github.com/hadley/reshape
ROSE (v1.3.1) (Whyte et al. 2013; Lin et al. 2013) - https://github.com/stjude/ROSE
S-LDSC (v1.0.1) (Finucane et al. 2015) - https://github.com/bulik/ldsc
samtools (v1.3.1) (Danecek et al. 2021) - https://www.htslib.org
scales (v1.2.1) (Wickham et al. 2023) - https://CRAN.R-project.org/package=scales
Scanpy (v1.9.1) (Wolf et al. 2018) - https://scanpy.readthedocs.io
score (v2022-12-21) (Loh et al. 2018; Loh et al. 2020) - https://github.com/freeseek/mochawdl
scPred (v1.9.2) (Alquiciria-Hernandez et al. 2019) - https://github.com/powellgenomicslab/scPred
sctransform (v0.3.1) (Hafemeister et al. 2019) - https://github.com/satijalab/sctransform
seaborn (v0.10.1) (Waskom et al. 2021) - https://seaborn.pydata.org
Seurat (v3.2.2) (Stuart et al. 2019) - https://satijalab.org/seurat
SeuratDisk (v0.0.0.9010) (Hoffmann et al. 2022) - https://mojaveazure.github.io/seurat-disk
SHAPEIT4 (v4.2.2) (Delaneau et al. 2019) - https://github.com/odelaneau/shapeit4
splitstackshape (v1.4.8) (Mahto 2019) - https://CRAN.R-project.org/package=splitstackshape
stats (v4.1.3) (R Core Team 2022) - https://www.R-project.org
stringi (v1.7.12) (Gagolewski 2022) - https://stringi.gagolewski.com
stringr (v1.5.0) (Wickham 2022) - https://CRAN.R-project.org/package=stringr
tidyr (v1.3.0) (Wickham et al. 2023) - https://CRAN.R-project.org/package=tidyr
Trimmomatic (v0.33) (Bolger et al. 2014) - http://www.usadellab.org/cms/?page=trimmomatic
viridis (v0.6.2) (Garnier et al. 2021) - https://sjmgarnier.github.io/viridis

For manuscripts utilizing custom algorithms or software that are central to the research but not yet described in published literature, software must be made available to editors and reviewers. We strongly encourage code deposition in a community repository (e.g. GitHub). See the Nature Portfolio guidelines for submitting code & software for further information.

# Data

Policy information about availability of data

All manuscripts must include a data availability statement. This statement should provide the following information, where applicable:
- Accession codes, unique identifiers, or web links for publicly available datasets
- A description of any restrictions on data availability
- For clinical datasets or third party data, please ensure that the statement adheres to our policy

Sequencing data generated in this study and processed sequencing files are available through the Neuroscience Multi-omic Data Archive (NeMO) (RRID:SCR_016152) at https://assets.nemoarchive.org/dat-bmx7s1t. The data are available under controlled use conditions set by human privacy regulations. To access the data, the requester must first create an account in DUOS (https://duos.broadinstitute.org) using their institutional email address. The Signing Official from the requester's institution must also register in DUOS to issue the requester a Library Card Agreement. The requester will then need to fill out a Data Access Request through DUOS, which will be reviewed by the Broad Institute's Data Access Committee. Once a request is processed, NeMO will be notified to authorize access to the data. Processed expression data can also be queried using an interactive public web interface that we created (https://sz.mccarrolllab.org/app/SZ). Source data with anonymized donor IDs are provided with this paper.

The following publicly available datasets were also analyzed: ProteomeXchange Dataset PXD026491 (Karayel et al. 2022) and Gene Expression Omnibus Series GSE147672 (Corces et al. 2020).

# Research involving human participants, their data, or biological material

Policy information about studies with <u>human participants or human data</u>. See also policy information about <u>sex, gender (identity/presentation), and sexual orientation</u> and <u>race, ethnicity and racism</u>.

| | |
|---|---|
| Reporting on sex and gender | Brain donors were selected for this study so that within every batch (containing 10 schizophrenia cases and 10 controls), every donor with schizophrenia had a sex- and age-matched control. Information relevant to consent for sharing of individual-level data is described under "Ethics oversight". Key findings from this paper apply to both sexes (i.e. SNAP, SNAP-a). |
| Reporting on race, ethnicity, or other socially relevant groupings | This study does not use socially constructed or socially relevant categorization variables. |
| Population characteristics | Brain tissue samples were obtained from 191 postmortem donors (97 controls, 94 schizophrenia cases). Median age, 64; median postmortem interval, 22.7 hours; no significant differences in these covariates between cases and controls. |
| Recruitment | Brain donors were recruited by the Harvard Brain Tissue Resource Center/NIH NeuroBioBank (HBTRC/NBB), in a community-based manner, across the USA. To minimize biases for this specific study, donors unaffected by nervous system disorders were selected as sex- and age-matched controls for donors with schizophrenia. Consensus diagnosis of schizophrenia was carried out by retrospective review of medical records and extensive questionnaires concerning social and medical history provided by family members. Several regions from each brain were examined by a neuropathologist. We excluded subjects with evidence for gross and/or macroscopic brain changes, or with clinical history consistent with cerebrovascular accident or other neurological disorders. Subjects with Braak stages III or higher (modified Bielchowsky stain) were not included. None of the subjects had significant history of substance dependence within 10 or more years from death, as further corroborated by negative toxicology reports. Absence of recent substance abuse is typical for samples from the HBTRC, which receives exclusively community-based tissue donations. |
| Ethics oversight | Human brain tissue was obtained from the HBTRC/NBB. The HBTRC procedures for informed consent by the donor's legal next-of-kin and distribution of de-identified postmortem tissue samples and demographic and clinical data for research purposes are approved by the Mass General Brigham Institutional Review Board. Post-mortem tissue collection followed the provisions of the United States Uniform Anatomical Gift Act of 2006 described in the California Health and Safety Code section 7150 and other applicable state and federal laws and regulations. Federal regulation 45 CFR 46 and associated guidance indicates that the generation of data from de-identified post-mortem specimens does not constitute human participant research that requires institutional review board review. |

Note that full information on the approval of the study protocol must also be provided in the manuscript.

# Field-specific reporting

Please select the one below that is the best fit for your research. If you are not sure, read the appropriate sections before making your selection.

☒ Life sciences  ☐ Behavioural & social sciences  ☐ Ecological, evolutionary & environmental sciences

For a reference copy of the document with all sections, see <u>nature.com/documents/nr-reporting-summary-flat.pdf</u>

# Life sciences study design

All studies must disclose on these points even when the disclosure is negative.

| | |
|---|---|
| Sample size | No sample size calculation was performed. Sample size was determined by the limited availability of donors per diagnosis who met the study criteria (as described in "Recruitment"); these sizes were sufficient as they are similar to or exceed sample sizes from published bulk RNA-seq studies on postmortem human brain tissue. We aimed to analyze at least 50 donors per group (schizophrenia cases and age/sex-matched controls) to account for potential drop-outs in the analysis pipeline, which could be due to factors such as poor tissue quality and genotyping issues that prevented assignment of nuclei to an expected donor. |
| Data exclusions | Singlet nuclei were excluded if they could not be confidently assigned to a single expected donor (at FDR < 0.05), if they strongly expressed markers of more than one cell type and grouped together in a distinct cluster in UMAP space, or were assigned to donors with gene-expression profiles and/or cell-type-proportions that were distinct from the other donors in the cohort. Additional details are described in Methods. |
| Replication | Replication of the single-nucleus RNA-seq experiments was not attempted due to the limited availability of donors per diagnosis. We attempted to validate our findings using an orthogonal approach, and found evidence that an analogous constellation of changes also manifests at a protein level in variation among different individuals' cerebrospinal fluid protein profiles (analysis of data from Karayel et al. 2022). |
| Randomization | Randomization was not used. Specimens were allocated into batches of 20 specimens per batch, ensuring that the same number of cases and age-matched controls (10 per diagnosis), and men and women (10 per sex) were included in each batch. Specimens from cases and age- |

matched controls were also processed in alternating order within each batch.

Blinding

To ensure that batch compositions were balanced, investigators were not blinded to the batch allocation or processing order of each specimen (as described for "Randomization"). Researchers had access to unique numerical codes assigned to the donor-of-origin of each specimen as well as basic donor metadata (e.g. case-control status, age, sex).

# Reporting for specific materials, systems and methods

We require information from authors about some types of materials, experimental systems and methods used in many studies. Here, indicate whether each material, system or method listed is relevant to your study. If you are not sure if a list item applies to your research, read the appropriate section before selecting a response.

## Materials & experimental systems

| n/a | Involved in the study |
|---|---|
| ☒ | ☐ Antibodies |
| ☒ | ☐ Eukaryotic cell lines |
| ☒ | ☐ Palaeontology and archaeology |
| ☒ | ☐ Animals and other organisms |
| ☒ | ☐ Clinical data |
| ☒ | ☐ Dual use research of concern |
| ☒ | ☐ Plants |

## Methods

| n/a | Involved in the study |
|---|---|
| ☒ | ☐ ChIP-seq |
| ☒ | ☐ Flow cytometry |
| ☒ | ☐ MRI-based neuroimaging |

## Plants

Seed stocks

*Report on the source of all seed stocks or other plant material used. If applicable, state the seed stock centre and catalogue number. If plant specimens were collected from the field, describe the collection location, date and sampling procedures.*

Novel plant genotypes

*Describe the methods by which all novel plant genotypes were produced. This includes those generated by transgenic approaches, gene editing, chemical/radiation-based mutagenesis and hybridization. For transgenic lines, describe the transformation method, the number of independent lines analyzed and the generation upon which experiments were performed. For gene-edited lines, describe the editor used, the endogenous sequence targeted for editing, the targeting guide RNA sequence (if applicable) and how the editor was applied.*

Authentication

*Describe any authentication procedures for each seed stock used or novel genotype generated. Describe any experiments used to assess the effect of a mutation and, where applicable, how potential secondary effects (e.g. second site T-DNA insertions, mosiacism, off-target gene editing) were examined.*

