## [Peer Review File · Nature]

Manuscript Title: A concerted neuron-astrocyte program declines in aging and schizophrenia

Reviewer Comments & Author Rebuttals

Reviewer Reports on the Initial Version:

Referees' comments:

Referee #1 (Remarks to the Author):

The authors present a compelling analysis of single cell transcriptome data collected from a sizable number of human donors with and without schizophrenia diagnoses. The consideration of data from distinct cell types here allows for the discovery of complementary cellular activities that are coordinated in their expression across different cell types, including gene expression changes in neurons and astrocytes pertaining to synaptic functions (SNAP). These SNAP genes are differentially expressed in schizophrenia, suggesting a specific domain of altered cellular activities in schizophrenia cortex.

This paper makes an especially strong case for the field to critically reconsider the frequently used shorthand assumption that highest gene expression indicates highest importance in specific contexts. Also, the application of latent factor analysis here provides a method for other single cell studies for moving beyond consideration of genes most abundantly and specifically expressed in a certain cell type (cell type marker genes) to identify additional gene sets with altered activity in particular cell types, which could be linked to disease biology.

The authors additionally use these data to contextualize known schizophrenia genetic risk factors, including defining the strong astrocytic contribution of NRXN1 and C4 expression, and the identification of cell type-specific eQTL colocalization with schizophrenia GWAS loci.

I enjoyed the opportunity to review this manuscript and to get a firsthand look at the new ideas presented here. The logical flow of analyses and findings are clearly presented in both the text and figures. A few comments and questions:

(1) Characterization of cell type-specific eQTLs is a sensible analysis to run on these data, though these results feel somewhat separate from the primary focus on SNAP. I encourage the authors to more directly consider how one might jointly interpret cell type-specific eQTL colocalization and SNAP's relationship with schizophrenia genetic risk, in Discussion or elsewhere. A majority (but not all) of schizophrenia-colocalized eQTLs are associated with neuronal expression changes (Fig 4F), while SNAP-a and SNAP-n expression correlate with subsets of schizophrenia risk genes' expression in the corresponding cell type (Fig 3F, S17F). Is the take home message simply that both neurons and astrocytes perform cellular activities that are impacted by schizophrenia risk variants? And/or, a simple way to directly link these analyses could be to note (e.g. in Fig 4F) each colocalized gene's SNAP-a and SNAP-n factor loading. Are neuron-specific colocalized eQTL genes more strongly loaded

on SNAP-n, and astrocyte-specific eQTL genes on SNAP-a?

(2) Lines 120-121 note that five factors point to coordinated changes across cell types, though it is not immediately apparent in Fig 2B which five factors those are. To clarify, list factor names in text, or add notation to Fig 2B.

(3) Though there was no difference in the relative abundance of astrocyte subtypes in schizophrenia, is SNAP-a/cNMF2 more strongly driven by one particular subtype? Fig 3D would seem to suggest that expression is predominantly coming from protoplasmic astrocytes, though this could also be an effect of their high abundance. Does this gene set also have reduced expression in interlaminar and fibrous astrocytes from schizophrenia donor samples?

(4) This may be more common knowledge than I am aware, but a quick definition of “regulon” ~line 266 would be helpful.

(5) If there is room, I would suggest adding a quick sentence after line 280 to note that different regulons are associated with SNAP-n than with SNAP-a. It is interesting that the expression of these gene sets are likely driven by different primary regulators.

(6) The comparison of eQTL cell type vs. brain region specificity is an interesting one (Fig 4C), and the choice to avoid highlighting effect size differences in the main figures is logical. However, comparing p-values is also potentially tricky, given the influence of sample size on p-value. Sample size is not perfectly matched across this study and GTEx, nor between GTEx brain regions. The brain regions with the most depressed p-values across the 7733 genes displayed are also the regions with the fewest genotyped samples in GTEx (spinal cord c-1, substantia nigra, amygdala, N=114-129 as listed in current GTEx portal). Perhaps comparing effect size estimates that have been normalized across all genes tested in each data set (beyond the 7733 displayed) would more fairly present this point?

(7) I could not find this information in figure legends – how has column and/or row order been determined in Fig 1D, E, F (donors), Fig 3F (gene order in rows), Fig 4C (genes in rows, cell types or regions in columns)?

(8) Note, paragraph at lines 192-199 references Fig 2H, should be 2I.

Referee #2 (Remarks to the Author):

Ling and colleagues explore the inter-individual variability of transcript levels in human brain cells as a new approach to detect genes that are regulated in tandem across cell types thus implying "transcellular" functions. In a first part, the authors identify groups of genes expressed by neurons and – importantly – astrocytes (named Synaptic Neuron Astrocyte Program) that show correlated transcription across neurotypical donors and reduced expression in schizophrenia patients. This "program" comprises structural components of synapses, enzymes mediating cholesterol biosynthesis and genes that have previously been identified as risk factors for schizophrenia.

Notably, the results implicate that two of these established risk factors, NRXN1 and C4, are expressed by astrocytes. In a second smaller part, the authors use the same dataset to reveal the impact of expression quantitative trait loci on transcript levels of selected genes in different cell types.

The authors' approach to learn about cell-specific contributions to brain function and disease based on correlated cell-specific transcriptomic variation is – to this referee's knowledge – new and highly original. The authors' work suggests that "control groups" – ubiquitous in biomedical research – represent a treasure trove. This is of extraordinary interest and could mark a milestone opening new perspectives to reveal cell-cell interactions in the brain and in other organs/tissues in healthy and sick people based on "natural" inter-individual variation. There is a substantial body of work exploring inter-individual variation in biological systems and the authors should refer to relevant articles (for example, impact of biological noise reviewed by Eling et al., 2019 Nat Rev Genetics; brain-specific: Makowski et al., 2022 Science).

The methods and the display of data are state-of-the-art except for some specific points outlined below. Notably, the authors deliver an extremely precious – and possibly expandable – dataset comprising an impressive number of donors of either sex covering a wide range of ages. There are two general points of criticism with respect to data analysis. The authors' approach throughout the ms to compare neurotypical controls and patients without proper stratification based on sex and age is hard to accept given the impact of these factors on brain cells (for the latest with respect to age see e.g. Allen et al., Cell 2023) and possibly confounding effects on the authors' analyses. Looking naively at the large number of graphs comparing neurotypical and diseased donors one gets the impression that for most measures, there is substantial overlap of the values between these groups rather than "real" differences. Curiously, the authors show a sex-related difference (Fig. S6B), which they do not comment on in the text. And they mention the lack of age ("life time")-related differences (line 128) without showing the results. Analysis of results using proper stratification appears as indispensable. A second fundamental issue that admittedly applies to many single-cell transcriptome studies is whether the numbers of nuclei collected from individual brains – although truly impressive – can be considered representative of the cell counts in the specific region of interest. This is all the more important as the authors seek to delineate insight from inter-individual and inter-cellular variation. Can the authors estimate, which percentage of cells of each type in the prefrontal cortex they have analysed compared to the "real" number of cells per individual? Can the authors comment on the errors induced by this systematic "undersampling"?

Unfortunately, the manuscript presents severe weaknesses with respect to clarity and context that undermine its value, notably in the introductory paragraphs, where the overall goals of the study are not clearly stated and too little background is given. The abstract and introduction contain vague and diverging statements such as "to better understand neurobiological variation" (lines 25) or "we explored the idea that inter-individual variation could reveal regulatory coordination at a tissue level" (lines 49-50) or "understand how human neurobiology varies" (line 57). The background information is too "selective" and seems to anticipate the results rather than to inform the reader about the state of the art. Again, statements are rather vague (e.g. lines 50-51 "the tendency of many gene-expression changes, perhaps in multiple cell types to be implemented together in the same individuals and contexts") or lack proper reference to previous work (line 48). Importantly, the

relevance of schizophrenia for this work remains unclear. Is the disease the main topic of the study? One would think so based on the results. The authors must formulate concisely the goals of their work and expose the relevant background in sufficient detail.

The conclusions stated by the authors are not fully justified by the data. Obviously, a major point of concern that again applies to many publications in the field is whether results that are purely based on however elaborate bioinformatic analyses have biologic meaning. Evidently, one wishes to have independent validation or lines of evidence that – for example – SNAP is "real", that it is important for brain function and that it breaks down in the prefrontal cortex of schizophrenic patients. Obviously, this is a challenge for everybody working in the field. Still, a step in this direction may be to show, at least for a defined set of genes contained in SNAP, whether transcript and protein levels are correlated in human brain cells. A second approach could be to explore in more detail whether SNAP components are in fact controlled by well-known transcriptional programs. The authors provide first evidence, but this could be expanded. Finally, the authors could analyse age- and sex-dependent changes and test whether they reveal "ground truth" impacts of these factors that have been established previously by other approaches. This could provide a sort of internal validation of their transcript analyses.

Apart from these fundamental points, the authors may consider the following comments and suggestions:

- A general concern with the ms that also undermines its value is the combination of two stories that are based on the same dataset, but obviously unrelated. The results presented in lines 315-409 analyse cell type-specific gene expression using eQTL following up on the recent Yazar and Perez papers. Obviously, this is a very important and exciting aspect, but this part resembles more a last-minute addition than an integral component of the SNAP story. The authors should consider to publish the results from the two studies separately.

- Research over the last decade has revealed that the term "astrocyte" comprises a heterogeneous group of cells. Latest evidence comes for example from Su et al. 2022 Cell Stem Cell focusing on the hippocampus. This notion should alert the authors to explore in more detail (and similar to what they did for neurons) subtypes of astrocytes. This applies for example to the abundances and impact of schizophrenia on astrocytes subtypes (lines 79-96), cell-type specificity of latent factors including LF4 (lines 119-140), biological processes (lines 162-172), and certainly the exploration of SNAP programs (lines 201-262) etc. It seems that the data shown in Fig. 3 and in corresponding supplementary figures stop short of a profound analysis of astrocytes given the current evidence for their heterogeneity. A more in-depth exploration may reveal that only specific astrocytes subtypes are involved in SNAP and that only these are affected by disease: they may downregulate cholesterol synthesis (or lipid metabolism) as claimed by the authors – or degenerate altogether.

- As mentioned above, inter-individual variation has distinct sources including "noise" in the largest sense. A concrete question is whether the neuronal and astrocytic genes contained in the SNAP have particular genomic properties that "pretend" co-regulation although there is none. Such a particular feature could be – for example – the presence of TATA boxes and CpG islands that have been shown to increase and decrease transcriptional variability, respectively (Faure et al., 2017). Higher

transcript variability may imply correlation just "by chance". The authors should consider to address these points to exclude structural genomic features as cause for apparent transcriptional co-regulation.

- The authors' exposure of astrocytic cholesterol synthesis is interesting and could provide further support for a long-standing hypothesis. However, their choice to focus on this particular GO term appears as subjective. In fact, half of the genes indicated in Fig. 2G under the headline "cholesterol biosynthesis" are unrelated to this process, and more concerned with fatty acid or lipid metabolism in general, possibly reflecting SREBF1 activation. This suggests that the focus on cholesterol is too narrow. The authors should provide a more detailed analysis and display of the astrocyte- (and certainly neuron-) specific pathways that are "contained" in SNAP and SNAP-a. In this context, one wonders how the SREBF-dependent modules behave in neurons, do they change in inverted directions? Finally, given established functions of astrocytes, one wonders whether SNAP comprises components that handle neurotransmitter and energy requirements.

Minor points

- Line 32 etc. The authors should consider to replace the word "program" by "module" or similar, which seem more neutral with respect to function.

- Line 42: The term "transcellular" is somewhat ill-defined, it could be replaced by "intercellular" or "multicellular" depending on the context.

- Lines 61-77: This part should be rewritten or removed altogether. In any case, technical details should be moved to the Methods section, duplicate statement (number of donors etc.) must be avoided.

- Line 65: To provide a global view on their donor cohort, authors should show the age/sex distribution of control subjects and schizophrenia patients using histograms.

- Lines 65: The authors should indicate in the main text (here or before) that they analysed frozen post-mortem samples.

- Results: The headlines (and the subdivision) need to be revised as there is a mix of simple terms ("experimental approach", "cell-type abundances") and summarizing statements. Moreover, some paragraphs do not fit under the respective headline (example: lines 271-280).

- Line 98: The term "cell-nonautonomous biology" should be replaced.

- Line 105: The use of the PEER algorithm (please explain abbreviation!) should be explained and justified (see Xue et al. 2022 Biorxiv).

- Lines 102-106: The explanation could be moved to the Methods section.

- Lines 106: One "each" should be removed.

- Lines 123-124: The authors should explain how the association was established. This is not clear

from their description.

- Line 163 etc: This must read Fig. 1G.
- Line 196: The reference to Fig. 2C is unclear.
- Lines 232-243, 401-409: Large parts of these paragraph should be placed in the discussion.
- Line 489: The abbreviation BA46 should be explained.
- Fig. 1C: The small fleck located at 6 o clock / south of the larger endothelial cell island is not represented by the selected markers. Is this part inhabited by pericytes?
- Fig. S5F: Overlap of gene names renders them unreadable.

Referee #3 (Remarks to the Author):

The paper by McCarroll and colleagues represents a major contribution to our understanding of multicellular contributions to schizophrenia-associated pathophysiology. By using sophisticated analyses of healthy controls and schizophrenia postmortem PFC snRNAseq data, the authors set out to pinpoint cellular contributions. Perhaps as expected, neurons were implicated, but surprisingly astrocytes were too. Among these, a shared synaptic-neuron-astrocyte-program (SNAP) was identified, that existed even among neurotypical samples and was reduced in schizophrenia. This is the central finding from this study, along with its subsequent exploration. The existence of SNAP clearly implicates astrocytes and neurons in genetic vulnerability to schizophrenia. This is an important and conceptually deep contribution, as it is emerging that astrocytes contribute to psychiatric phenotypes in causal ways.

The paper is beautifully written and the figures are clear. The discovery of SNAP is significant and markedly enriches our understanding of schizophrenia mechanisms. The detailed analyses of SNAPa reveals aspects of astrocyte biology and its intersection with psychiatric disease that are new and compelling. Overall, this study will likely have a major impact and garner interest from diverse readership across the board. The authors should consider the following suggestions.

Suggestions

The section near lines 250-264 is important as it shows that astrocytes in schizophrenia do not show defined polarized states as defined by several suggested astrocyte reactivity markers. On line 263 the authors cite a review on this topic, but since then two studies have used experimental and/or disease contexts to explore this issue and found that astrocytes display context-specific transcriptional responses beyond simple polarized states, which I believe is in accord with the suggestions made in the current study. These two papers are: PMID: 35614216, PMID: 33086039. The authors may consider citing them appropriately near line 263.

In fig 2g, astrocytic genes involved in cholesterol biosynthesis are highlighted. This is remarkable, as the same genes are also now strongly implicated in several neurodegenerative diseases (AD, HD for example). It may be worth mentioning this as it suggests possible convergence of astrocytic mechanisms.

The reference to Perea et al's review on tripartite synapses near line 423 needs to be thoughtfully considered. First, it remains unclear if tripartite synapses comprising pre, post and astrocytic elements are the norm. The latest electron microscopy data suggest that many synapses may not include an astrocyte element and even those that do have variable placement of astrocyte processes relative to the synapse. Therefore, the term tripartite synapse has no anatomical meaning and should be avoided (it's not at all as definitive as bipartite synapse). Instead of referring to tripartite synapses, the authors could simply state ".....SNAP involves the biology of astrocyte-synapse functional and spatial interactions...". Then, they could reference a more compelling review such as pubmed ID 29096081. Furthermore, the subsequent discussion of cell adhesion molecules, synaptic biogenesis cues, neurotransmitter uptake and cholesterol biosynthesis that SNAPa beautifully identified were never part of the "tripartite synapse" idea, which focused on astrocytes being active participants of fast synaptic signaling by means of release of their own neurotransmitters. This idea has not held true and the term tripartite synapse is a loaded and highly controversial term (with little meaning). By using it, the authors hinder the broad appeal of their findings that in fact are consistent with much recent biology on how astrocytes do regulate synapses in a proven and measurable way. They may consider citing the following reviews instead: PMID: 30309945. PMID: 29096081

Minor comment: I like the acronym of SNAP. However, I wonder if this will cause superficial confusion with SNAP proteins involved in the vesicle cycle? Perhaps consider neuron-astrocyte-synaptic program (NASP) or neuron-astrocyte-program (NAP) to make it separable. This is a minor suggestion.

Minor comment: line 213. Change "180 thousand" to "180,000"

Author Rebuttals to Initial Comments:

Neurons and astrocytes coordinate a synaptic gene expression program that declines in schizophrenia and with advancing age

(Manuscript # 2022-12-19313A)

Responses to reviewer comments, suggestions, and ideas

Table of contents

Responses to Reviewer 1 comments, suggestions, and ideas.....	1
Responses to Reviewer 2 comments, suggestions, and ideas.....	5
Responses to Reviewer 3 comments, suggestions, and ideas.....	26
References.....	29

Responses to Reviewer 1 comments, suggestions, and ideas

The authors present a compelling analysis of single cell transcriptome data collected from a sizable number of human donors with and without schizophrenia diagnoses. The consideration of data from distinct cell types here allows for the discovery of complementary cellular activities that are coordinated in their expression across different cell types, including gene expression changes in neurons and astrocytes pertaining to synaptic functions (SNAP). These SNAP genes are differentially expressed in schizophrenia, suggesting a specific domain of altered cellular activities in schizophrenia cortex.

This paper makes an especially strong case for the field to critically reconsider the frequently used shorthand assumption that highest gene expression indicates highest importance in specific contexts. Also, the application of latent factor analysis here provides a method for other single cell studies for moving beyond consideration of genes most abundantly and specifically expressed in a certain cell type (cell type marker genes) to identify additional gene sets with altered activity in particular cell types, which could be linked to disease biology.

We are grateful for this assessment of the work.

The authors additionally use these data to contextualize known schizophrenia genetic risk factors, including defining the strong astrocytic contribution of NRXN1 and C4 expression, and the identification of cell type-specific eQTL colocalization with schizophrenia GWAS loci.

I enjoyed the opportunity to review this manuscript and to get a firsthand look at the new ideas presented here. The logical flow of analyses and findings are clearly presented in both the text and figures.

We are very grateful for this assessment and these thoughts.

A few comments and questions:

(1) Characterization of cell type-specific eQTLs is a sensible analysis to run on these data, though these results feel somewhat separate from the primary focus on SNAP.

We appreciate this perspective – expressed even more forcefully by reviewer 2 – that the eQTL results felt somewhat separate from the paper’s primary focus on SNAP. Reflecting on these comments, on Reviewer 2’s additional comment that the manuscript seemed uncertain about whether it was primarily about SNAP (vs. about schizophrenia genomics in general), and on our

hope to add new findings of age-associated changes in SNAP to a manuscript that (at 5,000 words) was quite long, we decided to take these suggestions and remove the eQTL analysis so that SNAP is the clear focus. The implication of astrocytes (in addition to neurons) as a setting for genetic effects in schizophrenia is made more strongly and directly anyway by the various genetic analyses which are now in the various panels of **Fig. 4**.

This may make the below analytical suggestion moot, but we still appreciated it, followed up, and wanted to reply:

I encourage the authors to more directly consider how one might jointly interpret cell type-specific eQTL colocalization and SNAP's relationship with schizophrenia genetic risk, in Discussion or elsewhere. A majority (but not all) of schizophrenia-colocalized eQTLs are associated with neuronal expression changes (Fig 4F), while SNAP-a and SNAP-n expression correlate with subsets of schizophrenia risk genes' expression in the corresponding cell type (Fig 3F, S17F). Is the take home message simply that both neurons and astrocytes perform cellular activities that are impacted by schizophrenia risk variants?

Yes, that is the take-home message of the analysis finding concentrations of schizophrenia risk loci in SNAP-a and SNAP-n (now **Fig. 4A**, previously Fig. 3F). Modest additional confirmation of this came from the colocalization of neuronal and astrocytic eQTLs with schizophrenia risk alleles (Fig. 4F in the initial submission). An additional complementary finding is that schizophrenia polygenic risk associates with the extent to which SNAP declines in individuals affected with schizophrenia (**Fig. 4B** in the revised manuscript). The strongest evidence is the finding (**Fig. 4A and Supplementary Note**) that common and rare variation that contribute to schizophrenia risk are strongly concentrated in SNAP-a and SNAP-n genes (the astrocytic and neuronal activities regulated by SNAP).

And/or, a simple way to directly link these analyses could be to note (e.g. in Fig 4F) each colocalized gene's SNAP-a and SNAP-n factor loading. Are neuron-specific colocalized eQTL genes more strongly loaded on SNAP-n, and astrocyte-specific eQTL genes on SNAP-a?

We appreciate this effort to help us think about analytical directions with the potential to make the eQTL analysis more connected to SNAP, and we did try these analyses, but the results were underwhelming – mainly because of small numbers, since only about 10% of schizophrenia GWAS hits resolved to eQTLs at all (as also observed for a large set of disease phenotypes by (Yao et al. 2020)).

(2) Lines 120-121 note that five factors point to coordinated changes across cell types, though it is not immediately apparent in Fig 2B which five factors those are. To clarify, list factor names in text, or add notation to Fig 2B.

Thank you for pointing this out. We have revised the main text to state explicitly in **lines 114-115**, "... with five of the factors (4, 5, 6, 7, and 10) pointing to strong changes that spanned multiple cell types (**Fig. 1G**)."

(3) Though there was no difference in the relative abundance of astrocyte subtypes in schizophrenia, is SNAP-a/cNMF2 more strongly driven by one particular subtype? Fig 3D would seem to suggest that expression is predominantly coming from protoplasmic astrocytes, though this could also be an effect of their high abundance. Does this gene set also have reduced expression in interlaminar and fibrous astrocytes from schizophrenia donor samples?

Thank you for pointing this out. This is indeed at least partly due to the astrocyte subtypes' relative abundances (and consequent statistical power). Among donors with schizophrenia, we observe reduced SNAP-a expression in all three astrocyte subtypes (this is now clarified in **Fig. S18I**), just with a stronger p-value for protoplasmic astrocytes ($p = 8.6 \times 10^{-7}$, two-sided Wilcoxon rank-sum test) compared to the fibrous and interlaminar astrocytes ($p = 7.2 \times 10^{-4}$ and $p = 1.2 \times 10^{-3}$ respectively).

While all three astrocyte subtypes exhibit reduced overall SNAP-a expression in persons with schizophrenia, specific details of SNAP-a and its relationship to schizophrenia appear to be more pronounced in protoplasmic astrocytes. For example, protoplasmic astrocytes exhibited strong case-control differences in expression of synaptic receptors and transporters ($p = 2.3 \times 10^{-6}$, two-sided Wilcoxon rank-sum test) while fibrous and interlaminar astrocytes (which tend to have less contact with synapses) did not ($p > 0.01$) (**Fig. S21C**). On the other hand, case-control expression differences for cholesterol-biosynthesis genes were shared across protoplasmic, fibrous and interlaminar astrocytes (**Fig. S21D**).

(4) This may be more common knowledge than I am aware, but a quick definition of "regulon" ~line 266 would be helpful.

Thank you for pointing this out. We have revised this to read at **lines 300-301**, "To find patterns of transcription factor activities (regulons) that might underlie this inter-individual variation, we used pySCENIC (Van de Sande et al. 2020) (**Fig. 3K, Fig. S21B, and Fig. S23C**)."

(5) If there is room, I would suggest adding a quick sentence after line 280 to note that different regulons are associated with SNAP-n than with SNAP-a. It is interesting that the expression of these gene sets are likely driven by different primary regulators.

We agree, this is a good point to make explicitly. We have revised this to read at **lines 302-304**, "Expression of SNAP associated strongly with the expression levels and predicted targets of

many transcription factors, and appeared to engage distinct pathways in astrocytes and neurons”.

(6) The comparison of eQTL cell type vs. brain region specificity is an interesting one (Fig 4C), and the choice to avoid highlighting effect size differences in the main figures is logical. However, comparing p-values is also potentially tricky, given the influence of sample size on p-value. Sample size is not perfectly matched across this study and GTEx, nor between GTEx brain regions. The brain regions with the most depressed p-values across the 7733 genes displayed are also the regions with the fewest genotyped samples in GTEx (spinal cord c-1, substantia nigra, amygdala, N=114-129 as listed in current GTEx portal). Perhaps comparing effect size estimates that have been normalized across all genes tested in each data set (beyond the 7733 displayed) would more fairly present this point?

We completely agree with your point that p-values are also strongly shaped by sample size and power. As discussed above, we have removed the eQTL analysis, since your above comment suggested (and Reviewer 2 expressed strongly) that it distracted from the paper’s focus on SNAP, and since we hoped to add new findings of age-associated changes in SNAP to a manuscript that (at 5,000 words) was already quite long.

(7) I could not find this information in figure legends – how has column and/or row order been determined in Fig 1D, E, F (donors), Fig 3F (gene order in rows), Fig 4C (genes in rows, cell types or regions in columns)?

Thank you for pointing out the need to clarify this in the figure legends. The donors in the barplots in the earlier Fig. 1, D, E, F (now **Fig. S3C, Fig. S5A, and Fig. S7A**) are ordered by sample IDs (i.e. in a pseudorandom order that had no particular relationship to phenotypes) (**lines 638-639, 661-662, and 680-681**); the genes in the earlier Fig. 3F (now **Fig. 3K**) are listed in decreasing order by their loadings onto SNAP-a (**lines 571-572**). The earlier Fig. 4C has been removed for reasons discussed above.

(8) Note, paragraph at lines 192-199 references Fig 2H, should be 2I.

Thank you, we have corrected the figure call-out in the text (now **Fig. 4B, lines 359-367**).

Responses to Reviewer 2 comments, suggestions, and ideas

Ling and colleagues explore the inter-individual variability of transcript levels in human brain cells as a new approach to detect genes that are regulated in tandem across cell types thus implying "transcellular" functions. In a first part, the authors identify groups of genes expressed by neurons and – importantly – astrocytes (named Synaptic Neuron Astrocyte Program) that show correlated transcription across neurotypical donors and reduced expression in schizophrenia patients. This "program" comprises structural components of synapses, enzymes mediating cholesterol biosynthesis and genes that have previously been identified as risk factors for schizophrenia. Notably, the results implicate that two of these established risk factors, NRXN1 and C4, are expressed by astrocytes. In a second smaller part, the authors use the same dataset to reveal the impact of expression quantitative trait loci on transcript levels of selected genes in different cell types.

The authors' approach to learn about cell-specific contributions to brain function and disease based on correlated cell-specific transcriptomic variation is – to this referee's knowledge – new and highly original. The authors' work suggests that "control groups" – ubiquitous in biomedical research – represent a treasure trove.

This is of extraordinary interest and could mark a milestone opening new perspectives to reveal cell-cell interactions in the brain and in other organs/tissues in healthy and sick people based on "natural" inter-individual variation.

We so appreciate this assessment of the work and these thoughts about its implications.

There is a substantial body of work exploring inter-individual variation in biological systems and the authors should refer to relevant articles (for example, impact of biological noise reviewed by Eling et al., 2019 Nat Rev Genetics; brain-specific: Makowski et al., 2022 Science).

Thank you for pointing us to these references. We have added citations to these (lines 56-58) and other papers in the introduction as examples of how inter-individual variation reveals ways in which biology fluctuates across contexts.

The methods and the display of data are state-of-the-art except for some specific points outlined below. Notably, the authors deliver an extremely precious – and possibly expandable – dataset comprising an impressive number of donors of either sex covering a wide range of ages.

We are grateful for this evaluation, and below seek to address the specific points that were exceptions to this.

There are two general points of criticism with respect to data analysis. The authors' approach throughout the ms to compare neurotypical controls and patients without proper stratification based on sex and age is hard to accept given the impact of these factors on brain cells (for the latest with respect to age see e.g. Allen et al., Cell 2023) and possibly confounding effects on the authors' analyses. Looking naively at the large number of graphs comparing neurotypical and diseased donors one gets the impression that for most measures, there is substantial overlap of the values between these groups rather than "real" differences. Curiously, the authors show a sex-related difference (Fig. S6B), which they do not comment on in the text. And they mention the lack of age ("lifetime")-related differences (line 128) without showing the results. Analysis of results using proper stratification appears as indispensable.

We appreciate these thoughts, which led to interesting new analyses and results that we have incorporated into the revised manuscript. Incorporating age into the analyses proved especially interesting and expanded the scope of the results: We find that SNAP also dims with advancing age, which we now incorporate into many analyses in the manuscript, both as a covariate in the schizophrenia analyses (that indeed explains much of the earlier overlap in SNAP expression between affected and unaffected donors) and as an important relationship in its own right.

(Also: thank you for pointing out the ambiguity in our statement, "Expression of LF4 also did not associate with lifetime or recent use of antipsychotic medications", in which "lifetime" could be read as suggesting lifespan and thus suggesting a lack of age-related variation. We have revised this to "long-term or recent use of antipsychotic medications" in **line 318**.)

Advancing age has long been known to involve many changes that are also the cognitive symptoms of schizophrenia, including declines in cognitive flexibility, reasoning and problem solving. (As you likely know, although schizophrenia is sometimes equated with psychosis in the wider culture, psychosis is just an occasional, relapsing/remitting symptom of schizophrenia, a symptom that is usually treated successfully with medications; the cognitive losses in schizophrenia are its earliest and continuous feature, have commenced before the onset of psychosis, and are not treated successfully with today's medications.) Brain aging and schizophrenia also share longstanding observations of reduction in cortical volume, reduction in neuropil, and reduced numbers of synapses. Neuropsychologic and neuroimaging studies have long noted parallels between the changes observed in schizophrenia and those observed with advancing age (nicely reviewed in the introduction to (Adamowicz and Lee 2023); also (Constantinides et al. 2023; Dietsche, Kircher, and Falkenberg 2017; Head et al. 2008; Fucetola et al. 2000; Koutsouleris et al. 2014)). Schizophrenia greatly increases the risk of dementia later in life (Stroup et al. 2021; Adamowicz and Lee 2023; Ribe et al. 2015; Cai and Huang 2018; Lin et al. 2018), and inherited genetic risk for schizophrenia associates with decreased measures of cognition in older individuals (Liebers et al. 2016).

In the new analyses, we find that expression of SNAP is diminished with advancing age, including declines in both its neuronal and astrocytic components. The effect of age is clear in separate analyses of controls-only and schizophrenia-only subsets of the cohort, indicating that the effect of age is independent of the effect of schizophrenia (**Fig. 1I**). The effect of schizophrenia is in no way due to the effect of age: adjusting donors' SNAP expression levels for age actually increases the association of these measurements with schizophrenia (SNAP: $p = 9.9 \times 10^{-6}$, SNAP-a: $p = 2.5 \times 10^{-8}$, SNAP-n: $p = 4.6 \times 10^{-9}$), by reducing variation within the affected and unaffected groups (**Fig. 1J, Fig. 3E, and Fig. 3J**). Simultaneous linear regression of the donors' SNAP expression levels against age, sex and schizophrenia status confirms the separate effects of age ($p = 3.4 \times 10^{-4}$) and schizophrenia ($p = 2.1 \times 10^{-5}$) and detects no effect of sex ($p = 0.40$) (**Table S3**).

Unlike the strong effect of age, the same analyses did not find any effect of sex (**Fig. S9, D and E; Table S3**). Though some genes (especially on the sex chromosomes) differed in expression between males and females, these differences were orthogonal to the changes in SNAP. The association of schizophrenia with reduced SNAP expression was apparent in both sexes separately. (Also, to clarify: we had not meant for the supplementary figure in the initial submission to suggest that there was a sex difference in the relationship of schizophrenia to SNAP. The association of SNAP expression with case-control status has a somewhat stronger p-value in males, but this appears to reflect statistical power due to the difference in sample size between these groups ($n = 70$ females and 110 males) rather than a sex difference in the effect of schizophrenia upon SNAP. We have clarified this in the revised supplementary figure legend (**Fig. S9B and lines 712-715**) and additional analyses (**Fig. S18H**).)

We are grateful for the suggestion of this analytical direction, which we feel enhanced the work and the paper.

A second fundamental issue that admittedly applies to many single-cell transcriptome studies is whether the numbers of nuclei collected from individual brains – although truly impressive – can be considered representative of the cell counts in the specific region of interest. This is all the more important as the authors seek to delineate insight from inter-individual and inter-cellular variation. Can the authors estimate, which percentage of cells of each type in the prefrontal cortex they have analysed compared to the "real" number of cells per individual? Can the authors comment on the errors induced by this systematic "undersampling"?

In the revised manuscript, we clarify something that was perhaps unclear in the initial main text: that the analyses use single-nucleus rather than single-cell RNA-seq. (We now say, in **lines 82-84** in the first Results paragraph, "We used single-nucleus rather than single-cell RNA-seq to avoid effects of cell morphology upon ascertainment, and because nuclear membranes (but not plasma membranes) remain intact in frozen post-mortem tissue.")

In early single-cell RNA-seq studies, in which we and others dissociated and analyzed entire cell soma from the mouse brain, we detected (by comparison to stereological data) significant distortions in representation related to cell morphology, since cells needed to endure dissociation with their cell-soma membrane intact and resealed, and the ability of cells to do this differed by cell type and morphology (Saunders et al. 2018). However, the current work utilizes single-nucleus rather than cell single-cell analysis. Single-nucleus analysis largely eliminates such distortion, since nuclei are more uniform in morphology and in those characteristics (such as density) used to isolate and purify nuclei from brain tissue. Stereology analysis indicates that the cellular proportions quantified by snRNA-seq (but not those quantified by scRNA-seq) match those estimated by smFISH ((Bakken et al. 2018) and (Krienen et al. 2020), Extended Data Fig. 1).

You are of course right that single-cell/single-nucleus studies only *sample* nuclei rather than analyzing all of the nuclei. (This is particularly true in the human brain, which is large – isotropic fractionator experiments corroborated by stereology have yielded estimates of 1.3 billion neuronal cells in human prefrontal cortex (Gabi et al. 2016; Herculano-Houzel 2009) and similar numbers of non-neuronal cells (Azevedo et al. 2009; Bahney and von Bartheld 2014); based on these numbers, we estimate that our dataset has sampled 0.05% of cells in this region, with similar ratios of neuronal to non-neuronal cells.) The key question is whether the sampling is random and unbiased – i.e. whether cell types and cell states are sampled in proportion to their abundances *in vivo*. (Our results relate not to absolute numbers of cells, but certain analyses assume that we are ascertaining cell states in proportion to their abundances in the brain.) Here, yes, it is essential that the work used single-nucleus rather than single-cell RNA-seq – especially since SNAP-a may relate to variation in astrocyte morphological characteristics – and the earlier work comparing snRNA-seq-based estimates to estimates from stereology (e.g. in (Bakken et al. 2018) and (Krienen et al. 2020)) is encouraging in this regard.

Unfortunately, the manuscript presents severe weaknesses with respect to clarity and context that undermine its value, notably in the introductory paragraphs, where the overall goals of the study are not clearly stated and too little background is given.

We appreciate this advice on the introductory paragraphs, which we took to heart in rewriting them. The earlier manuscript had (as you recognized above) some ambiguity about whether its primary focus was SNAP, or broader genomics observations about schizophrenia. Reflecting on your comments, and on the new results about the decline of SNAP with advancing age, we have edited the manuscript to emphasize a primary focus on SNAP, with the relationship of SNAP to schizophrenia (and now also to advancing age) illustrating the biological significance of SNAP.

In particular,

The abstract and introduction contain vague and diverging statements such as "to better understand neurobiological variation" (lines 25) or "we explored the idea that inter-individual variation could reveal regulatory coordination at a tissue level" (lines 49-50) or "understand how human neurobiology varies" (line 57).

We have replaced these with more concrete and focused statements.

The background information is too "selective" and seems to anticipate the results rather than to inform the reader about the state of the art.

We removed these background statements that seemed to anticipate the specific results in a selective way.

Again, statements are rather vague (e.g. lines 50-51 "the tendency of many gene-expression changes, perhaps in multiple cell types to be implemented together in the same individuals and contexts") or lack proper reference to previous work (line 48).

We removed the vague introductory statements, and supplied a review-article reference for the observation that almost all aspects of biology exhibit quantitative variation across individuals.

Importantly, the relevance of schizophrenia for this work remains unclear. Is the disease the main topic of the study? The authors must formulate concisely the goals of their work and expose the relevant background in sufficient detail.

We appreciate this comment's surfacing the ambiguity (in the earlier manuscript) about whether the primary focus was SNAP, or schizophrenia. We have revised so as to clearly focus on SNAP, with the application to schizophrenia more illustrating the biological significance of SNAP rather than being the driver of the questions or approaches.

The revised introduction in **lines 44-73** reads,

Cells collaborate to perform key functions in the brain and other tissues, working together to construct and regulate multicellular structures such as synaptic networks. The transcriptional programs that are coordinated in a non-autonomous manner by cells of distinct types are, for the most part, not yet known. While single-cell and single-nucleus RNA-seq are now routinely used to describe cell-type-specific gene expression (Macosko et al. 2015; Slovin et al. 2021), less is known about how gene expression arises from specific transcriptional programs, nor whether such programs are coordinated by cells of multiple types.

In free-living species – in which individuals have diverse genetic inheritances, environments and life histories – almost all aspects of biology exhibit quantitative variation (Mackay, Stone, and Ayroles 2009). Natural variation makes it possible to

observe a biological system across many contexts and potentially learn principles that govern its dynamics (Makowski et al. 2022; Eling, Morgan, and Marioni 2019). Across such contexts, correlation structure in high-volume measurements can reveal aspects of the system's underlying architecture: for example, in gene expression studies, if the expression levels of a set of genes consistently vary together, it implies that their expression is biologically coordinated (Brown and Botstein 1999).

Here we sought to generalize this idea to recognize coordinated changes across multiple cell types in the brain and how such changes may relate to interindividual differences in health and in disease. We analyzed the dorsolateral prefrontal cortex (dlPFC, Brodmann area 46) from 191 human brain donors (ages 22 to 97) by single-nucleus RNA-seq (snRNA-seq) and developed a computational approach, based on latent factor analysis, to recognize multicellular gene-expression patterns in such data. Tissue-level programs could provide new ways to understand both healthy brain function and genetically complex brain disorders, since disease processes likely act through endogenous pathways in cells and tissues. A longstanding challenge in complex brain disorders is to identify specific aspects of brain biology on which many genetic effects converge. Here we applied this idea to try to better understand schizophrenia.

We appreciate your advice, which we think has improved the introduction and better frames the approaches and results.

The conclusions stated by the authors are not fully justified by the data. Obviously, a major point of concern that again applies to many publications in the field is whether results that are purely based on however elaborate bioinformatic analyses have biologic meaning. Evidently, one wishes to have independent validation or lines of evidence that – for example – SNAP is "real", that it is important for brain function and that it breaks down in the prefrontal cortex of schizophrenic patients. Obviously, this is a challenge for everybody working in the field. Still, a step in this direction may be to show, at least for a defined set of genes contained in SNAP, whether transcript and protein levels are correlated in human brain cells. A second approach could be to explore in more detail whether SNAP components are in fact controlled by well-known transcriptional programs. The authors provide first evidence, but this could be expanded. Finally, the authors could analyse age- and sex-dependent changes and test whether they reveal "ground truth" impacts of these factors that have been established previously by other approaches. This could provide a sort of internal validation of their transcript analyses.

We appreciate your suggestions of three potential ways to increase certainty that SNAP is connected to other "ground truth" phenomena – such as age and sex, transcriptional programs, and protein expression. We pursued all three of these; all three provided supporting evidence which are now included in the revised manuscript. Particularly impactful was your suggestion of

looking for age-dependent changes as “ground truth”: as described above, this has led to very interesting new results in the manuscript. We also found supporting evidence along the additional lines suggested (protein expression and well-known transcriptional programs).

Age-associated changes in SNAP

As described above, we find that expression of SNAP declines with advancing age, including parallel declines in its neuronal and astrocytic components. The effect of age is clear among unaffected individuals, as well as among individuals affected with schizophrenia, even when the two groups are analyzed separately (**Fig. 1I**). The effect of schizophrenia is in no way due to confounding by age: adjusting donors’ SNAP expression levels for age actually increases the association of these measurements with schizophrenia (SNAP: $p = 9.9 \times 10^{-6}$, SNAP-a: $p = 2.5 \times 10^{-8}$, SNAP-n: $p = 4.6 \times 10^{-9}$), by reducing variation within the case and control groups (**Fig. 1J, Fig. 3E, and Fig. 3J**). Simultaneous linear regression of the donors’ SNAP expression levels against age, sex and schizophrenia status confirms the separate effects of age ($p = 3.4 \times 10^{-4}$) and schizophrenia ($p = 2.1 \times 10^{-5}$) and detects no effect of sex ($p = 0.40$) (**Table S3**). The strong relationship of SNAP expression to donors’ chronological age adds to the evidence that SNAP is “real” and also that it is not caused simply by the life circumstances or medications associated with schizophrenia.

SNAP components and well-known transcriptional programs

To analyze the relationship of SNAP to expression levels of transcription factors and their computationally predicted targets, we used a computational approach (pySCENIC (Van de Sande et al. 2020)) that has become widely used in single-cell genomics for such analyses. **Fig. 3K** now includes the strongest pySCENIC results, including five transcription-factor activities predicted to be recruited by SNAP in astrocytes and three recruited by SNAP in neurons. Notably, the transcription factor activities recruited in neurons and astrocytes are different from each other.

At the same time, we realize that these are still largely computational predictions of transcription-factor activities (albeit from a computational approach that was developed independently of our work). So, in the revised manuscript, we also include more-detailed analyses of two well-known transcriptional programs for which the transcription-factor *targets* are also well-known and established by *other kinds of evidence* (biochemical etc.) in earlier, independent work. These are the SREBP-regulated transcriptional pathway (Horton et al. 2003; Eberlé et al. 2004; Shimano and Sato 2017) in astrocytes, and the JUNB (AP-1)-regulated (late-response) transcriptional pathway (Yap and Greenberg 2018; Malik et al. 2014; Yap et al. 2021) in neurons. As described in the new **Fig. S25**, expression of SNAP associates strongly with both the RNA expression of SREBF1 (in astrocytes) and JUNB (in neurons), and with the expression of their well-established transcriptional targets in the same cell types. These relationships exhibit strong cell-type specificity. (The reduced neuronal expression of JUNB

targets among neurons in affected individuals is also consistent with longstanding observations of reduced levels of neuronal activity in the PFC (a phenomenon termed “hypofrontality”) in schizophrenia (Callicott 2003.)

On SNAP components and well-known transcriptional programs, the manuscript now reads in **lines 300-312**,

To find patterns of transcription factor activities (regulons) that might underlie this inter-individual variation, we used pySCENIC (Van de Sande et al. 2020) (**Fig. 3K, Fig. S21B, and Fig. S23C**). Expression of SNAP associated strongly with the expression levels and predicted targets of many transcription factors, and appeared to engage distinct pathways in astrocytes and neurons: for example, inducing expression of SREBF1 and its well-known transcriptional targets (Horton et al. 2003; Eberlé et al. 2004; Shimano and Sato 2017) in astrocytes, and JUNB (AP-1) and its well-known transcriptional targets (Yap and Greenberg 2018; Malik et al. 2014; Yap et al. 2021) in neurons (**Fig. S25**). (The latter may reflect average neuronal activity levels in the PFC, which have been found by neuroimaging to decline (“hypofrontality”) in schizophrenia (Callicott 2003).) Also among the regulons whose activities associated most strongly with SNAP-a expression in astrocytes were a RORB regulon (under-expressed in SNAP-low donors) and a KLF6 regulon (over-expressed) (**Fig. 3K and Fig. S21B**); common genetic variation at *RORB* and *KLF6* associates with schizophrenia (Trubetskoy et al. 2022).

Expression of SNAP components at a protein level

We also sought evidence that SNAP manifests in the proteins that can be sampled from cerebrospinal fluid (CSF). We analyzed available data from a mass-spectrometry proteomics analysis of cerebrospinal fluid (CSF) from 22 healthy human donors (Karayel et al. 2022), performing a latent factor analysis that is conceptually analogous to our analysis (in **Fig. 1F**) of cell-type-specific RNA-expression measurements in the brain donors (but of an independent data set, derived from a distinct set of donors). The top latent factor in analysis of the CSF proteomics data (explaining >15% of inter-individual variation in CSF protein measurements) bore a strong resemblance to SNAP, with a highly significant correlation between the SNAP/LF4 gene loadings and the protein-expression loadings for the corresponding proteins in LF1 of the CSF analysis (Spearman’s $\rho = 0.32$, $p < 2.2 \times 10^{-16}$; **Fig. S11A**, reproduced below). For the 134 proteins (top decile) most strongly recruited by this CSF-protein latent factor, the corresponding genes tended to be positively regulated by SNAP ($p = 2.1 \times 10^{-28}$, two-sided Wilcoxon rank-sum test), and 47% were among the 10% most strongly regulated by SNAP (**Fig. S11C**, reproduced below). Genes/proteins driving this relationship included genes that are principally expressed by neurons (including *GAP43*, *NPTXR*, *NPTX1*, and *NPTX2*) as well as genes that are principally expressed by astrocytes (*APOE*, *FGFR3*, and *NTM*). Like SNAP, this CSF latent factor also exhibited diminished expression with advancing age in the independent data set

(Fig. S11B, reproduced below). Consistent with the reduction of SNAP in schizophrenia, one of these proteins (NPTX2) is known to exhibit reduced CSF levels in schizophrenia (Göverti et al. 2022; Xiao et al. 2021). These results are now included as Fig. S11, which we also reproduce below.

Latent factor analysis of cerebrospinal fluid (CSF) proteomics data from different individuals identifies a factor resembling SNAP.

Reproduced from Fig. S11 of the revised manuscript.

(A) Relationship of SNAP gene loadings to the top latent factor in an analysis of inter-individual variation in CSF protein levels (CSF LF1) using quantitative protein abundance measurements from (Karayel et al. 2022) ($n = 1,341$ genes/proteins shared between both analyses). Shaded region around the regression line represents 95% confidence interval.

(B) Relationship of CSF LF1 donor scores to age ($n = 22$ donors). Shaded region around the regression line represents 95% confidence interval.

(C) Density plot showing distribution of SNAP gene loadings for (black) all genes and genes encoding proteins that are strongly recruited (top decile) by (blue) CSF LF1. Distributions were found to be different by Wilcoxon test ($p = 2.1 \times 10^{-28}$, two-sided Wilcoxon rank-sum test).

Apart from these fundamental points, the authors may consider the following comments and suggestions:

- A general concern with the ms that also undermines its value is the combination of two stories that are based on the same dataset, but obviously unrelated. The results presented in lines 315-409 analyse cell type-specific gene expression using eQTL following up on the recent Yazar and Perez papers. Obviously, this is a very important and exciting aspect, but this part resembles more a last-minute addition than an integral component of the SNAP story. The authors should consider to publish the results from the two studies separately.

We agree that this section distracted from the paper's primary focus on SNAP, and have removed it as suggested. Doing so also makes room for the new results on age-associated changes in SNAP, which are more novel and exciting and more organically connected to the paper's primary focus on SNAP. (The eQTL results had provided modest additional evidence corroborating the finding here that astrocytes as well as neurons contribute effects of common genetic variants in schizophrenia – but this point is made with more direct and SNAP-connected evidence in **Fig. 4, A to D** of the revised manuscript, and the more-modest eQTL results can be published separately as you suggest, as additional confirmation of the contribution of astrocytes to the disorder.)

- Research over the last decade has revealed that the term "astrocyte" comprises a heterogeneous group of cells. Latest evidence comes for example from Su et al. 2022 Cell Stem Cell focusing on the hippocampus. This notion should alert the authors to explore in more detail (and similar to what they did for neurons) subtypes of astrocytes. This applies for example to the abundances and impact of schizophrenia on astrocytes subtypes (lines 79-96), cell-type specificity of latent factors including LF4 (lines 119-140), biological processes (lines 162-172), and certainly the exploration of SNAP programs (lines 201-262) etc. It seems that the data shown in Fig. 3 and in corresponding supplementary figures stop short of a profound analysis of astrocytes given the current evidence for their heterogeneity. A more in-depth exploration may reveal that only specific astrocytes subtypes are involved in SNAP and that only these are affected by disease: they may downregulate cholesterol synthesis (or lipid metabolism) as claimed by the authors – or degenerate altogether.

We agree that this dataset presents an opportunity to examine astrocyte heterogeneity in greater depth, and we worked to increase the depth of such analysis in the revised manuscript. Some aspects of astrocyte heterogeneity described in earlier work appear to have been differences between astrocytes in different brain areas and are not captured in the current work,

as we focus on just one brain area (BA46, prefrontal cortex) in many people. Even within BA46, however, there is meaningful astrocyte heterogeneity which we now further describe.

Within BA46, our analyses readily recognized a high-level, categorical distinction among protoplasmic, interlaminar and fibrous astrocytes (**Fig. 3A**). Much of the remaining heterogeneity is more continuously distributed, both within and across donors – it is possible to chunk this variation up into “clusters”, as is often done in papers, but the clusters are arbitrary and may not represent enduring categories that will re-appear across many studies. Thus, we decided it was most useful to focus on the continuously varying factors. The revised manuscript includes (in the section titled, “Astrocyte gene-expression programs and SNAP”) both (i) more reporting of the other latent factors that shape quantitative variation in single-astrocyte gene expression, many of which affect very specific aspects of astrocyte biology; and (ii) a deeper analysis of SNAP in relation to aspects of astrocyte heterogeneity, such as morphology and reactivity, previously studied in mice.

On *BA46 astrocyte heterogeneity beyond SNAP*, the revised manuscript now reads in **lines 203-224**,

To better appreciate astrocyte variation and the astrocytic contribution to SNAP, we further analyzed the RNA-expression data from 179,764 individual astrocytes from the 180 donors. Analysis readily recognized a known, categorical distinction among the transcriptional profiles of three subtypes of adult cortical astrocytes (Hodge et al. 2019; Bakken et al. 2021): protoplasmic astrocytes, which populate the gray matter; fibrous astrocytes, which populate the underlying white matter; and interlaminar astrocytes (**Fig. 3A and Fig. S17, A to D**). Neither schizophrenia nor age associated with variation in the relative abundances of these astrocyte subtypes (**Fig. S17, E and F**).

We then identified latent factors that explained quantitative gene-expression variation among individual astrocytes within these astrocyte subtypes (**Fig. S18, A and B**). (We used consensus nonnegative matrix factorization (cNMF, (Kotliar et al. 2019)) because of its scalability to the 180,000-astrocyte data set.) The various latent factors appeared to capture diverse biological activities that exhibited quantitative variation among individual astrocytes, including translation (cNMF1); zinc and cadmium ion homeostasis (cNMF7); and inflammatory responses (cNMF8) (**Table S5**). One factor (cNMF2) corresponded to SNAP: its gene loadings and donor expression levels matched those of the astrocyte component of SNAP (gene loadings: Spearman’s $\rho = 0.68$, $p < 2.2 \times 10^{-16}$; donor expression levels: Spearman’s $\rho = 0.75$, $p < 2.2 \times 10^{-16}$; **Fig. S18, C and D and Table S6**). The identification of such similar sets of astrocyte gene-expression changes by distinct computational approaches – one based on inter-individual variation, the other based on single-cell variation – suggests these strong biological relationships are robust to the computational approach used (**Fig. S18, C to E and Fig. S19**).

On additional analyses of SNAP and its relationship to other aspects of astrocyte heterogeneity, the revised manuscript now reads in **lines 234-312**,

To better understand the biology supported by SNAP-a, we performed GSEA of its gene loadings. The strongest positive gene-set associations involved adhesion to synaptic membranes (GO:0099560) and intrinsic components of synaptic membranes (GO:0098889) (**Table S5**). The 20 genes most strongly associated with SNAP-a included eight genes with roles in adhesion of cells to synapses (*NRXN1*, *NTM*, *CTNND2*, *LSAMP*, *GPM6A*, *LRR4C*, *LRRTM4*, and *EPHB1*) (as established by earlier work including (Trotter et al. 2021; Hashimoto, Maekawa, and Miyata 2009; Turner et al. 2015; Hack et al. 1998; Formoso et al. 2016; León, Aparicio, and Scorticati 2021; Choi et al. 2019; de Wit et al. 2013; Siddiqui et al. 2013; Henderson and Dalva 2018) and reviewed in (Tan and Eroglu 2021; Saint-Martin and Goda 2022)) (**Fig. S20**). SNAP-a also strongly recruited genes encoding synaptic neurotransmitter reuptake transporters: *SLC1A2* and *SLC1A3* (encoding glutamate transporters EAAT1 and EAAT2) and *SLC6A1* and *SLC6A11* (encoding GABA transporters GAT1 and GAT3) were all among the 1% of genes most strongly induced by SNAP-a.

We sought to relate SNAP-a to an emerging appreciation of astrocyte heterogeneity and its basis in gene expression (Khakh and Deneen 2019). An earlier analysis of astrocyte molecular and morphological diversity in mice identified 62 gene-expression modules based on their co-expression relationships (Endo et al. 2022). Among these gene modules, the genes recruited by SNAP-a exhibited the strongest overlap ($p = 3.5 \times 10^{-4}$, $q = 0.015$) with the module that had correlated most closely with astrocyte morphological parameters related to the size of the territory covered by astrocyte processes (the “turquoise” module in (Endo et al. 2022), with overlap driven by genes including *Ezr* and *Ntm*) (**Table S5**). A potential interpretation is that SNAP-a supports these perisynaptic astrocytic processes (PAPs, reviewed in (Lawal, Ulloa Severino, and Eroglu 2022; Ghézali, Dallérac, and Rouach 2016; Aboufaires El Alaoui et al. 2020)). Further evidence for a role supporting PAPs was provided by the strong SNAP-a recruitment of the genes encoding synaptic neurotransmitter transporters (*SLC1A2*, *SLC1A3*, *SLC6A1* and *SLC6A11*) and synaptic-adhesion proteins (*NRXN1*, *NTM*, *CTNND2*, *LSAMP*, *GPM6A*, *LRR4C*, *LRRTM4*, and *EPHB1*).

The individual astrocytes sampled from any one donor exhibited quantitative variation in SNAP-a expression, but tended to have similar SNAP-a-expression levels relative to the much-broader spectrum of variation in SNAP-a expression across different donors (**Fig. S18, F and G**). As a result, the same donors tended to have higher-than-average (or lower-than-average) astrocytic expression of all of these genes together (**Fig. 3K and Fig. S21A**).

Earlier work has identified “reactive” astrocyte states induced by strong experimental perturbations and injuries (Liddelow et al. 2017). We found that more than half of the human orthologs of markers for these states were expressed at levels that correlated

negatively and in a graded manner with expression of SNAP-a (**Fig. S22**). Consistent with this observation, SNAP-a involved reduced expression of gene sets related to inflammatory responses (by GSEA, at false discovery rate (FDR) < 0.01), such as complement activation (GO:0006956) and response to type I interferons (GO:0034340) (**Table S5**). Reactive astrocytes have been described as polarized cell states. At the single-astrocyte level, expression of SNAP-a exhibited continuous, quantitative variation rather than discrete state shifts: a donor's location on the SNAP-a expression spectrum arose from a quantitative shift of their entire distribution of astrocytes, rather than a discrete shift by a subpopulation of astrocytes (**Fig. S18, F and G**). This suggests that these pathways are regulated by graded expression programs that exhibit quantitative variation across individual cells and among healthy control donors (**Fig. S18, F and G**), supporting the idea that astrocyte biological variation extends beyond polarized states (Burda et al. 2022; Yu et al. 2020; Escartin et al. 2021).

We also performed an analogous cNMF analysis on the nuclear RNA-expression profiles of 76,000 glutamatergic neurons, focusing on a single, abundant subtype (layer 5 IT-projecting neurons) so that the variation among individual cells would be driven primarily by dynamic cellular programs rather than by subtype identity (**Fig. 3F**). cNMF identified a latent factor that corresponded to the neuronal gene-expression changes in SNAP; we refer to this factor as SNAP-n (**Fig. 3, G to J and Table S7**). Like SNAP-a in astrocytes, average expression of SNAP-n was associated with age (Spearman's $\rho = -0.47$, $p = 2.1 \times 10^{-11}$) (**Fig. 3I**) and with schizophrenia (age-adjusted $p = 4.6 \times 10^{-9}$, two-sided Wilcoxon rank-sum test) (**Fig. 3J**). SNAP-n and SNAP-a associated with each other still more strongly (Spearman's $\rho = 0.80$, $p < 2.2 \times 10^{-16}$) (**Fig. S23A**), even in a controls-only, age-adjusted analysis (Spearman's $\rho = 0.61$, $p < 2.2 \times 10^{-16}$) (**Fig. S23B**), underscoring the apparently close coupling of gene-expression dynamics in neurons and astrocytes. Although GSEA of SNAP-n revealed associations to synaptic gene-sets, the specific genes driving these enrichments were distinct from those driving SNAP-a (**Fig. 3K, Fig. S24, and Table S8**).

To find patterns of transcription factor activities (regulons) that might underlie this inter-individual variation, we used pySCENIC (Van de Sande et al. 2020) (**Fig. 3K, Fig. S21B, and Fig. S23C**). Expression of SNAP associated strongly with the expression levels and predicted targets of many transcription factors, and appeared to engage distinct pathways in astrocytes and neurons: for example, inducing expression of SREBF1 and its well-known transcriptional targets (Horton et al. 2003; Eberlé et al. 2004; Shimano and Sato 2017) in astrocytes, and JUNB (AP-1) and its well-known transcriptional targets (Yap and Greenberg 2018; Malik et al. 2014; Yap et al. 2021) in neurons (**Fig. S25**). (The latter may reflect average neuronal activity levels in the PFC, which have been found by neuroimaging to decline ("hypofrontality") in schizophrenia (Callicott 2003).) Also among the regulons whose activities associated most strongly with SNAP-a expression in astrocytes were a RORB regulon (under-expressed in SNAP-low donors) and a KLF6 regulon (over-expressed) (**Fig. 3K and Fig. S21B**); common

genetic variation at *RORB* and *KLF6* associates with schizophrenia (Trubetsky et al. 2022).

With regard to relationship to schizophrenia, we find that protoplasmic, fibrous, and interlaminar astrocytes all exhibit the association of schizophrenia to SNAP-a, but that key details of this relationship vary across protoplasmic, fibrous, and interlaminar astrocytes. For example, while under-expression of lipid biosynthesis genes is shared across these subtypes, the relationship of schizophrenia to expression of neurotransmitter transporters and receptors arises much more clearly in protoplasmic astrocytes, which also express these genes much more strongly than other astrocytes do (**Fig. S21, C and D**, reproduced below).

Transcriptional investments in gene-sets across astrocyte subtypes.

Reproduced from **Fig. S21, C and D** of the revised manuscript. Transcriptional investments (by donor, separated by schizophrenia case-control status) in genes encoding synaptic transporters and receptors and cholesterol biosynthesis genes, in subtypes of astrocytes. Quantities plotted are the fraction of all UMIs in each subtype that are derived from these genes. P-values from a two-sided Wilcoxon rank-sum test comparing the affected to the unaffected donors are reported at the top of each panel. Box plots show interquartile ranges; whiskers extend to 1.5 times the interquartile

interval; central lines represent medians; notches represent confidence intervals around medians.

- As mentioned above, inter-individual variation has distinct sources including "noise" in the largest sense. A concrete question is whether the neuronal and astrocytic genes contained in the SNAP have particular genomic properties that "pretend" co-regulation although there is none. Such a particular feature could be – for example – the presence of TATA boxes and CpG islands that have been shown to increase and decrease transcriptional variability, respectively (Faure et al., 2017). Higher transcript variability may imply correlation just "by chance". The authors should consider to address these points to exclude structural genomic features as cause for apparent transcriptional co-regulation.

We agree that it is important to be alert to single-cell noise – including both biological noise (such as the stochastic biological fluctuations of individual cells) and statistical noise (from having only sampled each cell's RNA transcripts in the nucleus) – and the manner in which such noise would affect any analyses. In that regard:

Presence of these relationships at aggregated levels. Although the data are collected at single-cell resolution, many of our key analyses aggregate a donor's cells of the same type. (The main text in our initial manuscript did not sufficiently describe this.) An important feature of the analyses in **Fig. 1, G to J** (discovering SNAP) and **Fig. 2** (further analyzing SNAP) is that these analyses use a common approach (called "pseudobulking") to minimize the effects of biological and statistical noise. Each analysis is based on (per donor) on average 1,000 astrocyte nuclei (whose expression levels are summed together), an approach which we believe all but eliminates the single-cell-level biological "noise" that (Faure, Schmiedel, and Lehner 2017) observed in pluripotent stem cells, as that noise is single-cell in nature. (And similarly, the neuronal analyses in **Fig. 1, G to J** and **Fig. 2** are based on 100s to 1,000s of neuronal nuclei per donor.) To manifest in our analyses, hundreds of a donor's astrocytes (or neurons) would have to change together in a common way, which is the kind of effect we seek to detect, rather than the kind of single-cell fluctuations that concerned (Faure, Schmiedel, and Lehner 2017). On the other hand, the cNMF analysis of 180,000 individual astrocyte nuclei in **Fig. 3C** and 76,000 individual L5 IT glutamatergic neuron nuclei in **Fig. 3H** can indeed be affected by such single-cell fluctuations; here though we can be reassured by the similarity of the latent factor found, to the latent factor found in the earlier, "pseudobulk" analysis, as shown in **Fig. S18, C and D**.

Cell-type specificity. Another line of evidence that effects are not genomic in origin involves the cell-type specificity of the gene-expression changes in SNAP: though a gene has the same genomic features in different cell types, and though cDNAs from all cell types are processed together in the same experiment (distinguished only by short DNA barcodes that indicate single-cell-of-origin) and thus experience the same technical effects, the genes that are associated with SNAP in astrocytes are distinct from the genes that are associated with in SNAP in

neurons, even when expressed in both cell types (**Fig. 2 and Fig. 3K**). For example, SNAP and schizophrenia associate with astrocytic (but not neuronal) expression of cholesterol/fatty-acid biosynthesis genes (**Fig. S14**) and of genes such as *NRXN1* (**Fig. 4C**); none of these genes are known to utilize different promoters in astrocytes and neurons.

Relationships to donor age and schizophrenia (affected/unaffected) status. The statistically very strong relationship of SNAP to donor age and schizophrenia further enhances confidence that these relationships are not “by chance” nor due to technical effects of genomic features, since such effects would not correlate with donor age or clinical histories.

Considering transcriptional variability and cell-type-specific super-enhancers. As noted in (Faure, Schmiedel, and Lehner 2017), target genes of super-enhancers – enhancers with the highest density of transcription factors and active histone modifications – exhibit high and correlated expression variability within individual cells. Here a key thing to note (as mentioned above) is that our initial discovery of SNAP (**Fig. 1**) and further analyses of SNAP (**Fig. 2**) are not based on single-cell-level fluctuations, but on “pseudobulk” meta-cell analyses in which each donor’s astrocytes (on average about 1,000 astrocytes per donor) are considered as a group – to average away the kinds of single-cell-level fluctuations that (Faure, Schmiedel, and Lehner 2017) identified.

To further consider super-enhancers, and because the relationship of SNAP to super-enhancers is also of interest as a question in its own right, we also asked whether SNAP genes are enriched for targets of super-enhancers. Using bulk H3K27ac HiChIP and scATAC-seq data from a recently published study (Corces et al. 2020), we identified super-enhancers in human cortical astrocytes and determined whether the promoters of SNAP-recruited genes would be predicted to contact these super-enhancers. We found that genes whose promoters are predicted to contact super-enhancers in astrocytes actually tended to have weaker loadings onto SNAP-a ($p < 2.2 \times 10^{-16}$, two-sided Wilcoxon rank-sum test of SNAP-a gene loadings between super-enhancer-associated and -non-associated genes) (**Fig. S21E**), consistent with our observation that SNAP-a involves relatively few of the kinds of “cell-identity” genes that all astrocytes express much more than other neural cell types do – the kinds of genes that tend to be regulated by super-enhancers – but rather is about modules with more dynamic forms of regulation.

- The authors' exposure of astrocytic cholesterol synthesis is interesting and could provide further support for a long-standing hypothesis. However, their choice to focus on this particular GO term appears as subjective. In fact, half of the genes indicated in Fig. 2G under the headline "cholesterol biosynthesis" are unrelated to this process, and more concerned with fatty acid or lipid metabolism in general, possibly reflecting SREBF1 activation. This suggests that the focus on cholesterol is too narrow.

This is a good point. We now describe the primary result as follows in **lines 169-177**,

In astrocytes, LF4 involved gene-expression changes very different from those in neurons (**Fig. 2A**), even for genes robustly expressed in both cell types (**Fig. S9M**). Gene sets with roles in fatty acid and cholesterol biosynthesis and export, including genes that encode the SREBP1 and SREBP2 transcription factors and the many genes they regulate (**Fig. 2D and Table S4**), were positively correlated with LF4 and under-expressed in the cortical astrocytes of donors with schizophrenia ($p = 7.2 \times 10^{-5}$, two-sided Wilcoxon rank-sum test, **Fig. 2D**) or advanced age ($p = 8.1 \times 10^{-3}$, **Fig. S13A**). Though many central nervous system (CNS) cell types express these lipid-biosynthesis genes, the repression of these genes occurred specifically in astrocytes (**Fig. S14**).

In the legend for **Fig. 2**, we also now write in **lines 531-533**,

referred to as “cholesterol biosynthesis” genes according to their annotation by GO, though subsets of these genes contribute to cholesterol export and/or to synthesis of additional fatty acids

We also now write, in **lines 414-426** in the Discussion,

An intriguing aspect of SNAP involved the astrocyte-specific regulation of genes with roles in fatty acid and cholesterol biosynthesis and cholesterol export, including genes that encode the SREBP1 and SREBP2 transcription factors and the many genes they regulate. Expression of these genes by astrocytes was strongly correlated (across donors) with expression of synaptic-component genes by neurons (**Fig. 2, D and E**). Earlier research has defined a potential rationale for this neuron-astrocyte coordination: synapses and dendritic spines require large amounts of cholesterol, which they derive from astrocytes (Pfrieger 2003; Goritz, Mauch, and Pfrieger 2005; Pfrieger and Ungerer 2011). Declines in cholesterol biosynthesis have previously been noted in mouse models of brain disorders (Valenza et al. 2010; Gangwani et al. 2023) that (like schizophrenia and aging) involve cognitive losses, cortical thinning, and reduction in neuropil. Products of this biosynthetic pathway also include other lipids and cholesterol metabolites with roles at synapses, including 24S-hydroxycholesterol, a positive allosteric modulator of NMDA receptors (Paul et al. 2013).

The authors should provide a more detailed analysis and display of the astrocyte- (and certainly neuron-) specific pathways that are "contained" in SNAP and SNAP-a.

Thank you for this suggestion. We now provide this in a revised **Fig. 3K**, which includes affected sets of genes in SNAP-a and SNAP-n. (This also makes it possible to visualize how a given set of genes can be coordinately regulated in astrocytes but not in neurons, or in neurons but not in astrocytes.)

More broadly, the revised manuscript also develops results for additional astrocyte gene sets (beyond lipid biosynthesis) whose expression is also strongly correlated with neuronal

expression levels of synaptic-component genes in the same donors. These include astrocytic expression of genes with synapse-adhesion functions (*NRXN1*, *NTM*, *CTNND2*, *LSAMP*, *GPM6A*, *LRRC4C*, *LRRTM4*, and *EPHB1*) and of genes with neurotransmitter-uptake functions (*SLC1A2* and *SLC1A3* (encoding glutamate transporters EAAT1 and EAAT2) and *SLC6A1* and *SLC6A11* (encoding GABA transporters GAT1 and GAT3)) (**Fig. 2E**). So, the changes related to lipid biosynthesis are just one part of a much larger astrocytic transcriptional program.

In this context, one wonders how the SREBF-dependent modules behave in neurons, do they change in inverted directions?

We do not find a significant change in neuronal expression of these genes in SNAP-n or in schizophrenia. In a new Supplementary Figure (**Fig. S14**) we observe that case-control differences in the GO “cholesterol-biosynthesis” gene set (which as you pointed out, strongly overlaps with the SREBF-dependent modules and contains genes with additional functions) are specific to astrocytes; neither glutamatergic nor GABAergic neurons (nor oligodendrocytes) exhibit this case-control difference (nor do they change in the opposite direction, at least not to an extent we can detect).

Finally, given established functions of astrocytes, one wonders whether SNAP comprises components that handle neurotransmitter and energy requirements.

This was a great suggestion that has led to additional analyses and text. SNAP-a very strongly regulates these neurotransmitter-handling genes in astrocytes: *SLC1A2* and *SLC1A3* (encoding the glutamate transporters EAAT1 and EAAT2) and *SLC6A1* and *SLC6A11* (encoding the GABA transporters GAT1 and GAT3) are all among the 1% of genes most strongly regulated by SNAP-a. This result is now in **Fig. 2E** of the revised manuscript.

Minor points

- Line 32 etc. The authors should consider to replace the word "program" by "module" or similar, which seem more neutral with respect to function.

We considered this but would like to keep the term “program”, as our results suggest that SNAP contains many modules (each implemented by different transcription factors) that *can* be regulated independently but are co-recruited in the context of SNAP. We reserve the term “module” for those smaller sets of genes (such as the sterol-biosynthesis genes, regulated by SREBF1/SREBF2) that share the same transcription-factor regulators and appear to be co-regulated in almost all contexts.

- Line 42: The term "transcellular" is somewhat ill-defined, it could be replaced by "intercellular" or "multicellular" depending on the context.

We agree, and have replaced "transcellular" with "multicellular" in **lines 47, 67, and 99** to improve clarity.

- Lines 61-77: This part should be rewritten or removed altogether. In any case, technical details should be moved to the Methods section, duplicate statement (number of donors etc.) must be avoided.

Thank you for pointing this out. We have shortened this section by removing redundancies and moved most of the details to the Methods section in **lines 1273-1327**.

- Line 65: To provide a global view on their donor cohort, authors should show the age/sex distribution of control subjects and schizophrenia patients using histograms.

We now provide these histograms in a new **Fig. S1** and reference this figure in **line 81**.

- Lines 65: The authors should indicate in the main text (here or before) that they analysed frozen post-mortem samples.

We have updated the text in **lines 79-84** to mention that we analyzed frozen post-mortem tissue samples: "Analyses included frozen post-mortem dIPFC from 191 donors (ages 22-97, median 64), including 97 neurotypical individuals and 94 affected by schizophrenia, and including 116 men and 75 women (**Fig. S1 and Table S1**). We used single-nucleus rather than single-cell RNA-seq to avoid effects of cell morphology upon ascertainment, and because nuclear membranes (but not plasma membranes) remain intact in frozen post-mortem tissue."

- Results: The headlines (and the subdivision) need to be revised as there is a mix of simple terms ("experimental approach", "cell-type abundances") and summarizing statements. Moreover, some paragraphs do not fit under the respective headline (example: lines 271-280).

We have revised the headlines in **lines 75, 99, 140, 181, 202, and 314** so that they are all summarizing statements rather than the simple terms used for the first several sections. We have also reorganized the paragraphs so that the headlines better describe their respective sections (e.g. introducing a new section on "Inference of multicellular gene-expression programs"). Regarding the paragraph in lines 271-280, we would like to keep it in the current section as it emphasizes that the genes and regulons associated with SNAP in astrocytes are distinct from those that participate in SNAP in neurons. We have modified the introductory

sentence of this paragraph in **lines 285-288** so that the purpose of this analysis as it relates to astrocyte gene-expression programs is more clear.

- Line 98: The term "cell-nonautonomous biology" should be replaced.

We have removed the line containing this term during the revision of this paragraph.

- Line 105: The use of the PEER algorithm (please explain abbreviation!) should be explained and justified (see Xue et al. 2022 Biorxiv).

We have now introduced the full term (probabilistic estimation of expression residuals) prior to the abbreviation PEER in the Methods section in **lines 1440-1441**. We selected PEER over other approaches (e.g. PCA) for inferring latent variables as it is more sensitive and less dependent on the number of factors modeled. As described in (Xue et al. 2023), a major pitfall to avoid when performing latent factor analysis with PEER is the problem of obtaining highly correlated factors due to overfitting when too many latent factors are sought in analysis. The latent factors we have inferred are independent from each other when we compare their gene loadings (**Fig. S8C**), enabling us to proceed with downstream analyses based on these factors.

- Lines 102-106: The explanation could be moved to the Methods section.

We have moved these lines as well as the above explanation and justification to the Methods section under "Latent factor analysis" in **lines 1437-1446**.

- Lines 106: One "each" should be removed.

We have removed the redundant text in what is now **line 103**.

- Lines 123-124: The authors should explain how the association was established. This is not clear from their description.

Thank you for pointing this out. We now include this explanation where we describe the association in **lines 117-119**: "Schizophrenia associated with just one of these latent factors (LF4) ($p = 6.1 \times 10^{-5}$, two-sided Wilcoxon rank-sum test of latent factor expression levels (by donor) between cases and controls) (**Fig. 1H, Fig. S9, A to E, and Table S2**)."

- Line 163 etc: This must read Fig. 1G.

We have corrected the reference to this figure, which (after reorganization of the figures) is now **Fig. 2D**, in **lines 169-179**.

- Line 196: The reference to Fig. 2C is unclear.

Thank you for pointing this out. We had meant to refer to the right panel of Fig. 2C, which compares individual donors' expression levels for LF4, grouped by case-control status. We have reorganized this figure so that we now refer to a separate panel that only displays these plots (**Fig. 1J** in the revised manuscript), and mention the relevance of this figure to the paragraph in the figure call-out (previously line 196, now **line 365**).

- Lines 232-243, 401-409: Large parts of these paragraph should be placed in the discussion.

We have condensed lines 232-243 and moved most of this text to the Discussion in **lines 400-412**. We have removed lines 401-409 together with the rest of the section on the eQTL results as recommended above.

- Line 489: The abbreviation BA46 should be explained.

We have now introduced the full term (Brodmann area 46) prior to the abbreviation BA46 in what is now **line 635**.

- Fig. 1C: The small fleck located at 6 o'clock / south of the larger endothelial cell island is not represented by the selected markers. Is this part inhabited by pericytes?

This group of ~1,400 cells expresses *GRM3*, *PDGFRB*, and *SHISA6*, which corresponds to a distinct subpopulation of pericytes named "Pericyte 2" in (Garcia et al. 2022). These have been described to express genes related to cardiac muscle cell action potential, cell-cell adhesion, and vascular smooth muscle contraction. We have updated our Methods section in **lines 1343-1346** to note that our analyses on endothelial cells combine these pericytes with the *CLDN5*⁺ population.

- Fig. S5F: Overlap of gene names renders them unreadable.

Thank you for pointing out this formatting error in Fig. S5F (now **Fig. S8F**). The gene names have been repositioned so that they no longer overlap.

Responses to Reviewer 3 comments, suggestions, and ideas

The paper by McCarroll and colleagues represents a major contribution to our understanding of multicellular contributions to schizophrenia-associated pathophysiology. By using sophisticated analyses of healthy controls and schizophrenia postmortem PFC snRNAseq data, the authors set out to pinpoint cellular contributions. Perhaps as expected, neurons were implicated, but surprisingly astrocytes were too. Among these, a shared synaptic-neuron-astrocyte-program (SNAP) was identified, that existed even among neurotypical samples and was reduced in schizophrenia. This is the central finding from this study, along with its subsequent exploration. The existence of SNAP clearly implicates astrocytes and neurons in genetic vulnerability to schizophrenia. This is an important and conceptually deep contribution, as it is emerging that astrocytes contribute to psychiatric phenotypes in causal ways.

The paper is beautifully written and the figures are clear. The discovery of SNAP is significant and markedly enriches our understanding of schizophrenia mechanisms. The detailed analyses of SNAPa reveals aspects of astrocyte biology and its intersection with psychiatric disease that are new and compelling. Overall, this study will likely have a major impact and garner interest from diverse readership across the board.

Thank you. We are grateful for your assessment of the work and for this perspective on its impact.

The authors should consider the following suggestions.

Suggestions

The section near lines 250-264 is important as it shows that astrocytes in schizophrenia do not show defined polarized states as defined by several suggested astrocyte reactivity markers. On line 263 the authors cite a review on this topic, but since then two studies have used experimental and/or disease contexts to explore this issue and found that astrocytes display context-specific transcriptional responses beyond simple polarized states, which I believe is in accord with the suggestions made in the current study. These two papers are: PMID: 35614216, PMID: 33086039. The authors may consider citing them appropriately near line 263.

We appreciate these suggestions for additional references, which we have added in that paragraph.

We have also added an additional analysis in **lines 248-261** that finds connections of SNAP to specific gene-expression modules found in that earlier work:

We sought to relate SNAP-a to an emerging appreciation of astrocyte heterogeneity and its basis in gene expression (Khakh and Deneen 2019). An earlier analysis of astrocyte molecular and morphological diversity in mice identified 62 gene-expression modules based on their co-expression relationships (Endo et al. 2022). Among these gene modules, the genes recruited by SNAP-a exhibited the strongest overlap ($p = 3.5 \times 10^{-4}$, $q = 0.015$) with the module that had correlated most closely with astrocyte morphological parameters related to the size of the territory covered by astrocyte processes (the “turquoise” module in (Endo et al. 2022), with overlap driven by genes including *Ezr* and *Ntm*) (**Table S5**). A potential interpretation is that SNAP-a supports these perisynaptic astrocytic processes (PAPs, reviewed in (Lawal, Ulloa Severino, and Eroglu 2022; Ghézali, Dallérac, and Rouach 2016; Aboufares El Alaoui et al. 2020)). Further evidence for a role supporting PAPs was provided by the strong SNAP-a recruitment of the genes encoding synaptic neurotransmitter transporters (*SLC1A2*, *SLC1A3*, *SLC6A1* and *SLC6A11*) and synaptic-adhesion proteins (*NRXN1*, *NTM*, *CTNND2*, *LSAMP*, *GPM6A*, *LRRC4C*, *LRRTM4*, and *EPHB1*).

In fig 2g, astrocytic genes involved in cholesterol biosynthesis are highlighted. This is remarkable, as the same genes are also now strongly implicated in several neurodegenerative diseases (AD, HD for example). It may be worth mentioning this as it suggests possible convergence of astrocytic mechanisms.

Thank you for pointing this out. We now include a paragraph in the Discussion about these effects, including a mention of this relationship to that earlier work in **lines 421-424**:

Declines in cholesterol biosynthesis have previously been noted in mouse models of brain disorders (Valenza et al. 2010; Gangwani et al. 2023) that (like schizophrenia and aging) involve cognitive losses, cortical thinning, and reduction in neuropil.

The reference to Perea et al’s review on tripartite synapses near line 423 needs to be thoughtfully considered. First, it remains unclear if tripartite synapses comprising pre, post and astrocytic elements are the norm. The latest electron microscopy data suggest that many synapses may not include an astrocyte element and even those that do have variable placement of astrocyte processes relative to the synapse. Therefore, the term tripartite synapse has no anatomical meaning and should be avoided (it’s not at all as definitive as bipartite synapse). Instead of referring to tripartite synapses, the authors could simply state “.....SNAP involves the biology of astrocyte-synapse functional and spatial interactions....”. Then, they could reference a more compelling review such as pubmed ID 29096081. Furthermore, the subsequent discussion of cell adhesion molecules, synaptic biogenesis cues, neurotransmitter uptake and cholesterol biosynthesis that SNAPa beautifully identified were never part of the “tripartite synapse” idea, which focused on astrocytes being active participants of fast synaptic signaling by

means of release of their own neurotransmitters. This idea has not held true and the term tripartite synapse is a loaded and highly controversial term (with little meaning). By using it, the authors hinder the broad appeal of their findings that in fact are consistent with much recent biology on how astrocytes do regulate synapses in a proven and measurable way. They may consider citing the following reviews instead: PMID: 30309945. PMID: 29096081

We appreciate this suggestion and have revised these passages accordingly in **lines 400-412**. We had not meant to reify any one position about synapses, just wanted to acknowledge that the idea of significant astrocyte participation at synapses has a much earlier provenance than our work (in way that might not be obvious to e.g. a reader from human genetics or single-cell analysis methods). The reviews/citations suggested in the above comments provide a better way of doing that. Thank you.

Minor comment: I like the acronym of SNAP. However, I wonder if this will cause superficial confusion with SNAP proteins involved in the vesicle cycle? Perhaps consider neuron-astrocyte-synaptic program (NASP) or neuron-astrocyte-program (NAP) to make it separable. This is a minor suggestion.

We went through this same discussion internally, and arrived at SNAP for the following reasons. One reason is that the soluble NSF attachment proteins with SNAP within their names all have longer names that distinguish them by molecular weights: SNAP25, SNAP23, SNAP29, etc. "NASP" (the alternative suggested) actually is already in formal use as a gene symbol, and "NAP" is (like SNAP) a part of several longer gene names (e.g. genes encoding NSF attachment proteins NAPA, NAPB, etc., and genes encoding the nucleosome assembly proteins NAP1L1, ... , NAP1L5). Finally, SNAP25 and other synaptic-vesicle genes (SV2, etc.) are actually among the genes most strongly recruited by SNAP-n, so in some ways even the partial similarity of acronyms helps evoke a key component of the gene-expression program SNAP recruits.

Minor comment: line 213. Change "180 thousand" to "180,000"

We have changed this wording in the text in **lines 215 and 227**.

Overall, an excellent study.

Thank you. We really appreciate the above thoughts and suggestions.

References

- Aboufaires El Alaoui, Amina, Molly Jackson, Mara Fabri, Luisa de Vivo, and Michele Bellesi. 2020. "Characterization of Subcellular Organelles in Cortical Perisynaptic Astrocytes." *Frontiers in Cellular Neuroscience* 14: 573944.
- Adamowicz, David H., and Ellen E. Lee. 2023. "Dementia among Older People with Schizophrenia: An Update on Recent Studies." *Current Opinion in Psychiatry* 36 (3): 150–55.
- Azevedo, Frederico A. C., Ludmila R. B. Carvalho, Lea T. Grinberg, José Marcelo Farfel, Renata E. L. Ferretti, Renata E. P. Leite, Wilson Jacob Filho, Roberto Lent, and Suzana Herculano-Houzel. 2009. "Equal Numbers of Neuronal and Nonneuronal Cells Make the Human Brain an Isometrically Scaled-up Primate Brain." *The Journal of Comparative Neurology* 513 (5): 532–41.
- Bahney, Jami, and Christopher S. von Bartheld. 2014. "Validation of the Isotropic Fractionator: Comparison with Unbiased Stereology and DNA Extraction for Quantification of Glial Cells." *Journal of Neuroscience Methods* 222 (January): 165–74.
- Bakken, Trygve E., Rebecca D. Hodge, Jeremy A. Miller, Zizhen Yao, Thuc Nghi Nguyen, Brian Aevermann, Eliza Barkan, et al. 2018. "Single-Nucleus and Single-Cell Transcriptomes Compared in Matched Cortical Cell Types." *PloS One* 13 (12): e0209648.
- Bakken, Trygve E., Nikolas L. Jorstad, Qiwen Hu, Blue B. Lake, Wei Tian, Brian E. Kalmbach, Megan Crow, et al. 2021. "Comparative Cellular Analysis of Motor Cortex in Human, Marmoset and Mouse." *Nature* 598 (7879): 111–19.
- Brown, P. O., and D. Botstein. 1999. "Exploring the New World of the Genome with DNA Microarrays." *Nature Genetics* 21 (1 Suppl): 33–37.
- Burda, Joshua E., Timothy M. O'Shea, Yan Ao, Keshav B. Suresh, Shinong Wang, Alexander M. Bernstein, Ashu Chandra, et al. 2022. "Divergent Transcriptional Regulation of Astrocyte Reactivity across Disorders." *Nature* 606 (7914): 557–64.
- Cai, Laisheng, and Jingwei Huang. 2018. "Schizophrenia and Risk of Dementia: A Meta-Analysis Study." *Neuropsychiatric Disease and Treatment* 14 (August): 2047–55.
- Callicott, Joseph H. 2003. "An Expanded Role for Functional Neuroimaging in Schizophrenia." *Current Opinion in Neurobiology* 13 (2): 256–60.
- Choi, Yeonsoo, Haram Park, Hwajin Jung, Hanseul Kweon, Seoyeong Kim, Soo Yeon Lee, Hyemin Han, et al. 2019. "NGL-1/LRRC4C Deletion Moderately Suppresses Hippocampal Excitatory Synapse Development and Function in an Input-Independent Manner." *Frontiers in Molecular Neuroscience* 12 (May): 119.
- Constantinides, Constantinos, Laura K. M. Han, Clara Alloza, Linda Antonella Antonucci, Celso Arango, Rosa Ayesa-Arriola, Nerisa Banaj, et al. 2023. "Brain Ageing in Schizophrenia: Evidence from 26 International Cohorts via the ENIGMA Schizophrenia Consortium." *Molecular Psychiatry* 28 (3): 1201–9.
- Corces, M. Ryan, Anna Shcherbina, Soumya Kundu, Michael J. Gludemans, Laure Frésard, Jeffrey M. Granja, Bryan H. Louie, et al. 2020. "Single-Cell Epigenomic Analyses Implicate Candidate Causal Variants at Inherited Risk Loci for Alzheimer's and Parkinson's Diseases." *Nature Genetics* 52 (11): 1158–68.
- Dietsche, Bruno, Tilo Kircher, and Irina Falkenberg. 2017. "Structural Brain Changes in Schizophrenia at Different Stages of the Illness: A Selective Review of Longitudinal Magnetic Resonance Imaging Studies." *The Australian and New Zealand Journal of Psychiatry* 51 (5): 500–508.
- Eberlé, Delphine, Bronwyn Hegarty, Pascale Bossard, Pascal Ferré, and Fabienne Foufelle. 2004. "SREBP Transcription Factors: Master Regulators of Lipid Homeostasis." *Biochimie*

- 86 (11): 839–48.
- Eling, Nils, Michael D. Morgan, and John C. Marioni. 2019. “Challenges in Measuring and Understanding Biological Noise.” *Nature Reviews. Genetics* 20 (9): 536–48.
- Endo, Fumito, Atsushi Kasai, Joselyn S. Soto, Xinzhu Yu, Zhe Qu, Hitoshi Hashimoto, Viviana Gradinaru, Riki Kawaguchi, and Baljit S. Khakh. 2022. “Molecular Basis of Astrocyte Diversity and Morphology across the CNS in Health and Disease.” *Science* 378 (6619): eadc9020.
- Escartin, Carole, Elena Galea, András Lakatos, James P. O’Callaghan, Gabor C. Petzold, Alberto Serrano-Pozo, Christian Steinhäuser, et al. 2021. “Reactive Astrocyte Nomenclature, Definitions, and Future Directions.” *Nature Neuroscience* 24 (3): 312–25.
- Faure, Andre J., Jörn M. Schmiedel, and Ben Lehner. 2017. “Systematic Analysis of the Determinants of Gene Expression Noise in Embryonic Stem Cells.” *Cell Systems* 5 (5): 471–84.e4.
- Formoso, Karina, Micaela D. Garcia, Alberto C. Frasch, and Camila Scorticati. 2016. “Evidence for a Role of Glycoprotein M6a in Dendritic Spine Formation and Synaptogenesis.” *Molecular and Cellular Neurosciences* 77 (December): 95–104.
- Fucetola, R., L. J. Seidman, W. S. Kremen, S. V. Faraone, J. M. Goldstein, and M. T. Tsuang. 2000. “Age and Neuropsychologic Function in Schizophrenia: A Decline in Executive Abilities beyond That Observed in Healthy Volunteers.” *Biological Psychiatry* 48 (2): 137–46.
- Gabi, Mariana, Kleber Neves, Carolinne Masseron, Pedro F. M. Ribeiro, Lissa Ventura-Antunes, Laila Torres, Bruno Mota, Jon H. Kaas, and Suzana Herculano-Houzel. 2016. “No Relative Expansion of the Number of Prefrontal Neurons in Primate and Human Evolution.” *Proceedings of the National Academy of Sciences of the United States of America* 113 (34): 9617–22.
- Gangwani, Mohitkumar R., Joselyn S. Soto, Yasaman Jami-Alahmadi, Srushti Tiwari, Riki Kawaguchi, James A. Wohlschlegel, and Baljit S. Khakh. 2023. “Neuronal and Astrocytic Contributions to Huntington’s Disease Dissected with Zinc Finger Protein Transcriptional Repressors.” *Cell Reports* 42 (1): 111953.
- Garcia, Francisco J., Na Sun, Hyeseung Lee, Brianna Godlewski, Hansruedi Mathys, Kyriaki Galani, Blake Zhou, et al. 2022. “Single-Cell Dissection of the Human Brain Vasculature.” *Nature* 603 (7903): 893–99.
- Ghézali, Grégory, Glenn Dallérac, and Nathalie Rouach. 2016. “Perisynaptic Astroglial Processes: Dynamic Processors of Neuronal Information.” *Brain Structure & Function* 221 (5): 2427–42.
- Goritz, Christian, Daniela H. Mauch, and Frank W. Pfrieger. 2005. “Multiple Mechanisms Mediate Cholesterol-Induced Synaptogenesis in a CNS Neuron.” *Molecular and Cellular Neurosciences* 29 (2): 190–201.
- Göverti, Diğdem, Nihan Büyüklüoğlu, Hasan Kaya, Rabia Nazik Yüksel, Çiğdem Yücel, and Erol Göka. 2022. “Neuronal Pentraxin-2 (NPTX2) Serum Levels during an Acute Psychotic Episode in Patients with Schizophrenia.” *Psychopharmacology* 239 (8): 2585–91.
- Hack, A. A., C. T. Ly, F. Jiang, C. J. Clendenin, K. S. Sigrist, R. L. Wollmann, and E. M. McNally. 1998. “Gamma-Sarcoglycan Deficiency Leads to Muscle Membrane Defects and Apoptosis Independent of Dystrophin.” *The Journal of Cell Biology* 142 (5): 1279–87.
- Hashimoto, Takashi, Shohei Maekawa, and Seiji Miyata. 2009. “IgLON Cell Adhesion Molecules Regulate Synaptogenesis in Hippocampal Neurons.” *Cell Biochemistry and Function* 27 (7): 496–98.
- Head, Denise, Karen M. Rodrigue, Kristen M. Kennedy, and Naftali Raz. 2008. “Neuroanatomical and Cognitive Mediators of Age-Related Differences in Episodic Memory.” *Neuropsychology* 22 (4): 491–507.
- Henderson, Nathan T., and Matthew B. Dalva. 2018. “EphBs and Ephrin-Bs: Trans-Synaptic

- Organizers of Synapse Development and Function.” *Molecular and Cellular Neurosciences* 91 (September): 108–21.
- Herculano-Houzel, Suzana. 2009. “The Human Brain in Numbers: A Linearly Scaled-up Primate Brain.” *Frontiers in Human Neuroscience* 3 (November): 31.
- Hodge, Rebecca D., Trygve E. Bakken, Jeremy A. Miller, Kimberly A. Smith, Eliza R. Barkan, Lucas T. Graybuck, Jennie L. Close, et al. 2019. “Conserved Cell Types with Divergent Features in Human versus Mouse Cortex.” *Nature* 573 (7772): 61–68.
- Horton, Jay D., Nila A. Shah, Janet A. Warrington, Norma N. Anderson, Sahng Wook Park, Michael S. Brown, and Joseph L. Goldstein. 2003. “Combined Analysis of Oligonucleotide Microarray Data from Transgenic and Knockout Mice Identifies Direct SREBP Target Genes.” *Proceedings of the National Academy of Sciences of the United States of America* 100 (21): 12027–32.
- Karayel, Ozge, Sebastian Virreira Winter, Shalini Padmanabhan, Yuliya I. Kuras, Duc Tung Vu, Idil Tuncali, Kalpana Merchant, Anne-Marie Wills, Clemens R. Scherzer, and Matthias Mann. 2022. “Proteome Profiling of Cerebrospinal Fluid Reveals Biomarker Candidates for Parkinson’s Disease.” *Cell Reports. Medicine* 3 (6): 100661.
- Khakh, Baljit S., and Benjamin Deneen. 2019. “The Emerging Nature of Astrocyte Diversity.” *Annual Review of Neuroscience* 42 (July): 187–207.
- Kotliar, Dylan, Adrian Veres, M. Aurel Nagy, Shervin Tabrizi, Eran Hodis, Douglas A. Melton, and Pardis C. Sabeti. 2019. “Identifying Gene Expression Programs of Cell-Type Identity and Cellular Activity with Single-Cell RNA-Seq.” *eLife* 8 (July). <https://doi.org/10.7554/eLife.43803>.
- Koutsouleris, Nikolaos, Christos Davatzikos, Stefan Borgwardt, Christian Gaser, Ronald Bottlender, Thomas Frodl, Peter Falkai, et al. 2014. “Accelerated Brain Aging in Schizophrenia and beyond: A Neuroanatomical Marker of Psychiatric Disorders.” *Schizophrenia Bulletin* 40 (5): 1140–53.
- Krienen, Fenna M., Melissa Goldman, Qiangge Zhang, Ricardo C H Del Rosario, Marta Florio, Robert Machold, Arpiar Saunders, et al. 2020. “Innovations Present in the Primate Interneuron Repertoire.” *Nature* 586 (7828): 262–69.
- Lawal, Oluwadamilola, Francesco Paolo Ulloa Severino, and Cagla Eroglu. 2022. “The Role of Astrocyte Structural Plasticity in Regulating Neural Circuit Function and Behavior.” *Glia* 70 (8): 1467–83.
- León, Antonella, Gabriela I. Aparicio, and Camila Scorticati. 2021. “Neuronal Glycoprotein M6a: An Emerging Molecule in Chemical Synapse Formation and Dysfunction.” *Frontiers in Synaptic Neuroscience* 13 (May): 661681.
- Liddel, Shane A., Kevin A. Guttenplan, Laura E. Clarke, Frederick C. Bennett, Christopher J. Bohlen, Lucas Schirmer, Mariko L. Bennett, et al. 2017. “Neurotoxic Reactive Astrocytes Are Induced by Activated Microglia.” *Nature* 541 (7638): 481–87.
- Liebers, David T., Mehdi Pirooznia, Fayaz Seiffudin, Katherine L. Musliner, Peter P. Zandi, and Fernando S. Goes. 2016. “Polygenic Risk of Schizophrenia and Cognition in a Population-Based Survey of Older Adults.” *Schizophrenia Bulletin* 42 (4): 984–91.
- Lin, Ching-En, Chi-Hsiang Chung, Li-Fen Chen, and Mei-Ju Chi. 2018. “Increased Risk of Dementia in Patients with Schizophrenia: A Population-Based Cohort Study in Taiwan.” *European Psychiatry: The Journal of the Association of European Psychiatrists* 53 (September): 7–16.
- Mackay, Trudy F. C., Eric A. Stone, and Julien F. Ayroles. 2009. “The Genetics of Quantitative Traits: Challenges and Prospects.” *Nature Reviews. Genetics* 10 (8): 565–77.
- Macosko, Evan Z., Anindita Basu, Rahul Satija, James Nemesh, Karthik Shekhar, Melissa Goldman, Itay Tirosh, et al. 2015. “Highly Parallel Genome-Wide Expression Profiling of Individual Cells Using Nanoliter Droplets.” *Cell* 161 (5): 1202–14.
- Makowski, Carolina, Dennis van der Meer, Weixiu Dong, Hao Wang, Yan Wu, Jingjing Zou, Cin

- Liu, et al. 2022. "Discovery of Genomic Loci of the Human Cerebral Cortex Using Genetically Informed Brain Atlases." *Science* 375 (6580): 522–28.
- Malik, Athar N., Thomas Vierbuchen, Martin Hemberg, Alex A. Rubin, Emi Ling, Cameron H. Couch, Hume Stroud, et al. 2014. "Genome-Wide Identification and Characterization of Functional Neuronal Activity-Dependent Enhancers." *Nature Neuroscience* 17 (10): 1330–39.
- Paul, Steven M., James J. Doherty, Albert J. Robichaud, Gabriel M. Belfort, Brian Y. Chow, Rebecca S. Hammond, Devon C. Crawford, et al. 2013. "The Major Brain Cholesterol Metabolite 24(S)-Hydroxycholesterol Is a Potent Allosteric Modulator of N-Methyl-D-Aspartate Receptors." *The Journal of Neuroscience: The Official Journal of the Society for Neuroscience* 33 (44): 17290–300.
- Pfrieger, Frank W. 2003. "Outsourcing in the Brain: Do Neurons Depend on Cholesterol Delivery by Astrocytes?" *BioEssays: News and Reviews in Molecular, Cellular and Developmental Biology* 25 (1): 72–78.
- Pfrieger, Frank W., and Nicole Ungerer. 2011. "Cholesterol Metabolism in Neurons and Astrocytes." *Progress in Lipid Research* 50 (4): 357–71.
- Ribe, Anette Riisgaard, Thomas Munk Laursen, Morten Charles, Wayne Katon, Morten Fenger-Grøn, Dimitry Davydow, Lydia Chwastiak, Joseph M. Cerimele, and Mogens Vestergaard. 2015. "Long-Term Risk of Dementia in Persons With Schizophrenia: A Danish Population-Based Cohort Study." *JAMA Psychiatry* 72 (11): 1095–1101.
- Saint-Martin, Margaux, and Yukiko Goda. 2022. "Astrocyte-Synapse Interactions and Cell Adhesion Molecules." *The FEBS Journal*, June. <https://doi.org/10.1111/febs.16540>.
- Shimano, Hitoshi, and Ryuichiro Sato. 2017. "SREBP-Regulated Lipid Metabolism: Convergent Physiology - Divergent Pathophysiology." *Nature Reviews. Endocrinology* 13 (12): 710–30.
- Siddiqui, Tabrez J., Parisa Karimi Tari, Steven A. Connor, Peng Zhang, Frederick A. Dobie, Kevin She, Hiroshi Kawabe, Yu Tian Wang, Nils Brose, and Ann Marie Craig. 2013. "An LRRTM4-HSPG Complex Mediates Excitatory Synapse Development on Dentate Gyrus Granule Cells." *Neuron* 79 (4): 680–95.
- Slovin, Shaked, Annamaria Carissimo, Francesco Panariello, Antonio Grimaldi, Valentina Bouché, Gennaro Gambardella, and Davide Cacchiarelli. 2021. "Single-Cell RNA Sequencing Analysis: A Step-by-Step Overview." *Methods in Molecular Biology* 2284: 343–65.
- Stroup, T. Scott, Mark Olfson, Cecilia Huang, Melanie M. Wall, Terry Goldberg, Davangere P. Devanand, and Tobias Gerhard. 2021. "Age-Specific Prevalence and Incidence of Dementia Diagnoses Among Older US Adults With Schizophrenia." *JAMA Psychiatry* 78 (6): 632–41.
- Tan, Christabel X., and Cagla Eroglu. 2021. "Cell Adhesion Molecules Regulating Astrocyte-Neuron Interactions." *Current Opinion in Neurobiology* 69 (August): 170–77.
- Trotter, Justin H., Zahra Dargaei, Alessandra Sclip, Sofia Essayan-Perez, Kif Liakath-Ali, Karthik Raju, Amber Nabet, Xinran Liu, Markus Wöhr, and Thomas C. Südhof. 2021. "Compartment-Specific Neurexin Nanodomains Orchestrate Tripartite Synapse Assembly." *bioRxiv*. <https://doi.org/10.1101/2020.08.21.262097>.
- Trubetsky, Vassily, Antonio F. Pardiñas, Ting Qi, Georgia Panagiotaropoulou, Swapnil Awasthi, Tim B. Bigdeli, Julien Bryois, et al. 2022. "Mapping Genomic Loci Implicates Genes and Synaptic Biology in Schizophrenia." *Nature*, April, 2020.09.12.20192922.
- Turner, Tychele N., Kamal Sharma, Edwin C. Oh, Yangfan P. Liu, Ryan L. Collins, Maria X. Sosa, Dallas R. Auer, et al. 2015. "Loss of δ -Catenin Function in Severe Autism." *Nature* 520 (7545): 51–56.
- Valenza, Marta, Valerio Leoni, Joanna M. Karasinska, Lara Petricca, Jianjia Fan, Jeffrey Carroll, Mahmoud A. Pouladi, et al. 2010. "Cholesterol Defect Is Marked across Multiple Rodent Models of Huntington's Disease and Is Manifest in Astrocytes." *The Journal of*

- Neuroscience: The Official Journal of the Society for Neuroscience* 30 (32): 10844–50.
- Van de Sande, Bram, Christopher Flerin, Kristofer Davie, Maxime De Waegeneer, Gert Hulselmans, Sara Aibar, Ruth Seurinck, et al. 2020. “A Scalable SCENIC Workflow for Single-Cell Gene Regulatory Network Analysis.” *Nature Protocols* 15 (7): 2247–76.
- Wit, Joris de, Matthew L. O’Sullivan, Jeffrey N. Savas, Giuseppe Condomitti, Max C. Caccese, Kristel M. Vennekens, John R. Yates 3rd, and Anirvan Ghosh. 2013. “Unbiased Discovery of Glypican as a Receptor for LRRTM4 in Regulating Excitatory Synapse Development.” *Neuron* 79 (4): 696–711.
- Xiao, Mei-Fang, Seung-Eon Roh, Jiechao Zhou, Chun-Che Chien, Brendan P. Lucey, Michael T. Craig, Lindsay N. Hayes, et al. 2021. “A Biomarker-Authenticated Model of Schizophrenia Implicating NPTX2 Loss of Function.” *Science Advances* 7 (48): eabf6935.
- Xue, Angli, Seyhan Yazar, Drew Neavin, and Joseph E. Powell. 2023. “Pitfalls and Opportunities for Applying Latent Variables in Single-Cell eQTL Analyses.” *Genome Biology* 24 (1): 33.
- Yao, Douglas W., Luke J. O’Connor, Alkes L. Price, and Alexander Gusev. 2020. “Quantifying Genetic Effects on Disease Mediated by Assayed Gene Expression Levels.” *Nature Genetics* 52 (6): 626–33.
- Yap, Ee-Lynn, and Michael E. Greenberg. 2018. “Activity-Regulated Transcription: Bridging the Gap between Neural Activity and Behavior.” *Neuron* 100 (2): 330–48.
- Yap, Ee-Lynn, Noah L. Pettit, Christopher P. Davis, M. Aurel Nagy, David A. Harmin, Emily Golden, Onur Dagliyan, et al. 2021. “Bidirectional Perisomatic Inhibitory Plasticity of a Fos Neuronal Network.” *Nature* 590 (7844): 115–21.
- Yu, Xinzhu, Jun Nagai, Maria Marti-Solano, Joselyn S. Soto, Giovanni Coppola, M. Madan Babu, and Baljit S. Khakh. 2020. “Context-Specific Striatal Astrocyte Molecular Responses Are Phenotypically Exploitable.” *Neuron* 108 (6): 1146–62.e10.

Reviewer Reports on the First Revision:

Referees' comments:

Referee #1 (Remarks to the Author):

This is a very thoughtful revision, which in its dedicated focus on presenting the SNAP phenomenon, is even more clear than before. In reading this revision, I agree that removing the eQTL analyses from this manuscript is the right way to go. The authors have addressed my previous comments, and I continue to find this work, its methods, and the presentation of SNAP to be exciting and impactful. I have no further comments or concerns.

Referee #2 (Remarks to the Author):

The revision in response to the referees' comments has greatly increased the interest in the authors' work notably by the addition of new data providing new insight and cross-validation. At the same time, however, the revision has also introduced several weaknesses mostly with respect to overall structure of the ms, but also at specific points in the text and figures. As stated in the previous assessment, these are a very important, groundbreaking and fascinating results for a wide readership. The weaknesses outlined below can certainly be eliminated ideally by a complete overhaul of the ms. Please accept my sincere apologies for any comments listed below that I should have made already during the first round of review:

Major:

1. This somewhat fundamental comment should be considered first, as it suggests a major reorganisation of the manuscript. Why? From a bird's-eye view, the authors apply latent factor analysis to explore gene expression variability to their precious dataset in two ways, first inter-individual variation among human donors, and second intercellular variation within specific cell populations. In the present ms, these analyses are presented in a highly dispersed manner (human donors: line 99ff; astrocytes: line 202ff; neurons: line 285ff), introduced with somehow weak and unconvincing arguments (line 203 "to better appreciate astrocyte variation" or line 285: "To assess the cell-type specificity of SNAP-a..."), leading to flawed terminology (SNAP-a and SNAP-n seem to be misnomers, as each lacks the neuronal and astrocytic component, respectively), tiring repetition ("identify biological processes" and GSEA: lines 146 and again lines 234; or Figs. 1I, J, 3D, E) and eclectic exploration of the different SNAPs (see lines 300ff: why only SNAP and not SNAP-a, and -n? See also 315-357 with back and fourth references to figures). Moreover, text (see lines 93-97 and lines 205-210) and graphs (for example Figs. 1D-E with Fig. 3A-C,F-H and Fig 1I,J with Fig 3D,E,I,J; Figs. S3-7 with S17) that conceptually belong together are scattered. A statement like in line 227-228 somehow undermines the value of what has been presented in the first half of the ms. A remedy to remove these weaknesses that do not justice to the authors' precious and interesting data could be to present the application of latent factor analyses on both, samples and distinct cell types, in a single place right at the beginning (lines 99ff). An integrated analysis could then deliver a unified, cross-validated SNAP based on donor variation and intercorrelated cell-specific variation (see the authors' statement lines 221-224) that is subsequently submitted to the impressive range of analyses that the authors have lined up. Admittedly, this is a radical suggestion, but it has the potential to deliver a better argued, seamlessly flowing, and more impactful story.

The following comments are more specific, and may guide the revision of the ms regardless of the previous fundamental suggestion.

2. A weak point relates to the new and exciting results showing the impact of age. The present version still seems centered on schizophrenia and the age-related findings appear somewhat as bycatch. This may be chronologically correct, but it is conceptually weak. It appears more adept and interesting to expose the focus on age AND disease already in the abstract and introduction. The authors should modify the ms accordingly, and they should check throughout the ms, where the presentation of age-related results needs to be revamped and put on a par with schizophrenia-related findings. An example can be found in lines 117ff. Here, schizophrenia is mentioned first, and age appears more as an appendix. This could be changed to "One of the latent factors, LF4, was strongly impacted by age and schizophrenia...". Evidently, an additional focus on age also requires modifications to figures, where data are currently only stratified by disease, and where stratification by age should be added (e.g. Figs. 1H; 2C-E, S3-S7, S14, S18H, I). Specifically, the authors' data could reveal whether the fractions of nuclei attributed to specific cell types (subtypes of neurons and astrocytes) in the human brain decrease with age.

3. The title, abstract and the introductory paragraphs, which are key to attract readers' attention and to situate the work, need to be revised. This merits a bit more detail as I consider this as an important point:

3.1. Lines 1ff. The title is suboptimal, it seems to turn the concept upside down: who coordinates what? Neurons/astrocytes the program or the program their function? Moreover, the verb "coordinate" seems a bit teleologic and the important term "human" is missing. Just a suggestion: "Age- and schizophrenia-induced decline of correlated gene expression programs in human neurons and astrocytes".

3.2. Lines 25ff: The first sentence seems cryptic/trivial, the second risks to put readers on the wrong track: It suggests that the authors aim to understand variation, which is not the case. A clear formulation of the goal and the approach is missing. As far as I understood, inter-individual and intercellular variation of gene expression represent a so far neglected "treasure trove", whose exploitation by the authors reveals correlated functions of brain cells and their decline in age and disease.

3.3. Lines 26ff. The statement "To assess variation in cell-type specific gene expression programs..." appears a bit "weak" as aim, and it seems to anticipate the result (the discovery of a gene expression program) rather than to expose the "big question". The second part of the statement seems too detailed. Clearly here the focus on age AND disease should be mentioned.

3.4. The abstract contains repetitive statements that make it hard to read and that dampen readers' interest and attention: lines 31-34: "In persons in whom neurons invested more ..." is followed by "astrocytes invested more transcript in...". This is more verbose than elegant.

3.5. The introduction – in contrast to the abstract – opens with cell function enabled by cell collaboration (Lines 46ff). Then, the next paragraph (lines 54ff) goes from "In-free living species" (a term that is somewhat ill-defined) to "variation" without an explanation how this relates to the first paragraph.

4. Lines 99-102: The authors should revise this pivotal part of the Results section. In its present form, it lacks key information that readers require to appreciate the results shown after. First, the term "Inter-individual variation" appears out of nowhere and it is unclear what it refers to. So, one immediate solution is to change to "Inter-individual variation of gene expression...". Second, the authors need to show the phenomenon, i.e. that gene expression indeed varies. Otherwise, the latent factor analysis comes *deus ex machina*. None of the figures referred to up to this point (Fig. 1, S1-7) illustrates this key point. The authors should include a graph in Fig. 1 between panels E and F that shows inter-individual (and intercellular, see comment 1) variation of gene expression that they uncovered in their data set. For example, expression levels of selected transcripts in cell types and across all individuals or a more global parameter illustrating variation of cell-specific gene expression across individuals. A proper explanation and graphic display will alert readers of the phenomenon *per se*, and help to appreciate the authors' subsequent exploration of this valuable resource. Note, the statement in line

107 that "Ten latent factors together explained 30% of inter-individual variation..." is not understandable without a proper display of what inter-individual (and intercellular) variation of gene expression looks like.

5. Lines 117ff, 132ff, 500ff, Figs. 1H, S10A: At least to this referee, it is not entirely clear what is shown in these plots, last not least because the legends (line 500ff and 738ff) are too cursory. The authors should improve the description in the legends, and check for other instances where legends may not describe the figure content in sufficient detail.

6. Lines 136-138: The presentation of these valuable results appears as *en passant* and superficial. The rationale to do this experiment and the conclusion could be highlighted a bit more. In the phrase "an analogous constellation of changes manifests" it is unclear what "analogous" and "changes" refer to.

7. Lines 140-179 and 181-200: The titles and purposes of these chapters are unclear. In lines 140-179, this weakness originates from the terms "changes" and "concentration" (e.g. lines 160, 516, 733 etc.). It is not clear in each statement what the authors refer to. Do they mean changes due to variability/enrichment (so, it should read co-variation or co-variability) or changes due to schizophrenia and age. This must be clarified. Moreover, there seems to be some overlap with the subsequent chapter (e.g. lines 192-194), where again schizophrenia- and age-related changes are presented (lines 151; 164-166 etc.). Symptomatic for this intermingling is the fact that references for figures go back and fourth. In lines 181-191, the weak terms are "coordinated" (see comment 3.1) and "investments". What does investment mean? Isn't the main finding a "correlated expression of synapse-related genes in neurons and astrocytes"? Further, it is not clear how the genes to be analyzed here were chosen? Based on their presence in LF4? What does the sentence in lines 188-191 mean? Did the authors assume that the correlations were only valid for schizophrenic patients?

8. Lines 159-162; 516-520, Figs. 2B, S9L: The enrichment of synaptic components in LF4 is impressive, however the presentation of the data should be improved. The sunburst plots shown in Fig. S9L could be moved to the main figure 2B, as they illustrate more vividly the loading of synaptic genes in LF4 than the more conventional GO term plots (Fig. 2B). An illustration of astrocyte-specific gene enrichment should also be added to the main figure 2 between panel B and C to highlight this finding and to set the stage for subsequent analyses.

9. Lines 170ff, Fig. 2D: As stated in the first round of revision, at least half of the genes indicated by the authors in Fig. 2D are related to fatty acid biosynthesis or regulation (GPAM, SCD, SREBF1, ACACA, ACACB, FASN, ELOVL6) and for some, the relation to fatty acid or cholesterol synthesis is far-fetched (NFYC, KPNB1, ERLIN2). The authors acknowledge this partially in the text, but they have not changed the term in the figures. For the sake of precision I would recommend to replace in the figs 2E and S13 the axis label "regulation of cholesterol biosynthesis" by "cholesterol/fatty acid biosynthesis".

10. Lines 212ff: Here the same point of criticism applies as outlined in comment 3 concerning lines 99-102 and Fig. 1. The authors write about the interindividual variation among astrocytes but they do not illustrate this. Instead, they jump right into latent factor analysis. Could this be illustrated in Fig. 3? In this context, I wonder whether it would make sense to show in Fig. S18 in analogy to the latent factor analysis for human samples a graph similar as in Fig. S8A,B which indicates how many latent factors explain which percentage of variation? Evidently, this comment is subordinate to comment 1.

11. Line 222-224: It seems a bit strange that the authors mention only two computational approaches, although they used in fact three, as indicated in Fig. S19. The data shown in this figure could be exposed in a bit more detail in the main text.

12. Lines 226ff. The paragraph is unclear, last not least because in Lines 227-228 and 229-231, "than" is missing, and so it is not known what is compared here. Assuming that the authors indicate that SNAP-a is more precise than SNAP, this conclusion seems a bit premature given that in the following paragraphs, the authors explore "the biology supported by SNAP-a" (a phrase that could be changed to a more clear statement) for

example by exposing the top 20 genes etc. Here, one misses a comparison to SNAP and the GO terms that were explored at length in Fig. 2 and all associated supplementary figure. Surprisingly, the authors do not expose whether SNAP-a contains lipid synthesis genes that figured so prominently in the first part of the ms.

13. Lines 236-261: The text should be rewritten. In line 243, the reference to Fig. S20 is unclear as the graphs shown seem to have nothing to do with what the text refers to. Moreover, parts of this and of the next block (lines 255-261) should be moved to the Discussion.

14. Lines 269-283: The exploration of reactive astrocyte markers is a highly welcome element, but in the current presentation, it looks more like an add-on. The authors focus on the question of continuous versus discrete state shifts, but surprisingly, schizophrenia is not mentioned in the text although in Fig. S22, the authors stratify by schizophrenia. An evident question is how schizophrenia and age affect established reactive astrocyte markers in the authors' dataset - even beyond their association with SNAP-a. In the relevant paragraph (lines 276ff), the authors refer to SNAP-a, but not to disease or age and not to the set of published reactive astrocyte markers.

15. Lines 564-573, Fig. 3K: It is unclear how the figure can illustrate "coordinated expression" between astrocytes and neurons. The patterns shown in the two columns seem rather distinct.

Minor

- Lines 65ff and 76ff: The authors should consider to reunite the statements distributed in the two places. In any case, the explanation and references given in lines 76-79 should be indicated the first place where these neurons are mentioned.

- Lines 28, 66 and 80. The number of donors and the age range are mentioned three times within two pages, which is clearly in the way of a streamlined introduction.

- Lines 103-105, lines 108-110: The author should revise the sentences, which seem verbose and difficult to understand.

- Lines 107-112. Here or elsewhere in the text, there is no reference to panel Fig. S8F.

- The authors should consider to move panels F-K in Fig. S09 to a new figure. This would also avoid the somewhat odd reference to a preceding figure on line 136 following the introduction of Fig. S10 (line 134).

- Lines 205-206: "Analysis recognized" could be improved to "Our snRNASeq data recapitulated transcriptional profiles of distinct subtypes of adult astrocytes described previously...".

- Line 207: The statement could be complemented by "..., which populate the gray matter and represent the most frequent subtype in our samples."

- Lines 213-15: The parentheses surrounding the sentence can be removed.

- Line 224, line 232: Here and at other places, the authors refer back and fourth to figures (first S19, then back to S18F-I), which may confuse readers, notably with respect to content design and coherency of figures.

- Line 252: The numbers shown in parantheses are not well described. Which test was used, what does q stand for?

- Lines 263-267: This paragraph stands alone without any relation to the previous or next ones. The authors should move it to a more suitable place in the Results with better integration in the context.

- Lines 483, 492: "Experimental approach" and "Computational approach" sound too generic. "Experimental approach of human sample preparation" and "Design of latent factor analysis"

- Lines 481-485: Here and throughout, authors should consider exclusive use of past tense.

- Line 503: The authors should state "Correlation of with age" assuming that the indicated values are correlation coefficients. The latter should be mentioned in the legend.

- Line 513: Is "...in both cell types..." correct? Should this read "...in each cell type..."?

- Line 635: The explanation of BA46 as "Brodmann area 46" given here should be moved to line 483. The abbreviation is first used in panel A of Fig. 1. Here, and in all further instances, only the abbreviation should be used for consistency.
- Lines 696-697: The phrase should be changed to "from Hodge et al. (2019)".
- Fig. S18D, lines 840-841: The two color codes need to be explained. Is this schizophrenic versus control?
- In Fig. S18E: The authors should consider to apply the same color spectrum used for LF4 (going from blue to yellow) also for cNMF2 (which goes now from blue to something like light blue), obviously by re-mapping to a different value range. Thereby, the similarity of the distributions may become more visible.
- Figure legends: It seems unnecessary to repeat in each figure legend and for each figure panel what boxplots etc. indicate, unless this is stipulated by the journal's policy. On the other hand, violin plots are mentioned only once in legend of Fig. S1 although they are also shown in main Fig. 1, and there is no explanation what they show.

Referee #3 (Remarks to the Author):

I read the revised paper and the response to reviewers by McCarroll and colleagues. In my assessment, the authors have done an outstanding job with revisions, including expanding the scope of the work to include consideration of aging as requested by another reviewer. I have no further comments other than to congratulate the authors on an excellent study that will surely stimulate many follow up studies by the field.

Author Rebuttals to First Revision:

Concerted neuron-astrocyte gene expression declines in aging and schizophrenia

(Manuscript # 2022-12-19313B)

Responses to reviewer comments, suggestions, and ideas

Table of contents

Responses to Reviewer 1 comments, suggestions, and ideas.....	1
Responses to Reviewer 2 comments, suggestions, and ideas.....	2
Responses to Reviewer 3 comments, suggestions, and ideas.....	19
References	20

Responses to Reviewer 1 comments, suggestions, and ideas

This is a very thoughtful revision, which in its dedicated focus on presenting the SNAP phenomenon, is even more clear than before. In reading this revision, I agree that removing the eQTL analyses from this manuscript is the right way to go. The authors have addressed my previous comments, and I continue to find this work, its methods, and the presentation of SNAP to be exciting and impactful. I have no further comments or concerns.

We are grateful for this assessment of the work and manuscript.

Responses to Reviewer 2 comments, suggestions, and ideas

The revision in response to the referees' comments has greatly increased the interest in the authors' work notably by the addition of new data providing new insight and cross-validation. At the same time, however, the revision has also introduced several weaknesses mostly with respect to overall structure of the ms, but also at specific points in the text and figures.

As stated in the previous assessment, these are a very important, groundbreaking and fascinating results for a wide readership.

We appreciate this assessment of the work and its implications.

The weaknesses outlined below can certainly be eliminated ideally by a complete overhaul of the ms. Please accept my sincere apologies for any comments listed below that I should have made already during the first round of review:

We appreciate these thoughtful suggestions about presentation of results, and have responded to them point-by-point below.

Major:

1. This somewhat fundamental comment should be considered first, as it suggests a major reorganisation of the manuscript. Why? From a bird's-eye view, the authors apply latent factor analysis to explore gene expression variability to their precious dataset in two ways, first inter-individual variation among human donors, and second intercellular variation within specific cell populations. In the present ms, these analyses are presented in a highly dispersed manner (human donors: line 99ff; astrocytes: line 202ff; neurons: line 285ff), introduced with somehow weak and unconvincing arguments (line 203 "to better appreciate astrocyte variation" or line 285: "To assess the cell-type specificity of SNAP-a..."), leading to flawed terminology (SNAP-a and SNAP-n seem to be misnomers, as each lacks the neuronal and astrocytic component, respectively), tiring repetition ("identify biological processes" and GSEA: lines 146 and again lines 234; or Figs. 1I, J, 3D, E) and eclectic exploration of the different SNAPs (see lines 300ff: why only SNAP and not SNAP-a, and -n? See also 315-357 with back and fourth references to figures). Moreover, text (see lines 93-97 and lines 205-210) and graphs (for example Figs. 1D-E with Fig. 3A-C,F-H and Fig 1I,J with Fig 3D,E,I,J; Figs. S3-7 with S17) that conceptually belong together are scattered. A statement like in line 227-228 somehow undermines the value of what has been presented in the first half of the ms. A remedy to remove these weaknesses that do not justice to the authors' precious and interesting data could be to present the application of latent factor analyses on both, samples and distinct cell types, in a single place right at the beginning (lines 99ff). An integrated analysis could then deliver a unified, cross-validated SNAP based on donor variation and intercorrelated cell-specific variation (see the authors' statement lines 221-224) that is subsequently submitted to the impressive range of analyses that the

authors have lined up. Admittedly, this is a radical suggestion, but it has the potential to deliver a better argued, seamlessly flowing, and more impactful story.

We appreciate the close reading that went into making these thoughtful suggestions reimagining the presentation of results, and the willingness to propose considering an “admittedly radical” suggestion for an alternative organization.

We were able to incorporate most of the specific suggestions, though for several reasons it is still essential to describe the two key analyses – of person-to-person variation, then of single-cell-level variation within people – sequentially rather than (as proposed) simultaneously:

- Both analyses require much from readers to understand; this explanatory problem would be harder if the analyses were dropped on readers all together at once.
- The first (person-to-person variation) analysis makes the key insight that the neuronal and astrocytic changes are so highly correlated with each other (they appear in the same donors, to corresponding degrees). This insight motivates the subsequent focus on astrocytes and neurons in the single-cell-resolution analysis, which can go deeper (by virtue of the single-cell resolution) in recognizing key details of these changes, but would not on its own have been able to recognize the key larger pattern that the neuronal and astrocytic changes are implemented in a concerted manner.
- Though it is not primarily a methods paper, we do expect that the methods we describe here will be adopted by many other studies, and it is useful for readers to see how, though the two analyses align in their central findings, specific biological details were recognized more clearly or powerfully by one analysis or the other.

Though we have maintained the two-stage presentation of the analyses, we have addressed most of the specific details in the comments above::

...introduced with somehow weak and unconvincing arguments (line 203 "to better appreciate astrocyte variation" or line 285: "To assess the cell-type specificity of SNAP-a...)", leading to flawed terminology (SNAP-a and SNAP-n seem to be misnomers, as each lacks the neuronal and astrocytic component, respectively)...

We have removed these phrases. We do keep the terms SNAP-a and SNAP-n to refer to the distinct cell-type-specific contributions of astrocytes and neurons (respectively) to SNAP.

...tiring repetition ("identify biological processes" and GSEA: lines 146 and again lines 234; or Figs. 1I, J, 3D, E)...

We have removed this second reference to GSEA, thank you. The figures involve less repetition than it seems at first. **Fig. 1i-j** shows the association of the overall neuron-astrocyte gene-expression program with advancing age and schizophrenia case-

control status – motivating our focus on this particular program in the rest of the paper. **Fig. 3d-e** makes the important point that *both* the astrocytic *and* neuronal transcriptional changes associate with age and schizophrenia, *even when uncovered in separate and independent analyses* – and that in fact these associations are now much stronger with the refined characterization of these programs that the single-cell-variation (cNMF) analysis allows.

...eclectic exploration of the different SNAPs (see lines 300ff: why only SNAP and not SNAP-a, and -n? See also 315-357 with back and fourth references to figures).

We have corrected this to “SNAP-a and SNAP-n”, thank you. Also, to explain the figure callouts: In the genetic analyses described in the last section, we focused on genes recruited by SNAP-a and SNAP-n, as these two latent factors (informed by the single-cell-resolution latent-factor analysis) had associated even more strongly with age and schizophrenia case-control status than the composite SNAP measurement did, and are likely somewhat more precise for identifying the underlying set of genes.

Moreover, text (see lines 93-97 and lines 205-210) and graphs (for example Figs. 1D-E with Fig. 3A-C,F-H and Fig 1I,J with Fig 3D,E,I,J; Figs. S3-7 with S17) that conceptually belong together are scattered.

Since these sub-clusterings are mainly used to support *other* analyses (rather than this being a “cell atlas” paper), we present them together with the analyses they support.

A statement like in line 227-228 somehow undermines the value of what has been presented in the first half of the ms.

Thank you for pointing this out; we revised it to “Because cNMF2 is informed by variation in the single-astrocyte expression profiles, we consider it a more precise description of the astrocyte-specific gene-expression effects in SNAP, and refer to it here as SNAP-a.”

The following comments are more specific, and may guide the revision of the ms regardless of the previous fundamental suggestion.

2. A weak point relates to the new and exciting results showing the impact of age. The present version still seems centered on schizophrenia and the age-related findings appear somewhat as bycatch. This may be chronologically correct, but it is conceptually weak. It appears more adept and interesting to expose the focus on age AND disease already in the abstract and introduction. The authors should modify the ms accordingly, and they should check throughout the ms, where the presentation of

age-related results needs to be revamped and put on a par with schizophrenia-related findings. An example can be found in lines 117ff. Here, schizophrenia is mentioned first, and age appears more as an appendix. This could be changed to "One of the latent factors, LF4, was strongly impacted by age and schizophrenia...". Evidently, an additional focus on age also requires modifications to figures, where data are currently only stratified by disease, and where stratification by age should be added (e.g. Figs. 1H; 2C-E, S3-S7, S14, S18H, I). Specifically, the authors' data could reveal whether the fractions of nuclei attributed to specific cell types (subtypes of neurons and astrocytes) in the human brain decrease with age.

We appreciate the desire to see the aging results more strongly emphasized. The following reasons guide the current presentation:

- The primary focus of the work is SNAP, rather than the genomics of aging or schizophrenia *per se*. (All three reviewers encouraged this focus on SNAP itself.)
- The decline of SNAP captures a very large fraction (surprisingly, almost all) of the schizophrenia-associated gene-expression changes in the dIPFC, but only some of the age-associated changes. Thus, we think SNAP is a profound insight about schizophrenia in the dIPFC, but is only one of very many things that will be learned about brain aging.
- A truly aging-focused study would need to have a different design (than the current work), with much greater representation of the earlier and later decades of life (i.e. of donors younger than 40yo and older than 80yo, who are just 4% and 13% of our schizophrenia case-control cohort respectively). We worked hard in the manuscript to find a balance of reporting the interesting connection between SNAP and aging – without suggesting in any way that ours is a definitive study of brain aging (which it was not designed to be).
- The ability to connect SNAP genes to human-genetic findings (**Fig. 4 and Supplementary Note**) provides key evidence that SNAP is not merely “reactive” – that genetic protection and vulnerability arise from variation in the genes that SNAP regulates and thus from the efficacy of SNAP itself, including specifically (and surprisingly) from its astrocyte component (**Fig. 4**). Far more is known about the human genetics of schizophrenia (due to genetic studies of >80,000 individuals) than the genetics of cognitive aging. Thus, we are able to reach this important level of insight much more strongly in schizophrenia.

3. The title, abstract and the introductory paragraphs, which are key to attract readers' attention and to situate the work, need to be revised. This merits a bit more detail as I consider this as an important point:

3.1. Lines 1ff. The title is suboptimal, it seems to turn the concept upside down: who coordinates what? Neurons/astrocytes the program or the program their function? Moreover, the verb "coordinate" seems a bit teleologic and the important term "human" is missing. Just a suggestion: "Age- and schizophrenia-induced decline of correlated gene expression programs in human neurons and astrocytes".

We appreciate this point about teleology, and revised the title (and some other places in the text) to avoid using “coordinate” as a verb with a cellular actor, now simply describing the changes as “concerted”. (We are also hopeful that “human” will be implicit in “schizophrenia” in a short 75-character title.)

3.2. Lines 25ff: The first sentence seems cryptic/trivial, the second risks to put readers on the wrong track: It suggests that the authors aim to understand variation, which is not the case. A clear formulation of the goal and the approach is missing. As far as I understood, inter-individual and intercellular variation of gene expression represent a so far neglected "treasure trove", whose exploitation by the authors reveals correlated functions of brain cells and their decline in age and disease.

We appreciate this feedback and rewrote the introduction to emphasize the core goal and approach (and also to shorten to address length requirements).

3.3. Lines 26ff. The statement "To assess variation in cell-type specific gene expression programs..." appears a bit "weak" as aim, and it seems to anticipate the result (the discovery of a gene expression program) rather than to expose the "big question". The second part of the statement seems too detailed. Clearly here the focus on age AND disease should be mentioned.

We appreciate this feedback and rewrote the first three sentences of the abstract.

3.4. The abstract contains repetitive statements that make it hard to read and that dampen readers' interest and attention: lines 31-34: "In persons in whom neurons invested more ..." is followed by "astrocytes invested more transcript in...". This is more verbose than elegant.

We revised this sentence for clarity and removed the "invest" language in it (and elsewhere), replacing it with more-literal language. We maintained the parallel structure in this sentence even if it is verbose, as it is such a key idea that *distinct* sets of genes are co-regulated between neurons and astrocytes.

3.5. The introduction – in contrast to the abstract – opens with cell function enabled by cell collaboration (Lines 46ff). Then, the next paragraph (lines 54ff) goes from "In-free living species" (a term that is somewhat ill-defined) to "variation" without an explanation how this relates to the first paragraph.

We appreciate this feedback and reordered the points in the Introduction. (Also, in revising for length, we removed the introductory sentence about cell function enabled by cell collaboration.)

4. Lines 99-102: The authors should revise this pivotal part of the Results section. In its present form, it lacks key information that readers require to appreciate the results shown after.

First, the term "Inter-individual variation" appears out of nowhere and it is unclear what it refers to. So, one immediate solution is to change to "Inter-individual variation of gene expression...".

We revised this phrase to "Inter-individual variation in gene expression" as suggested.

Second, the authors need to show the phenomenon, i.e. that gene expression indeed varies. Otherwise, the latent factor analysis comes deus ex machina. None of the figures referred to up to this point (Fig. 1, S1-7) illustrates this key point. The authors should include a graph in Fig. 1 between panels E and F that shows inter-individual (and intercellular, see comment 1) variation of gene expression that they uncovered in their data set. For example, expression levels of selected transcripts in cell types and across all individuals or a more global parameter illustrating variation of cell-specific gene expression across individuals. A proper explanation and graphic display will alert readers of the phenomenon per se, and help to appreciate the authors' subsequent exploration of this valuable resource. Note, the statement in line 107 that "Ten latent factors together explained 30% of inter-individual variation..." is not understandable without a proper display of what inter-individual (and intercellular) variation of gene expression looks like.

We added a sentence at the beginning of this section with a simple descriptive statistic to capture this idea: "The data revealed substantial inter-variation in cell-type-specific gene expression levels, with highly expressed genes in each cell type exhibiting a median coefficient of variation (across donors) of about 15%." An abundant earlier literature already testifies to the extensive quantitative variation in gene expression that humans and human cells exhibit. Many figures in the paper (such as **Fig. 3**, **Fig. 4c-f**, and **Extended Data Figs. 18 and 22**) do provide specific examples, and a substantial descriptive figure seems unnecessary given that this idea has a long history that precedes our work.

5. Lines 117ff, 132ff, 500ff, Figs. 1H, S10A: At least to this referee, it is not entirely clear what is shown in these plots, last not least because the legends (line 500ff and 738ff) are too cursory. The authors should improve the description in the legends, and check for other instances where legends may not describe the figure content in sufficient detail.

Thank you for this feedback. We revised the legend for **Fig. 1h** to make it less technical and more clear. In the legend for **Extended Data Fig. 10a**, where we have much more room, we also added descriptions of each type of plot shown; in particular, we have added additional details to the legend for the q-q plots in the first column that we hope will also be helpful in the interpretation of **Fig. 1h**, whose legend now also points to **Extended Data Fig. 10a**.

6. Lines 136-138: The presentation of these valuable results appears as en passant and superficial. The rationale to do this experiment and the conclusion could be highlighted

a bit more. In the phrase "an analogous constellation of changes manifests" it is unclear what "analogous" and "changes" refer to.

We revised this sentence to make it more clear, and also elaborated on the rationale for this analysis and the interpretation of its results in the legend for **Extended Data Fig. 11**. We appreciate the enthusiasm for these results, though since they are just a form of validation that SNAP is real, a more-extended development in the main text is hard to accommodate in a paper of this length.

7. Lines 140-179 and 181-200: The titles and purposes of these chapters are unclear. In lines 140-179, this weakness originates from the terms "changes" and "concentration" (e.g. lines 160, 516, 733 etc.). It is not clear in each statement what the authors refer to. Do they mean changes due to variability/enrichment (so, it should read co-variation or co-variability) or changes due to schizophrenia and age. This must be clarified.

Thank you for pointing out the need for clarification. We have removed the ambiguous word "change" in these (and additional) locations, whenever referring to the latent factor (as opposed to disease- or age-related changes): e.g. in the main text, "In glutamatergic neurons, LF4 also appeared to involve genes encoding postsynaptic components..."; and in the **Fig. 2b** legend, "Concentrations of synaptic gene sets (as annotated by SynGO) in LF4's neuronal components...".

Moreover, there seems to be some overlap with the subsequent chapter (e.g. lines 192-194), where again schizophrenia- and age-related changes are presented (lines 151; 164-166 etc.). Symptomatic for this intermingling is the fact that references for figures go back and fourth.

We found that if we presented such results *only* through the mathematical abstraction of latent factor analysis and then post hoc gene-set enrichment analysis, then the relationships did not feel real to many biologist readers. Apprehending these neuron-astrocyte relationships in concrete biological terms (in **Fig. 2e** and the short "Concerted neuron-astrocyte expression" chapter) makes these relationships real and concrete to many biologists, and also shows (in a way that latent-factor analysis does not) the surprisingly large magnitude of quantitative inter-individual variation in gene expression levels that is involved.

In lines 181-191, the weak terms are "coordinated" (see comment 3.1) and "investments". What does investment mean? Isn't the main finding a "correlated expression of synapse-related genes in neurons and astrocytes"?

We revised these lines to use more-literal language, e.g. "The proportion of astrocyte gene expression devoted to each of these three astrocyte activities strongly correlated with the proportion of neuronal gene expression devoted to synaptic components."

Further, it is not clear how the genes to be analyzed here were chosen? Based on their presence in LF4?

Yes, these gene sets were selected based on their strong contributions (by gene loadings) to the neuronal or astrocyte components of LF4. We have added a call-out to the section of the Methods that describes the selection of these genes (**Methods: Selected gene sets**).

What does the sentence in lines 188-191 mean? Did the authors assume that the correlations were only valid for schizophrenic patients?

We previously had this sentence because we wanted to make clear that this result wasn't simply due to the case-control differences – i.e. to clarify that the relationship is also present among healthy individuals. Anyway, we removed this sentence as we revised the manuscript for length, and because the new results on aging-associated changes already make this clear.

8. Lines 159-162; 516-520, Figs. 2B, S9L: The enrichment of synaptic components in LF4 is impressive, however the presentation of the data should be improved. The sunburst plots shown in Fig. S9L could be moved to the main figure 2B, as they illustrate more vividly the loading of synaptic genes in LF4 than the more conventional GO term plots (Fig. 2B). An illustration of astrocyte-specific gene enrichment should also be added to the main figure 2 between panel B and C to highlight this finding and to set the stage for subsequent analyses.

We find that readers tend to have diverse opinions about the informativeness of sunburst plots, which in any event don't fit into **Fig. 2**. We decided to keep **Fig. 2b** as the main figure panel, as this presentation allows direct and quantitative comparison of the enrichments of individual SynGO terms between glutamatergic and GABAergic neurons, supporting the statements in the text. (The sunburst plots in **Extended Data Fig. 9I** efficiently summarize the enrichments of related SynGO terms, but require cross-referencing with the SynGO database to find the underlying terms in each slice.) We keep **Fig. 2b** and **Fig. 2c** as consecutive panels as they both relate to neuronal changes, with **Fig. 2c** providing detail on the cell-type specificity of a key subset of the changes in **Fig. 2b**. The SynGO annotations/analysis are in general far less informative for astrocyte biology than for neuronal biology; we refer the reader to **Supplementary Table 4** for a list of gene sets enriched in the astrocyte component of LF4, but this is in general a less-illuminating analysis that we think doesn't merit promotion to the main figure.

9. Lines 170ff, Fig. 2D: As stated in the first round of revision, at least half of the genes indicated by the authors in Fig. 2D are related to fatty acid biosynthesis or regulation (GPAM, SCD, SREBF1, ACACA, ACACB, FASN, ELOVL6) and for some, the relation to fatty acid or cholesterol synthesis is far-fetched (NFYC, KPNB1, ERLIN2). The authors acknowledge this partially in the text, but they have not changed the term in the figures. For the sake of precision I would recommend to replace in the figs 2E and S13 the axis label "regulation of cholesterol biosynthesis" by "cholesterol/fatty acid biosynthesis".

The Gene Ontology categories and annotations (which are what we used here) do include genes with additional functions, or with indirect or regulatory roles on the biology in question. However, we hesitate to rename or edit these categories in the current work, since they are also used as analytical entities by many other researchers. The protein products of the three genes mentioned in the suggestion (*NFYC*, *KPNB1*, *ERLIN2*) appear to be in the “regulation of cholesterol biosynthetic process” GO category (GO:0045540) because they have all been found to regulate SREBP2, the transcription factor that regulates cholesterol biosynthesis gene expression (for which the encoding gene *SREBF2* contributes to the astrocyte component of SNAP). NFYC is a transcription factor that interacts with SREBP2 (Liu et al. 2023) and whose binding site upstream of the human *SREBF2* gene is also required for *SREBF2* transcription (Sato et al. 1996); KPNB1 (also known as importin-beta) interacts directly with SREBP2 and mediates its import into the nucleus (Nagoshi et al. 1999; Lee et al. 2003); and ERLIN2 regulates SREBP2 by sequestering its complex with SCAP and INSIGs in the endoplasmic reticulum (Huber et al. 2013; Luo, Yang, and Song 2019).

So, we revised to say, “Gene sets with roles in fatty acid and cholesterol biosynthesis and export, including genes that encode the SREBP1 and SREBP2 transcription factors and their **regulators and targets...**”

10. Lines 212ff: Here the same point of criticism applies as outlined in comment 3 concerning lines 99-102 and Fig. 1. The authors write about the interindividual variation among astrocytes but they do not illustrate this. Instead, they jump right into latent factor analysis. Could this be illustrated in Fig. 3?

We have now added a sentence earlier in the Results section with a simple descriptive statistic to capture this idea: “The data revealed substantial inter-variation in cell-type-specific gene expression levels, with highly expressed genes in each cell type exhibiting a median coefficient of variation (across donors) of about 15%.” An abundant earlier literature already testifies to the extensive quantitative variation in gene expression that humans and human cells exhibit. Many figures in the manuscript (such as **Fig. 3k**, **Fig. 4d-f**, and **Extended Data Figs. 18 and 22**) provide many specific examples, and a substantial descriptive figure seems unnecessary given that this idea has a long history that precedes our work.

In this context, I wonder whether it would make sense to show in Fig. S18 in analogy to the latent factor analysis for human samples a graph similar as in Fig. S8A,B which indicates how many latent factors explain which percentage of variation? Evidently, this comment is subordinate to comment 1.

cNMF analysis (the method used in **Extended Data Fig. 18**) does not permit a decomposition of the variance into non-overlapping factor-associated components (as in **Extended Data Fig. 8**), since the cNMF factors are not orthogonal to one another. However, we can (and now do) report both the total percent of variance explained by the factors collectively, which is 25% for the single-astrocyte cNMF analysis and 44% for the single-neuron cNMF analysis.

11. Line 222-224: It seems a bit strange that the authors mention only two computational approaches, although they used in fact three, as indicated in Fig. S19. The data shown in this figure could be exposed in a bit more detail in the main text.

We primarily used just these two computational approaches (PEER and cNMF) to identify and characterize SNAP. While we did corroborate the PEER and cNMF results with a third approach (CNA) (**Extended Data Fig. 19**), the primary results arose from the first two approaches, as CNA does not offer the same opportunities for downstream analyses. We have revised this sentence to “the strong co-expression relationships in SNAP were thus robust to the computational approach used.”

12. Lines 226ff. The paragraph is unclear, last not least because in Lines 227-228 and 229-231, "than" is missing, and so it is not known what is compared here. Assuming that the authors indicate that SNAP-a is more precise than SNAP, this conclusion seems a bit premature given that in the following paragraphs, the authors explore "the biology supported by SNAP-a" (a phrase that could be changed to a more clear statement) for example by exposing the top 20 genes etc.

Thank you, we revised this sentence to, “Because cNMF2 is informed by variation in the single-astrocyte expression profiles, we consider it a more precise description of the astrocyte-specific gene-expression effects in SNAP, and refer to it here as SNAP-a.” We also removed the sentence that includes “the biology supported by SNAP-a” as we revised to address the manuscript length target, and we now proceed directly to describing the gene set enrichment analysis results: “The strongest positive gene-set associations to SNAP-a involved...”

Extended Data Fig. 20 already enumerates/exposes (and further characterizes) the top 20 SNAP-a genes, as we now make more clear in the main text: “The 20 genes most strongly associated with SNAP-a (**Extended Data Fig. 20**)...”

Here, one misses a comparison to SNAP and the GO terms that were explored at length in Fig. 2 and all associated supplementary figure. Surprisingly, the authors do not expose whether SNAP-a contains lipid synthesis genes that figured so prominently in the first part of the ms.

The genome-wide gene loadings between SNAP (in astrocytes) and SNAP-a are extremely high, as we show in **Extended Data Fig. 18c**. To avoid redundant analyses, we emphasize new analyses that supplement the earlier insights with new information made visible by the single-cell resolution of the second analysis. For example, we specifically examine lipid-synthesis genes in the context of astrocyte heterogeneity in **Extended Data Fig. 24d**, in which we show that case-control expression differences for these genes were shared across astrocyte subtypes. We also show in **Extended Data Fig. 25** that expression of SNAP-a associated strongly (at the single-cell level) with the expression levels and predicted target genes of SREBP1 and its well-known transcriptional targets (Horton et al. 2003; Eberlé et al. 2004) in astrocytes (**Extended Data Fig. 25**); *SREBF1* is one of the core genes that drive enrichment of the lipid-synthesis gene set in the astrocyte component of SNAP (**Fig. 2d**).

In the main figure (**Fig. 3k**) we focus on the synaptic gene sets presented for SNAP-a, as these are the gene sets that most strongly drive this latent factor (i.e. have the strongest gene loadings). This likely reflects the slight differences in the two approaches used: the latent factor analysis used to infer SNAP was based on gene-expression variation between individual donors that are coordinated across different cell types, while the latent factor analysis used to infer SNAP-a was based on gene-expression variation between individual astrocytes.

13. Lines 236-261: The text should be rewritten. In line 243, the reference to Fig. S20 is unclear as the graphs shown seem to have nothing to do with what the text refers to. Moreover, parts of this and of the next block (lines 255-261) should be moved to the Discussion.

We greatly streamlined this text, also to address the manuscript-length requirement. We also moved the callout to this supplementary figure earlier in that sentence, so that it is more clear that it relates to “the 20 genes most strongly associated with SNAP-a (**Extended Data Fig. 20**)”.

14. Lines 269-283: The exploration of reactive astrocyte markers is a highly welcome element, but in the current presentation, it looks more like an add-on. The authors focus on the question of continuous versus discrete state shifts, but surprisingly, schizophrenia is not mentioned in the text although in Fig. S22, the authors stratify by schizophrenia. An evident question is how schizophrenia and age affect established reactive astrocyte markers in the authors' dataset - even beyond their association with SNAP-a. In the relevant paragraph (lines 276ff), the authors refer to SNAP-a, but not to disease or age and not to the set of published reactive astrocyte markers.

In **Extended Data Fig. 21**, we in fact did use the set of reactive astrocyte markers that were published in (Liddelow et al. 2017). The expression level of these markers (as a group) in astrocytes does indeed associate with age and schizophrenia – as one might expect given their very strong relationships to SNAP expression – but SNAP expression appears to be what mediates the more-modest relationships to age and schizophrenia.

15. Lines 564-573, Fig. 3K: It is unclear how the figure can illustrate "coordinated expression" between astrocytes and neurons. The patterns shown in the two columns seem rather distinct.

Your inference from the figure itself is correct – the genes recruited by SNAP are different in astrocytes vs. neurons – so we decided that it was the legend that was confusing, and revised it for clarity, removing words whose interpretation might be ambiguous here (e.g. “coordinated”) and now explaining directly, “One set of genes (SNAP-a, above) exhibits co-regulation in astrocytes; a distinct set of genes (SNAP-n, below) exhibits co-regulation in neurons...”.

Minor

- **Lines 65ff and 76ff: The authors should consider to reunite the statements distributed in the two places. In any case, the explanation and references given in lines 76-79 should be indicated the first place where these neurons are mentioned.**

We simplified the statement in the Introduction to reduce redundancy, thank you.

- **Lines 28, 66 and 80. The number of donors and the age range are mentioned three times within two pages, which is clearly in the way of a streamlined introduction.**

Thank you for pointing this out; we removed the redundant reference to this in the Introduction section, which is now substantially more streamlined.

- **Lines 103-105, lines 108-110: The author should revise the sentences, which seem verbose and difficult to understand.**

We revised both sentences to make them easier to understand. Thank you for this suggestion.

- **Lines 107-112. Here or elsewhere in the text, there is no reference to panel Fig. S8F.**

Due to length limitations in the main text, we moved the sentence specifically about panel F of **Extended Data Fig. 8** to the Methods section, where we elaborate on the latent factor analysis approach (**Methods: Latent factor analysis: snRNA-seq data**).

- **The authors should consider to move panels F-K in Fig. S09 to a new figure. This would also avoid the somewhat odd reference to a preceding figure on line 136 following the introduction of Fig. S10 (line 134).**

Thank you for this suggestion. All of the panels in **Extended Data Fig. 9** examine the relationship of LF4 to donor covariates, so we would like to keep them together. We have revised the paragraph and figure numbering so that **Extended Data Figs. 9, 10, and 11** are now called out in order.

- **Lines 205-206: "Analysis recognized" could be improved to "Our snRNASeq data recapitulated transcriptional profiles of distinct subtypes of adult astrocytes described previously..."**.

Thank you for this suggestion, but we kept this sentence as currently written since these are analytical constructs, and since it is more the global pattern (than the specific transcriptional profiles) described in earlier studies that is found.

- **Line 207: The statement could be complemented by "..., which populate the gray matter and represent the most frequent subtype in our samples."**

We have added this, thank you for the suggestion.

- Lines 213-15: The parentheses surrounding the sentence can be removed.

We removed these parentheses and have also moved this sentence to the Methods section as we edited the manuscript for length (**Methods: Analysis of astrocyte and glutamatergic L5 IT neuron gene-expression programs: Consensus non-negative matrix factorization**).

- Line 224, line 232: Here and at other places, the authors refer back and fourth to figures (first S19, then back to S18F-I), which may confuse readers, notably with respect to content design and coherency of figures.

These figure panels are grouped thematically: for example, **Extended Data Fig. 18** describes the properties of SNAP-a as inferred by cNMF, while **Extended Data Figure 19** describes the results of a separate computational approach we used to analyze expression variation in single-astrocytes (CNA). Sometimes we do point back to an earlier figure later in the manuscript, when that earlier result is helpful for thinking about a later result. In the revised manuscript we have sought to minimize confusion by adding call-outs to specific panels in each figure directly after they are described in the text.

- Line 252: The numbers shown in parantheses are not well described. Which test was used, what does q stand for?

Thank you for pointing out the need to clarify this. The p- and q-values are standard outputs from gene set enrichment analyses (GSEA, (Subramanian et al. 2005)) and refer respectively to the nominal p-value (i.e. statistical significance of the enrichment score for a particular gene set) and false discovery rate q-value (i.e. the estimated probability that the normalized enrichment score for a particular gene set represents a false positive result). To clarify that these are the standard GSEA metrics, we have added a reference to GSEA where we report the values “(p = 3.5×10^{-4} , q = 0.015 by GSEA)”.

- Lines 263-267: This paragraph stands alone without any relation to the previous or next ones. The authors should move it to a more suitable place in the Results with better integration in the context.

Thank you for this suggestion and we agree. We have removed this paragraph, as the next paragraph on reactive astrocytes makes a similar point: “At the single-astrocyte level, SNAP-a expression exhibited continuous, quantitative variation rather than discrete state shifts (**Extended Data Fig. 18f-g**), consistent with observations of abundant astrocyte biological variation less extreme than experimentally polarized states”.

- Lines 483, 492: "Experimental approach" and "Computational approach" sound too generic. "Experimental approach of human sample preparation" and "Design of latent factor analysis"

We revised both of these panel titles to make them more concrete and specific: "Generation of snRNA-seq data" and "Latent factor analysis".

- Lines 481-485: Here and throughout, authors should consider exclusive use of past tense.

Thank you, we revised to past tense wherever describing experiments that were done or analyses that were performed.

- Line 503: The authors should state "Correlation of with age" assuming that the indicated values are correlation coefficients. The latter should be mentioned in the legend.

Thank you for pointing this out. We now note here (as well as in other applicable figure legends) that the correlation values displayed are Spearman's rank correlation coefficients (Spearman's ρ).

- Line 513: Is "...in both cell types..." correct? Should this read "...in each cell type..."?

Yes, "both cell types" is correct, as each plot displays values from a different set of genes depending on which genes have shared expression in each indicated pair of cell types. The number of shared genes plotted in each pairwise comparison is listed in the figure legend.

- Line 635: The explanation of BA46 as "Brodmann area 46" given here should be moved to line 483. The abbreviation is first used in panel A of Fig. 1. Here, and in all further instances, only the abbreviation should be used for consistency.

We have removed "Brodmann area 46" from this line, as the abbreviation is already explained in the first sentence of the Results section.

- Lines 696-697: The phrase should be changed to "from Hodge et al. (2019)".

Thanks for pointing this out, we have reformatted the reference to this work to the citation style of this journal.

- Fig. S18D, lines 840-841: The two color codes need to be explained. Is this schizophrenic versus control?

Thank you for pointing out this omission. We have added these explanations to **Extended Data Fig. 18d**. Yes, this is the same color scheme used throughout the manuscript to represent schizophrenia cases (purple) and controls (green).

- In Fig. S18E: The authors should consider to apply the same color spectrum used for LF4 (going from blue to yellow) also for cNMF2 (which goes now from blue to something like light blue), obviously by re-mapping to a different value range. Thereby, the similarity of the distributions may become more visible.

We used different color scales for LF4 and cNMF2 to illustrate that these are different latent factors inferred by different approaches. We would like to keep the current value ranges, as they maximize visual perception of the range of values plotted in each UMAP.

- Figure legends: It seems unnecessary to repeat in each figure legend and for each figure panel what boxplots etc. indicate, unless this is stipulated by the journal's policy. On the other hand, violin plots are mentioned only once in legend of Fig. S1 although they are also shown in main Fig. 1, and there is no explanation what they show.

These descriptions in the figure legends were indeed included to comply with the journal's policy on reporting full descriptions of statistical parameters and figure elements (Editorial Policy Checklist). We also now report the use of violin plots in the legends for **Fig. 1j** and **Extended Data Fig. 1d-f**, thank you.

Responses to Reviewer 3 comments, suggestions, and ideas

I read the revised paper and the response to reviewers by McCarroll and colleagues. In my assessment, the authors have done an outstanding job with revisions, including expanding the scope of the work to include consideration of aging as requested by another reviewer. I have no further comments other than to congratulate the authors on an excellent study that will surely stimulate many follow up studies by the field.

We appreciate this assessment of the work and manuscript.

References

- Eberlé, Delphine, Bronwyn Hegarty, Pascale Bossard, Pascal Ferré, and Fabienne Foufelle. 2004. "SREBP Transcription Factors: Master Regulators of Lipid Homeostasis." *Biochimie* 86 (11): 839–48.
- Horton, Jay D., Nila A. Shah, Janet A. Warrington, Norma N. Anderson, Sahng Wook Park, Michael S. Brown, and Joseph L. Goldstein. 2003. "Combined Analysis of Oligonucleotide Microarray Data from Transgenic and Knockout Mice Identifies Direct SREBP Target Genes." *Proceedings of the National Academy of Sciences of the United States of America* 100 (21): 12027–32.
- Huber, Michael D., Paul W. Vesely, Kaustuv Datta, and Larry Gerace. 2013. "Erlins Restrict SREBP Activation in the ER and Regulate Cellular Cholesterol Homeostasis." *The Journal of Cell Biology* 203 (3): 427–36.
- Lee, Soo Jae, Toshihiro Sekimoto, Eiki Yamashita, Emi Nagoshi, Atsushi Nakagawa, Naoko Imamoto, Masato Yoshimura, et al. 2003. "The Structure of Importin-Beta Bound to SREBP-2: Nuclear Import of a Transcription Factor." *Science* 302 (5650): 1571–75.
- Liddelow, Shane A., Kevin A. Guttenplan, Laura E. Clarke, Frederick C. Bennett, Christopher J. Bohlen, Lucas Schirmer, Mariko L. Bennett, et al. 2017. "Neurotoxic Reactive Astrocytes Are Induced by Activated Microglia." *Nature* 541 (7638): 481–87.
- Liu, Zefu, Xianchong Zheng, Jiawei Chen, Lisi Zheng, Zikun Ma, Lei Chen, Minhua Deng, et al. 2023. "NFYC-37 Promotes Tumor Growth by Activating the Mevalonate Pathway in Bladder Cancer." *Cell Reports* 42 (8): 112963.
- Luo, Jie, Hongyuan Yang, and Bao-Liang Song. 2019. "Mechanisms and Regulation of Cholesterol Homeostasis." *Nature Reviews. Molecular Cell Biology* 21 (4): 225–45.
- Nagoshi, E., N. Imamoto, R. Sato, and Y. Yoneda. 1999. "Nuclear Import of Sterol Regulatory Element-Binding Protein-2, a Basic Helix-Loop-Helix-Leucine Zipper (bHLH-Zip)-Containing Transcription Factor, Occurs through the Direct Interaction of Importin Beta with HLH-Zip." *Molecular Biology of the Cell* 10 (7): 2221–33.
- Sato, R., J. Inoue, Y. Kawabe, T. Kodama, T. Takano, and M. Maeda. 1996. "Sterol-Dependent Transcriptional Regulation of Sterol Regulatory Element-Binding Protein-2." *The Journal of Biological Chemistry* 271 (43): 26461–64.
- Subramanian, Aravind, Pablo Tamayo, Vamsi K. Mootha, Sayan Mukherjee, Benjamin L. Ebert, Michael A. Gillette, Amanda Paulovich, et al. 2005. "Gene Set Enrichment Analysis: A Knowledge-Based Approach for Interpreting Genome-Wide Expression Profiles." *Proceedings of the National Academy of Sciences of the United States of America* 102 (43): 15545–50.